# A Path to Simpler Models Starts With Noise

**Lesia Semenova    Harry Chen    Ronald Parr    Cynthia Rudin**
Department of Computer Science, Duke University
{lesia.semenova,harry.chen084,ronald.parr,cynthia.rudin}@duke.edu

## Abstract

The Rashomon set is the set of models that perform approximately equally well on a given dataset, and the Rashomon ratio is the fraction of all models in a given hypothesis space that are in the Rashomon set. Rashomon ratios are often large for tabular datasets in criminal justice, healthcare, lending, education, and in other areas, which has practical implications about whether simpler models can attain the same level of accuracy as more complex models. An open question is why Rashomon ratios often tend to be large. In this work, we propose and study a mechanism of the data generation process, coupled with choices usually made by the analyst during the learning process, that determines the size of the Rashomon ratio. Specifically, we demonstrate that noisier datasets lead to larger Rashomon ratios through the way that practitioners train models. Additionally, we introduce a measure called pattern diversity, which captures the average difference in predictions between distinct classification patterns in the Rashomon set, and motivate why it tends to increase with label noise. Our results explain a key aspect of why simpler models often tend to perform as well as black box models on complex, noisier datasets.

## 1  Introduction

It is possible that for many datasets, a simple predictive model can perform as well as the best black box model – we simply have not found the simple model yet. Interestingly, however, we may be able to infer whether such a simple model exists without finding it first, and we may be able to determine conditions under which such simple models are likely to exist.

We already know that many datasets exhibit the "Rashomon effect" [7], which is that for many real-world tabular datasets, many models can describe the data equally well. If there are many good models, it is more likely that at least one of them is simple (e.g., sparse) [39]. Thus, a key question in determining the existence of simpler models is to understand why and when the Rashomon effect happens. This is a difficult question, and there has been little study of it. The literature on the Rashomon effect has generally been more practical, showing either that the Rashomon effect often exists in practice [39, 13, 45], showing how to compute or visualize the set of good models for a given dataset [52, 14, 15, 1, 32, 53, 54, 48], or trying to reduce underspecification by learning a diverse ensemble of models [25, 37]. However, no prior works have focused on understanding what causes this phenomenon in the first place.

Our thesis is that *noise* is both a theoretical and practical motivator for the adoption of simpler models. Specifically, in this work, we refer to noise in the generation process that determines the labels. In noisy problems, the label is more difficult to predict. Data about humans, such as medical data or criminal justice data, are often noisy because many things worth predicting (such as whether someone will commit a crime within 2 years of release from prison, or whether someone will experience a medical condition within the next year) have inherent randomness that is tied to random processes in the world (Will the person get a new job? How will their genetics interact with their diet?). It might sound intuitive that noisy data would lead to simpler models being useful, but this is not

37th Conference on Neural Information Processing Systems (NeurIPS 2023).

something most machine learning practitioners have internalized – even on noisy datasets, they often use complicated, black box models, to which post-hoc explanations are added. Our work shows how practitioners who understand the bias-variance trade-off naturally gravitate towards more interpretable modes in the presence of noise.

We propose a *path* which begins with noise, is followed by decisions made by human analysts to compensate for that noise, and that ultimately leads to simpler models. In more detail, our path follows these steps: 1) Noise in the world leads to increased variance of the labels. 2) Higher label variance leads to worse generalization (larger differences between training and test/validation performance). 3) Poor generalization from the training set to the validation set is detected by analysts on the dataset using techniques such as cross-validation. As a result, the analyst compensates for anticipated poor test performance in a way that follows statistical learning theory. Specifically, they choose a simpler hypothesis space, either through soft constraints (i.e., increasing regulation), hard constraints (explicit limits on model complexity, or model sparsification), or by switching to a simpler function class. Here, the analyst may lose performance on the training set but gain validation and test performance. 4) After reducing the complexity of the hypothesis space, the analyst's new hypothesis space has a larger *Rashomon ratio* than their original hypothesis space. The Rashomon ratio is the fraction of models in the function class that perform close to the empirical risk minimizer. It is the fraction of functions that performs approximately-equally-well to the best one. This set of "good" functions is called the Rashomon set, and the Rashomon ratio measures the size of the Rashomon set relative to the function class. This argument (that lower complexity function classes leads to larger Rashomon ratios) is not necessarily intuitive, but we show it empirically for 19 datasets. Additionally, we prove this holds for decision trees of various depths under natural assumptions. The argument boils down to showing that the set of non-Rashomon set models grows exponentially faster than the set of models inside the Rashomon set. As a result, since the analyst's hypothesis space now has a large Rashomon ratio, a relatively large fraction of models that are left in the simpler hypothesis are good, meaning they perform approximately as well as the best models in that hypothesis space. From that large set, the analyst may be able to find even a simpler model from a smaller space that also performs well, following the argument of Semenova et al. [39]. As a reminder, in Step 3 the analysts discovered that using a simpler model class improves test performance. This means that *these simple models attain test performance that is at least that of the more complex (often black box) models from the larger function class they used initially.*

In this work, we provide the mathematics and empirical evidence needed to establish this path, focusing on Steps 1, 2, and 4 because Step 3 follows directly (however, we provide empirical evidence for Step 3 as well). Moreover, for the case of ridge regression with additive attribute noise, we prove directly that adding noise to the dataset results in an increased Rashomon ratio. Specifically, the additive noise acts as $\ell_2$-regularization, thus it reduces the complexity of the hypothesis space (Step 3) and causes the Rashomon ratio to grow (Step 4).

Even if the analyst does not reduce the hypothesis space in Step 3, noise still gives us larger Rashomon sets. We show this by introducing *pattern diversity*, the average Hamming distance between all classification patterns produced by models in the Rashomon set. We show that under increased label noise, the pattern diversity tends to increase, which implies that when there is more noise, there are more differences in model predictions, and thus, there could be more models in the Rashomon set. Hence, a much shorter version of the path also works: Noise in the world causes an increase in pattern diversity, which means there are more diverse models in the Rashomon set, including simple ones.

It is becoming increasingly common to demand interpretable models for high-stakes decision domains (criminal justice, healthcare, etc.) for *policy* reasons such as fairness or transparency. Our work is possibly the first to show that the noise inherent in many such domains leads to *technical* justifications for demanding such models.

## 2    Related Work

**Rashomon set**. The Rashomon set, named after the Rashomon effect coined by Leo Breiman [7], is based on the observation that often there are many equally good explanations of the data. When these are contradictory, the Rashomon effect gives rise to predictive multiplicity [30, 6, 18]. Rashomon sets have been used to study variable importance [15, 14, 42], for characterizing fairness [40, 9, 2], to improve robustness and generalization, especially under distributional shifts [37, 25], to study

connections between multiplicity and counterfactual explanations [36, 53, 8], and to help in robust decision making [46]. Some works focused on trying to compute the Rashomon set for specific hypothesis spaces, such as sparse decision trees [52], generalized additive models [54], and decision lists [32]. Other works focus on near-optimality to find a diverse set of solutions to mixed integer problems [1], a set of targeted predictions under a Bayesian model [23], or estimate the Rashomon volume via approximating model in Reproducing Kernel Hilbert Space [31]. Black et al. [6] shows that the predictive multiplicity metric defined as expected pairwise disagreement increases with expected variance over the models in the Rashomon set. On the contrary, we focus on probabilistic variance in labels in the presence of noise.

**Metrics of the Rashomon set.** To characterize the Rashomon set, multiple metrics have been proposed [39, 38, 30, 49, 18, 6]. The Rashomon ratio [39], and the pattern Rashomon ratio [38] measure the Rashomon set as a fraction of models or predictions within the hypothesis space; ambiguity and discrepancy [30, 49] indicate the number of samples that received conflicting estimates from models in the Rashomon set; Rashomon capacity [18] measures the Rashomon set for probabilistic outputs. Here, we focus on the Rashomon ratio and pattern Rashomon ratio. We also introduce pattern diversity. Pattern diversity is close to expected pairwise disagreement (as in Black et al. [6]), however, it uses unique classification patterns (see Appendix G).

**Learning with noise.** Learning with noisy labels has been extensively studied [34], especially for linear regression [5] and, more recently, for neural networks [44] to understand and model effects of noise. Stochastic gradient descent with label noise acts as an implicit regularizer [12] and noise has been added to hidden units [35], labels [41], or covariances [50] to prevent overfitting in deep learning. When the labels are noisy, constructing robust loss [16], adding a slack variable for each training sample [19], or early stopping [27] also helps to improve generalization. In this work, we study why simpler models are often suitable for noisier datasets from the perspective of the Rashomon effect.

## 3 Notation and Definitions

Consider a training set of $n$ data points $S = \{z_1, z_2, ..., z_n\}$, such that each $z_i = (x_i, y_i)$ is drawn i.i.d. from an unknown distribution $\mathcal{D}$, where $\mathcal{X} \subset \mathbb{R}^m$, and we have binary labels $\mathcal{Y} \in \{-1, 1\}$. Denote $\mathcal{F}$ as a hypothesis space, where $f \in \mathcal{F}$ obeys $f : \mathcal{X} \to \mathcal{Y}$. Let $\phi : \mathcal{Y} \times \mathcal{Y} \to \mathbb{R}^+$ be a 0-1 loss function, where for point $z = (x, y)$ and hypothesis $f$, the loss function is $\phi(f(x), y) = \mathbb{1}_{[f(x) \neq y]}$. Finally, let $\hat{L}(f)$ be an empirical risk $\hat{L}(f) = \frac{1}{n} \sum_{i=1}^{n} \phi(f(x_i), y_i)$, and let $\hat{f}$ be an empirical risk minimizer: $\hat{f} \in \arg\min_{f \in \mathcal{F}} \hat{L}(f)$. If we want to specify the dataset on which $\hat{f}$ was computed, we will indicate it by an index, $\hat{f}_S$.

The *Rashomon set* contains all models that achieve near-optimal performance and can be defined as:

**Definition 1** (Rashomon set). *For dataset $S$, a hypothesis space $\mathcal{F}$, and a loss function $\phi$, given $\theta \geq 0$, the Rashomon set $\hat{R}_{set}(\mathcal{F}, \theta)$ is:*

$$\hat{R}_{set}(\mathcal{F}, \theta) := \{f \in \mathcal{F} : \hat{L}(f) \leq \hat{L}(\hat{f}) + \theta\},$$

*where $\hat{f}$ is an empirical risk minimizer for the training data $S$ with respect to loss function $\phi$: $\hat{f} \in \arg\min_{f \in \mathcal{F}} \hat{L}(f)$, and $\theta$ is the Rashomon parameter.*

Rashomon parameter $\theta$ determines the risk threshold ($\hat{L}(\hat{f}) + \theta$), such that all models with risk lower than this threshold are inside the set. For instance, if we stay within 1% of the accuracy of the best model, then $\theta = 0.01$. Given parameter $\gamma > 0$, we extend the definition to the true risk by defining the *true Rashomon set*, containing all models with a bound on true risk $R_{set}(\mathcal{F}, \gamma) = \{f \in \mathcal{F} : L(f) \leq L(f^*) + \gamma\}$, where $f^*$ is optimal model, $f^* = \arg\min_{f \in \mathcal{F}} L(f)$.

In this work, we study *how noise influences the Rashomon set* and choices that practitioners make in the presence of noise. We measure the Rashomon set in different ways, including the Rashomon ratio, pattern Rashomon ratio, and pattern diversity (defined in Section 6). For a discrete hypothesis space, the *Rashomon ratio* [39] is the ratio of the number of models in the Rashomon set compared to the hypothesis space, $\hat{R}_{ratio}(\mathcal{F}, \theta) = \frac{|\hat{R}_{set}(\mathcal{F}, \theta)|}{|\mathcal{F}|}$, where $|\cdot|$ denotes cardinality. It is possible to weight the hypothesis space by a prior to define a weighted Rashomon ratio if desired.

Given a hypothesis $f$ and a dataset $S$, a predictive pattern (or pattern) $p$ is the collection of outcomes from applying $f$ to each sample from $S$: $p^f = [f(x_1), ..., f(x_i), .., f(x_n)]$. We say that pattern $p$ is achievable on the Rashomon set if there exists $f \in \hat{R}_{set}(\mathcal{F}, \theta)$ such that $p^f = p$. Let *pattern Rashomon set* $\pi(\mathcal{F}, \theta) = \{p^f : f \in \hat{R}_{set}(\mathcal{F}, \theta), p^f = [f(x_i)]_{i=1}^n\}$ be all unique patterns achievable by functions from $\hat{R}_{set}(\mathcal{F}, \theta)$ on dataset $S$. Finally, let $\Psi(\mathcal{F})$ be the *pattern hypothesis set*, meaning that it contains all patterns achievable by models in the hypothesis space, $\Psi(\mathcal{F}) = \{p^f : f \in \mathcal{F}, p^f = [f(x_i)]_{i=1}^n\}$. The pattern Rashomon ratio is the ratio of patterns in the pattern Rashomon set to the pattern hypothesis set: $\hat{R}_{ratio}^{pat}(\mathcal{F}, \theta) = \frac{|\pi(\mathcal{F}, \theta)|}{|\Psi(\mathcal{F})|}$.

In the following sections, we walk along the steps of our proposed path. Rather than trying to prove these points for every possible situation (which would be volumes beyond what we can handle here), we aim to find at least some way to illustrate that each step is reasonable in a natural setting.

## 4 Increase in Variance due to Noise Leads to Larger Rashomon Ratios

### 4.1 Step 1. Noise Increases Variance

One would think that something as simple as uniform label noise would not really affect anything in the learning process. In fact, we would expect that adding such noise would just uniformly increase the losses of all functions, and the Rashomon set would stay the same. However, this conclusion is (surprisingly) not true. Instead, noise adds variance to the loss, which, in turn, prevents us from generalizing.

For infinite data distribution $\mathcal{D}$ consider uniform label noise, where each label is flipped independently with probability $\rho < \frac{1}{2}$. If $\tilde{y}$ is a flipped label, $P(\tilde{y} \neq y) = \rho$. If the empirical risk of $f$ is over $\frac{1}{2}$ after adding noise, we transform $f$ to $-f$. For a given model $f \in \mathcal{F}$ let $\sigma^2(f, \mathcal{D})$ be the variance of the loss, meaning that $\sigma^2(f, \mathcal{D}) = \text{Var}_{z \sim \mathcal{D}} l(f, z)$. We show in the following theorem that, for a given $f \in \mathcal{F}$, label noise increases the variance of the loss.

**Theorem 2** (Variance increases with label noise). *Consider infinite true data distribution $\mathcal{D}$, and uniform label noise, where each label is flipped independently with probability $\rho$. Let $\mathcal{D}_\rho$ denote the noisy version of $\mathcal{D}$. Consider 0-1 loss $l$, and assume that there exists at least one function $\bar{f} \in \mathcal{F}$ such that $L_\mathcal{D}(\bar{f}) < \frac{1}{2} - \gamma$. For a fixed $f \in \mathcal{F}$, let $\sigma^2(f, \mathcal{D}_\rho)$ be the variance of the loss, $\sigma^2(f, \mathcal{D}_\rho) = Var_{z \sim \mathcal{D}_\rho} l(f, z)$ on data distribution $\mathcal{D}_\rho$. For any $0 < \rho_1 < \rho_2 < \frac{1}{2}$,*

$$\sigma^2(f, \mathcal{D}_{\rho_1}) < \sigma^2(f, \mathcal{D}_{\rho_2}).$$

The proof of Theorem 2 is in Appendix A. This covers the uniform noise case, but variance increases more generally, and we prove this for several other common cases in Appendix A. More specifically, we show that the variance increases with other types of label noise, such as non-uniform label noise (see Theorem 12 in Appendix A) and margin noise (see Theorem 15 in Appendix A). For non-uniform label noise, for a sample $z = (x, y)$, each label $y$ is flipped independently with probability $\rho_x$, meaning that noise can depend on $x$. This noise model is more realistic than uniform label noise and allows modeling of cases when one sub-population has much more noise than another. We model margin noise such as that which arises from high-dimensional Gaussians. Because of the central limit theorem, data often follow Gaussian distributions, therefore this noise is realistic and models mistakes near decision boundary. Label noise in datasets is common. In fact, real-world datasets reportedly have between 8.0% and 38.5% label noise [44, 43, 24, 28, 51]. We hypothesize that a significant amount of label noise in real-world datasets is a combination of Gaussian (due to the central limit theorem) and random noise (for example, because of clerical errors causing label noise).

For the true Rashomon set $R_{set}(\mathcal{F}, \gamma)$, we consider the maximum variance for all models in the true set: $\sigma^2 = \sup_{f \in R_{set}(\mathcal{F}, \gamma)} \text{Var}_{z \sim \mathcal{D}} l(f, z)$. Then, from Theorem 2 we have that maximum expected variance over the Rashomon set increases with noise.

**Corollary 3** (Maximum variance increases with label noise). *Under the same assumptions as in Theorem 2, we have that*

$$\sup_{f \in R_{set_{\mathcal{D}_{\rho_1}}}(\mathcal{F}, \gamma)} \sigma^2(f, \mathcal{D}_{\rho_1}) < \sup_{f \in R_{set_{\mathcal{D}_{\rho_2}}}(\mathcal{F}, \gamma)} \sigma^2(f, \mathcal{D}_{\rho_2}).$$

The next step is to show that this increased maximum variance leads to worse generalization.

## 4.2 Step 2. Higher Variance Leads to Worse Generalization

Here we use an argument based on generalization bounds. Generalization bounds have been the key theoretical motivation for much of machine learning, including support vector machines (SVMs), because the margin that SVMs optimize appears in a bound. While bounds themselves are not directly used in practice, the terms in the bounds tend to be important quantities in practice. Our bound cannot be calculated in practice because it uses population information on the right side, but it still provides insight and motivation.

Unlike standard bounds, we will use the fact that the user is using empirical risk minimization, and cross-validation to assess overfitting. Thus, for $\hat{f}$, there are two possibilities: $\hat{f}$ is in the true Rashomon set, or it is not. If it is not, then for the empirical risk minimizer $\hat{f}$, the difference between the true and the empirical risk must be at least $\gamma$, which will be detected with high probability in cross-validation [20, 33]. In that case, the user will reduce their hypothesis space and we move to Step 3. If $\hat{f}$ is in the true Rashomon set, it obeys the following bound.

**Theorem 4** (Variance-based "generalization bound"). *Consider dataset S, 0-1 loss l, and finite hypothesis space $\mathcal{F}$. With probability at least $1 - \delta$, we have that for every $f \in R_{set}(\mathcal{F}, \gamma)$:*

$$L(f) - \hat{L}(f) \leq \frac{2}{3n} \log \left( \frac{|R_{set}(\mathcal{F}, \gamma)|}{\delta} \right) + \sqrt{\frac{2\sigma^2}{n} \log \left( \frac{|R_{set}(\mathcal{F}, \gamma)|}{\delta} \right)}, \qquad (1)$$

*where $\sigma^2 = \sup_{f \in R_{set}(\mathcal{F}, \gamma)} \mathrm{Var}_{z \sim \mathcal{D}} \, l(f, z)$, and n is number of samples in $S = \{z_i\}_{i=1}^{n} \sim \mathcal{D}$.*

The proof of Theorem 4 is in Appendix C. Note that generalization bounds are usually based on Hoeffding's inequality (see Lemma 17), which is a special case of Bernstein's inequality (see Lemma 16). In fact, we show in Appendix B that Bernstein's inequality, which we used to prove Theorem 4, can be sharper than Hoeffding's when the variance is less than $\sigma_f^2 < \frac{1}{12}$ for a given $f$. Theorem 4 is easily generalized to continuous hypothesis spaces through a covering argument over the true Rashomon set (as an example, see Theorem 20), where the complexity is measured as the size of the cover over the true Rashomon set instead of the number of models in the true Rashomon set.

Let $c(\mathcal{F}, n) = \frac{2}{3n} \log \left( \frac{|R_{set}(\mathcal{F}, \gamma)|}{\delta} \right)$, which is the first term in the bound (1) in Theorem 4. According to Theorem 8 in Semenova et al. [39] under random label noise, the true Rashomon set does not decrease in size. Therefore, $c(\mathcal{F}, n)$ at least does not decrease with more noise as it depends only on complexity. However, the second term $\sqrt{3\sigma^2 c(\mathcal{F}, n)}$ depends on the maximum loss variance, which increases with label noise, as motivated in the previous section. This means with more noise in the labels, we would expect worse generalization, which would generally lead practitioners who are using a validation set to reduce the complexity of the hypothesis space.

As discussed earlier, we use cross-validation to assess whether $\hat{f}$ overfits. From Proposition 5 (proved in Appendix D) we infer that if the empirical risk minimizer (ERM) does not highly overfit, it has a high chance to be in the true Rashomon set and thus Theorem 4 applies.

**Proposition 5** (ERM can be close to the true Rashomon set). *Assume that through the cross-validation process, we can assess $\xi$ such that $L(\hat{f}) - \hat{L}(\hat{f}) \leq \xi$ with high probability (at least $1 - \epsilon_\xi$) with respect to the random draw of data. Then, for any $\epsilon > 0$, with probability at least $1 - e^{-2n\epsilon^2} - \epsilon_\xi$ with respect to the random draw of training data, when $\xi + \epsilon \leq \gamma$, then $\hat{f} \in R_{set}(\mathcal{F}, \gamma)$.*

## 4.3 Step 3. Practitioner Chooses a Simpler Hypothesis Space

We have shown earlier that noisier datasets lead to higher variance and worse generalization. The question we consider here is whether one can see the results of these bounds in practice and would actually reduce the hypothesis space. For example, considered four real-world datasets and the hypothesis space of decision trees of various depths. In Figure 1 (a) we show that as label noise increases, so does the gap between risks ($1-$ accuracy) evaluated on the training set and on a hold out dataset for a fixed depth tree, and thus, as a result, during the validation process, the smaller depth would be chosen by a reasonable analyst. We simulated this by using cross validation to select the optimal tree depth in CART, as shown in Figure 1 (b). As predicted, the optimal tree depth decreases as noise increases. We also show that a similar trend happens for the gradient boosted trees in Figure

7 in Appendix K.2. More specifically, with more noise, the best number of estimators, as chosen based on cross-validation, decreases. We describe the setup in detail in Appendix K.2.

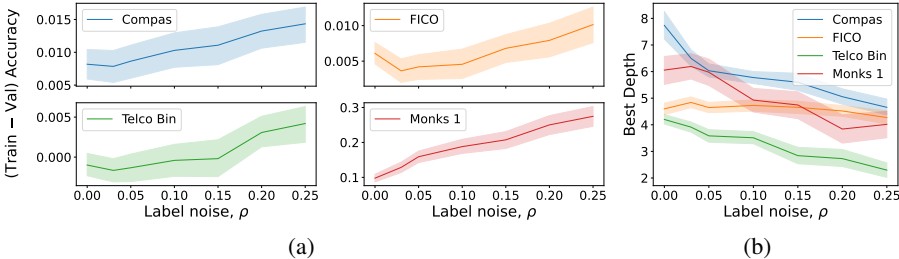

(a)                                                                 (b)

Figure 1: Practitioner's validation process in the presence of noise for CART. For a fixed tree depth, as we add noise, the gap between training and validation accuracy increases (Subfigure a). As we use cross validation to select tree depth, the best tree depth decreases with noise (Subfigure b).

### 4.4   Step 4. Rashomon Ratio is Larger for Simpler Spaces

For simpler hypothesis spaces, it may not be immediately obvious that the Rashomon ratio is larger, i.e., a larger fraction of a simpler model class consists of "good" models. Intuitively, this is because the denominator of the ratio (the total number of models) increases faster than the numerator (the number of good models) as the complexity of the model class increases. As we will see, this is because the good models in the simpler model class tend to give rise to more bad models in the more complex class, and the bad models do not tend to give rise to good models as much. We explore two popular model classes: decision trees and linear models. The results thus extend to forests (collections of trees) and generalized additive models (which are linear models in enhanced feature space).

Consider data that live on a hypercube (e.g., the data have been binarized, which is a common pre-processing step [29, 3, 52, 47]) and a hypothesis space of fully grown trees (complete trees with the last level also filled) of a given depth $d$. Denote this hypothesis space as $\mathcal{F}_d$. For example, a depth 1 tree has 1 node and 2 leaves. Under natural assumptions on the quality of features of classifiers and the purity of the leaves of trees in the Rashomon set, we show that the Rashomon ratio is larger for hypothesis spaces of smaller-depth trees.

**Proposition 6** (Rashomon ratio is larger for decision trees of smaller depth)**.** *For a dataset $S = X \times Y$ with binary feature matrix $X \in \{0,1\}^{n \times m}$, consider a hypothesis space $\mathcal{F}_d$ of fully grown trees of depth $d$. Let the number of dimensions $m < 2^{2^d}$. Assume: (Leaves are correct) all leaves in all trees in the Rashomon set have at least $\lceil \theta n \rceil$ more correctly classified points than incorrectly classified points; (Bad features) there is a set of $m_{bad} \geq d$ "bad" features such that the empirical risk minimizer of models using only the bad features is not in the Rashomon set. Then $\hat{R}_{ratio}(\mathcal{F}_{d+1}, \theta) < \hat{R}_{ratio}(\mathcal{F}_d, \theta)$.*

The proof of Proposition 6 is in Appendix E. Both assumptions are typically satisfied in practice.

To demonstrate our point, we computed the Rashomon ratio and pattern Rashomon ratio for 19 different datasets for hypothesis spaces of decision trees and linear models of different complexity (see Figure 2). As the complexity of the hypothesis space increases, we see an obvious decrease in the Rashomon ratio and pattern Rashomon ratio.

To compute the Rashomon ratio, for trees, we used TreeFARMS [52], which allows us to enumerate the whole Rashomon set for sparse trees. For linear models, we designed a two-step approach that allows us to compute all patterns in the Rashomon set. First, for every sample $z_i$ we solved a simple optimization problem that checks if there exists a model in the Rashomon set that misclassifies this sample. If there is no such model, changing the label of $z_i$ will not add more patterns to the Rashomon set, therefore, we can ignore $z_i$. In the second step, we grew a search tree and bounded all paths that lead to patterns outside of the Rashomon set. We describe our approach for pattern computation in detail in Appendix K.3 and discuss the experimental setup in Appendix K.4.

**Completion of the Path**. After reducing the complexity of the hypothesis space in Step 3, the practitioner has already arrived at a simpler hypothesis space, which is the goal. Continuing on the

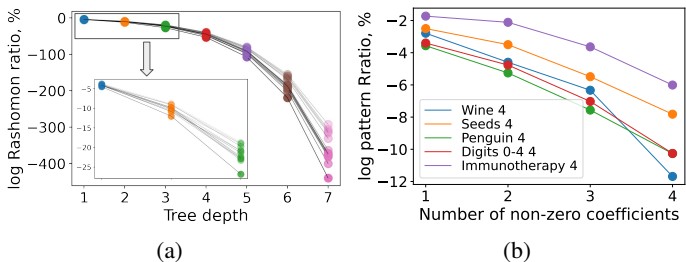

(a)                      (b)

Figure 2: Calculation showing that the Rashomon ratio (a) and pattern Rashomon ratio (b) decrease for the hypothesis space of decision trees of fixed depth from 1 to 7 for 14 different datasets (a) and for the hypothesis space of linear models of sparsity from 1 to 4 for 5 different datasets (b). Each line represents a different dataset, each dot represents the log of the Rashomon ratio or pattern Rashomon ratio. Both ratios decrease as we move to a more complex hypothesis space.

path, they can reduce complexity further. Specifically, they find a larger Rashomon ratio for the newly chosen lower complexity hypothesis space according to Step 4. As a reminder of the thesis of Semenova et al. [39], with large Rashomon ratios, there are many good models, among which may exist even simpler models that perform well. Thus, the path, starting from noise, is a powerful way to explain what we see in practice, which is that simple models often perform well [17].

Note that to follow the path, the machine learning practitioner does not need to know the exact amount of noise. As long as they suspect *some* noise present in the dataset, the results of this paper apply, and the practitioner would expect a good performance from simpler models.

## 5 Rashomon Ratio for Ridge Regression Increases under Additive Attribute Noise

For linear regression, adding multiplicative or additive noise to the training data is known to be equivalent to regularizing the model parameters [5]. Moreover, the more noise is added, the stronger this regularization is. Thus, noise leads directly to Step 3 and a choice of a smaller (simpler) hypothesis space. Also, for ridge regression, the Rashomon volume (the numerator of the Rashomon ratio) can be computed directly [39] and depends on the regularization parameter. Building upon these results, we prove that noise leads to an increase of the Rashomon ratio (as in Step 4 of the path).

Given dataset $S$, the ridge regression model is learned by minimizing the penalized sum of squared errors: $\hat{L}(\omega) = \hat{L}_{LS}(\omega) + C\omega^T\omega$, where $\hat{L}_{LS}(\omega) = \frac{1}{n}\sum_{i=1}^{n}\left(x_i^T\omega - y_i\right)^2$ is the least squares loss, $C$ is a regularization parameter, and $\omega \in \mathbb{R}^m$ is a parameter vector for a linear model $f = \omega^T x$.

We will assume that there is a maximum loss value $\hat{L}_{\max}$, such that any linear model that has higher regularized loss than $\hat{L}_{\max}$ is not being considered within the hypothesis space. For instance, an upper bound for $\hat{L}_{\max}$ is the value of the loss at the model that is identically $\bar{0}$, namely $\hat{L}(\bar{0}) = \frac{1}{n}\sum_i y_i^2$. Thus, for every reasonable model $f = \omega^T x$, $f \in \mathcal{F} \equiv \hat{L}(\omega) \leq \hat{L}_{\max}$. On the other hand, the best possible value of the least squares loss is $\hat{L}_{LS} = 0$, and therefore we get that $Cw^Tw \leq \hat{L}_{\max}$, or alternatively, $w^Tw \leq \hat{L}_{\max}/C$. This defines the hypothesis space as an $\ell_2$-norm ball in $m$-dimensional space, the volume of which we can compute.

To measure the numerator of the Rashomon ratio we will use the Rashomon volume $\mathcal{V}(\hat{R}_{set}(\mathcal{F}, \theta))$, as defined in Semenova et al. [39]. In the case of ridge regression, the Rashomon set is an ellipsoid in $m$-dimensions, thus the Rashomon volume can be computed directly. Therefore, we have the Rashomon ratio as the ratio of the Rashomon volume to the volume of the $\ell_2$-norm ball that defines the hypothesis space.

Next, we show that under additive attribute noise, the Rashomon ratio increases:

**Theorem 7** (Rashomon ratio increases with noise for ridge regression). *Consider dataset $S = X \times Y$, $X$ is a non-zero matrix, and a hypothesis space of linear models $\mathcal{F} = \{f = \omega^T x, \omega \in \mathbb{R}^m, \omega^T\omega \leq$*

$\hat{L}_{\max}/C\}$. *Let $\epsilon_i$, such that $\epsilon_i \sim \mathcal{N}(\bar{0}, \lambda I)$ ($\lambda > 0$, $I$ is identity matrix), be i.i.d. noise vectors added to every sample: $x'_i = x_i + \epsilon_i$. Consider options $\lambda_1 > 0$ and $\lambda_2 > 0$ that control how much noise we add to the dataset. For ridge regression, if $\lambda_1 < \lambda_2$, then the Rashomon ratios obey $\hat{R}_{ratio_{\lambda_1}}(\mathcal{F}, \theta)) < \hat{R}_{ratio_{\lambda_2}}(\mathcal{F}, \theta))$.*

The proof of Theorem 7 is in Appendix F. Note that while in Theorem 7 we directly show that adding noise to the training data is equivalent to stronger regularization leading us directly to Step 3, Steps 1 and 2 of the path identified in the previous section are automatically satisfied. We formally prove that additive noise still leads to an increase of the variance of losses for least squares loss (similar to Theorems 2, 12, and 15) in Theorem 19 in Appendix F, and we show that an increase in the maximum variance of losses leads to worse generalization bound (similar to Theorem 4) for the squared loss in Theorem 20 in Appendix F.

**Returning to the Path.** For ridge regression, we have now built *a direct noise-to-Rashomon-ratio argument* showing that, in the presence of noise, the Rashomon ratios are larger. As before, for larger Rashomon ratios, there are multiple good models, including simpler ones that are easier to find.

# 6 Rashomon Set Characteristics in the Presence of Noise

Now we discuss a different mechanism for obtaining larger Rashomon sets. Suppose the practitioner knows the data are noisy. They would then expect a large Rashomon set, which we speculate in this section is explained by noise in the data. To show this, we define *pattern diversity* as a characteristic of the Rashomon set and show that it is likely to increase with label noise. Pattern diversity is an empirical measure of differences in patterns on the dataset. It computes the average distance between the patterns, which allows us not only to assess how large the Rashomon set is but also how diverse it is. Once the practitioner knows they have a large Rashomon set, they could hypothesize from the reasoning in Semenova et al. [39] that a simple model might perform well for their dataset.

## 6.1 Pattern Diversity: Definition, Properties and Upper Bound

Recall that $\pi(\mathcal{F}, \theta)$ is the set of unique classification patterns produced by the Rashomon set of $\mathcal{F}$ with the Rashomon parameter $\theta$.

**Definition 8** (Pattern diversity). *For Rashomon set $\hat{R}_{set}(\mathcal{F}, \theta)$, the pattern diversity $div(\hat{R}_{set}(\mathcal{F}, \theta))$ is defined as:*

$$div(\hat{R}_{set}(\mathcal{F}, \theta)) = \frac{1}{n \, \Pi \, \Pi} \sum_{\substack{j \leq \Pi \\ p_j \sim \pi(\mathcal{F}, \theta)}} \sum_{\substack{k \leq \Pi \\ p_k \sim \pi(\mathcal{F}, \theta)}} H(p_j, p_k),$$

*where $n = |S|$, $\Pi = |\pi(\mathcal{F}, \theta)|$, and $H(p_j, p_k) = \sum_{i=1}^{n} \mathbb{1}_{[p_j^i \neq p_k^i]}$ is the Hamming distance (in our case it computes the number of samples at which predictions are different), and $|\cdot|$ denotes cardinality.*

Pattern diversity measures pairwise differences between patterns of functions in the pattern Rashomon set. Pattern diversity is in the range $[0, 1)$, where it is $0$ if the pattern set contains one pattern or no patterns. Among different measures of the Rashomon set, the pattern diversity is the closest to the pattern Rashomon ratio [38] and expected pairwise disagreement (as in [6]). In Appendix G, we discuss similarities and differences between pattern diversity and these measures.

Given a sample $z_i$, let $a_i = \frac{1}{\Pi} \sum_{k=1}^{\Pi} \mathbb{1}_{[p_k^i = y_i]}$, where $p_k^i$ is $i^{\text{th}}$ index of the $k^{\text{th}}$ pattern, denote the probability with which patterns from the pattern Rashomon set classify $z_i$ correctly. We will call $a_i$ *sample agreement* over the pattern Rashomon set. When $a_i = 1$, then all patterns agreed and correctly classified $z_i$. If $a_i = \frac{1}{2}$, only half of the models were able to correctly predict the label. As we will show, when more samples have sample agreement near $\frac{1}{2}$, we have higher pattern diversity. We can compute pattern diversity using average sample agreements instead of the Hamming distance according to the theorem below.

**Theorem 9** (Pattern diversity via sample agreement). *For 0-1 loss, dataset $S$, and pattern Rashomon set $\pi(\mathcal{F}, \theta)$, pattern diversity can be computed as $div(\hat{R}_{set}(\mathcal{F}, \theta)) = \frac{2}{n} \sum_{i=1}^{n} a_i(1 - a_i)$, where $a_i = \frac{1}{\Pi} \sum_{k=1}^{\Pi} \mathbb{1}_{[p_k^i = y_i]}$ is sample agreement over the pattern Rashomon set.*

The proof of Theorem 9 is in Appendix H. In Theorem 23 in Appendix I, we show that average sample agreement (over all samples $z_i$) is inversely proportional to the average loss of the patterns from the pattern Rashomon set. Using this intuition, we can upper bound the pattern diversity by the empirical risk of the empirical risk minimizer and the Rashomon parameter $\theta$.

**Theorem 10** (Upper bound on pattern diversity). *Consider hypothesis space $\mathcal{F}$, 0-1 loss, and empirical risk minimizer $\hat{f}$. For any $\theta \geq 0$, pattern diversity can be upper bounded by*

$$div(\hat{R}_{set}(\mathcal{F}, \theta)) \leq 2(\hat{L}(\hat{f}) + \theta)(1 - (\hat{L}(\hat{f}) + \theta)) + 2\theta. \tag{2}$$

The proof of Theorem 10 is in Appendix I. The bound (2) emphasizes how important the performance of the empirical risk minimizer is for understanding pattern diversity. If dataset is well separated so that the empirical risk is small, then pattern diversity will also be small, as there are not many different ways to misclassify points and stay within the Rashomon set. As dataset becomes noisier, on average, we expect the empirical risk to increase and thus pattern diversity as well. We will show theoretically and experimentally that pattern diversity is likely to increase under label noise.

## 6.2 Label Noise is Likely to Increase Pattern Diversity

Let $S_\rho$ be a version of $S$ with uniformly random label noise, creating randomly perturbed labels $\tilde{y}$ with probability $0 < \rho < \frac{1}{2}$: $P(\tilde{y}_i \neq y_i) = \rho$. Let $\Omega(S_\rho)$ be the uniform distribution over all $S_\rho$. From Theorem 10 denote the upper bound on pattern diversity as $U_{div}(\hat{R}_{set}(\mathcal{F}, \theta)) = 2(\hat{L}(\hat{f}) + \theta)(1 - (\hat{L}(\hat{f}) + \theta)) + 2\theta$. We show that it increases with uniform label noise.

**Theorem 11** (Upper bound on pattern diversity increases with label noise). *Consider a hypothesis space $\mathcal{F}$, 0-1 loss, and a dataset $S$. Let $\rho \in \left(0, \frac{1}{2}\right)$ be the probability with which each label $y_i$ is flipped independently, and let $S_\rho \sim \Omega(S_\rho)$ denote a noisy version of $S$. For the Rashomon parameter $\theta \geq 0$, if $\inf_{f \in \mathcal{F}} \mathbb{E}_{S_\rho} \hat{L}_{S_\rho}(f) < \frac{1}{2} - \theta$ and $\hat{L}_S(\hat{f}_S) < \frac{1}{2}$, then adding noise to the dataset increases the upper bound on pattern diversity of the expected Rashomon set:*

$$U_{div}(\hat{R}_{set_S}(\mathcal{F}, \theta)) < U_{div}(\hat{R}_{set_{\mathbb{E}_{S_\rho \sim \Omega(S_\rho)} S_\rho}}(\mathcal{F}, \theta)).$$

The proof of Theorem 11 is in Appendix J. In the general case, it is challenging to find a closed-form formula for pattern diversity or design a lower bound without strong assumptions about the data distribution or the hypothesis space. Therefore, we empirically examine the behavior of pattern diversity alongside other characteristics of the Rashomon set for different datasets and show these characteristics tend to increase with more noise.

## 6.3 Experiment for Pattern Diversity and Label Noise

We expect many different datasets to have larger Rashomon set measurements as data become more noisy. As before, we consider uniform label noise, where each label is flipped independently with probability $\rho$. For different noise levels, we computed diversity, and the number of patterns in the Rashomon set for 12 different datasets for the hypothesis space of sparse decision trees of depth 3 (note that we underfitted for some of the datasets); see Figure 3 (a)-(b). For every dataset, we introduced up to 25% label noise ($\rho \in [0, 0.25]$), or Accuracy($\hat{f}$) $- 50\%$, whichever is smaller. This means that data with lower accuracy (before adding noise) will have shorter plots, since noise is along the horizontal axis of our plots in Figure 3. For each noise level $\rho$, we performed 50 draws of $S_\rho$, and for each draw $S_\rho$ we recomputed the Rashomon set. For decision trees we used TreeFARMS [52], which allows us to compute the number of trees in the Rashomon set. We set the Rashomon parameter to 5%. We discuss results for the hypothesis space of linear models in Appendix K.5.

Some key observations from Figure 3: First, the number of trees and patterns in the Rashomon set on average increases with noise. This means that the Rashomon ratio and pattern Rashomon ratio increase with noise as well since the hypothesis space stays the same. Second, datasets, that initially had higher empirical risk (e.g., COMPAS, Coffee House, FICO) tend to have more models in the Rashomon set (and thus higher Rashomon ratios) as compared to datasets with lower empirical risk (Car Evaluation, Monks1, Monks3). Finally, pattern diversity, on average, tends to increase with noise for the majority of the datasets.

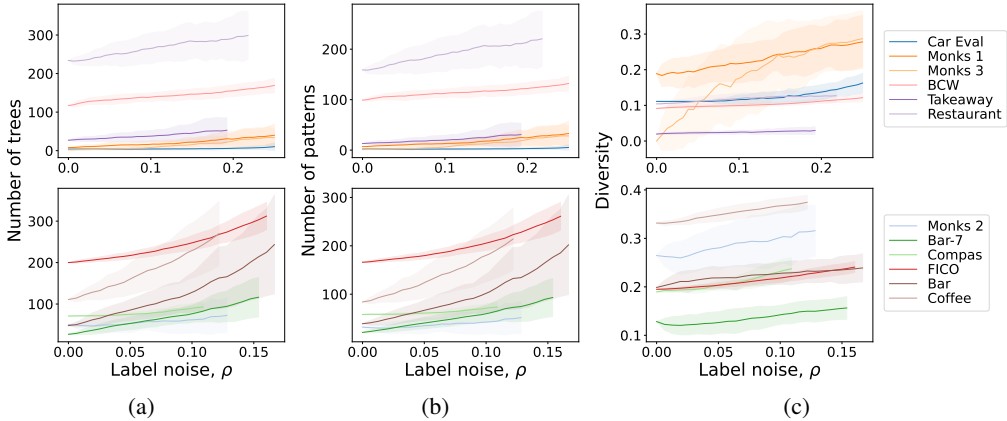

Figure 3: Rashomon set characteristics such as the number of trees in the Rashomon set (Subfigure a), the number of patterns in the Rashomon set (Subfigure b), and pattern diversity (Subfigure c) tend to increase with uniform label noise for hypothesis spaces of sparse decision trees. For readability, the top row of the figure shows datasets with lower empirical risk and the bottom row shows datasets with higher empirical risk.

**Returning to the Path**. If the practitioner observes more noise in the data, it could be already the case that the Rashomon set is large. Then, there are many good models, among which simpler or interpretable model are likely to exist [39].

# 7    Limitations

While we believe our results have illuminated that the Rashomon effect often exists when data are noisy, the connection between noise and increased Rashomon metrics is mostly supported by experiments, rather than a tight set of theoretical bounds. Specifically, we do not have a lower bound on diversity nor a correlation between diversity and the pattern Rashomon ratio (or Rashomon ratio) yet. This connection deserves further exploration and strengthening.

In previous sections, we showed that we expect an increase in pattern diversity while adding more label noise. In Section 6.3, we also observe an increase in the number of trees and patterns in the Rashomon set under label noise. In both cases, we used 0-1 loss. While in Section 5 we studied ridge regression (and thus used squared loss), ways to extend our results (both experimental and theoretical) to different distance-based classification loss functions are not yet fully clear. In the case of classification with distance-based losses, such as exponential loss, the effect of label noise can be difficult to model without taking properties of the data distribution into account. For example, regardless of the amount of noise in the dataset, a single misclassified outlier could essentially define the Rashomon set. The exponential loss could be very sensitive to the position of this outlier, leading to a small Rashomon set. This does not apply to the analysis in this paper, since the 0-1 loss we used is robust to outliers. Perhaps techniques that visualize the loss landscape [26] can be helpful in characterizing the Rashomon set for distance-based classification losses.

# Conclusion

Our results have profound policy implications, as they underscore the critical need to prioritize interpretable models in high-stakes decision-making. In a world where black box models are often used for high-stakes decisions, and yet the data generation processes are known to be noisy, our work sheds light on the false premise of this dangerous practice – that black box models are likely to be more accurate. Our findings have particular relevance for critical domains such as criminal justice, healthcare, and loan decisions, where individuals are subjected to the outputs of these models. The use of interpretable models in these areas can safeguard the rights and well-being of these individuals and ensure that decision-making processes are transparent, fair, and accountable.

## Acknowledgments

We gratefully acknowledge support from grants DOE DE-SC0023194, NIH/NIDA R01 DA054994, and NSF IIS-2130250.

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

# Supplemental materials
## A Path to Simpler Models Starts With Noise

## A  Proof for Theorem 2

We state and prove Theorem 2 below.

**Theorem 2** (Variance increases with label noise). *Consider infinite true data distribution $\mathcal{D}$, and uniform label noise, where each label is flipped independently with probability $\rho$. Let $\mathcal{D}_\rho$ denote the noisy version of $\mathcal{D}$. Consider 0-1 loss $l$, and assume that there exists at least one function $\bar{f} \in \mathcal{F}$ such that $L_\mathcal{D}(\bar{f}) < \frac{1}{2} - \gamma$. For a fixed $f \in \mathcal{F}$, let $\sigma^2(f, \mathcal{D}_\rho)$ be the variance of the loss, $\sigma^2(f, \mathcal{D}_\rho) = Var_{z \sim \mathcal{D}_\rho} l(f, z)$ on data distribution $\mathcal{D}_\rho$. For any $0 < \rho_1 < \rho_2 < \frac{1}{2}$,*

$$\sigma^2(f, \mathcal{D}_{\rho_1}) < \sigma^2(f, \mathcal{D}_{\rho_2}).$$

*Proof.* Recall that the true risk for 0-1 loss $L_\mathcal{D}(f) = \mathbb{E}_{z=(x,y) \sim \mathcal{D}}[l(f,z)] = \mathbb{E}_{(x,y) \sim \mathcal{D}}[\mathbb{1}_{[f(x) \neq y]}]$. Without loss of generality, let $y \in \{0, 1\}$. Drawing from $z_\rho \sim \mathcal{D}_\rho$ is equivalent to drawing $z \sim \mathcal{D}$ and changing label $y$ to $1 - y$ with probability $\rho$. More explicitly, let $\eta \sim$ Bernoulli$(\rho)$, then the flipped label is $XOR(y, \eta) = \mathbb{1}_{[y \neq \eta]}$. For any given $f \in \mathcal{F}$ we have that:

$$
\begin{aligned}
L_{\mathcal{D}_\rho}(f) &= \mathbb{E}_{\eta \sim Ber(\rho)} \mathbb{E}_{(x,y) \sim \mathcal{D}} \left[ \mathbb{1}_{[f(x) \neq XOR(y, \eta)]} \right] \\
&= \mathbb{E}_{\eta \sim Ber(\rho)} \mathbb{E}_{(x,y) \sim \mathcal{D}} \left[ \mathbb{1}_{[f(x) \neq y]}(1 - \eta) \right] + \mathbb{E}_{\eta \sim Ber(\rho)} \mathbb{E}_{(x,y) \sim \mathcal{D}} \left[ \mathbb{1}_{[f(x) = y]} \eta \right] \\
&= \mathbb{E}_{(x,y) \sim \mathcal{D}} \mathbb{E}_{\eta \sim Ber(\rho)} \left[ \mathbb{1}_{[f(x) \neq y]}(1 - \eta) \right] + \mathbb{E}_{(x,y) \sim \mathcal{D}} \mathbb{E}_{\eta \sim Ber(\rho)} \left[ \mathbb{1}_{[f(x) = y]} \eta \right] \\
&= \mathbb{E}_{(x,y) \sim \mathcal{D}} \left[ \mathbb{1}_{[f(x) \neq y]} \mathbb{E}_{\eta \sim Ber(\rho)} [(1 - \eta)] \right] + \mathbb{E}_{(x,y) \sim \mathcal{D}} \left[ \mathbb{1}_{[f(x) = y]} \mathbb{E}_{\eta \sim Ber(\rho)} [\eta] \right] \\
&= (1 - \rho) \mathbb{E}_{(x,y) \sim \mathcal{D}} \left[ \mathbb{1}_{[f(x) \neq y]} \right] + \rho \mathbb{E}_{(x,y) \sim \mathcal{D}} \left[ \mathbb{1}_{[f(x) = y]} \right] \\
&= (1 - \rho) \mathbb{E}_{(x,y) \sim \mathcal{D}} \left[ \mathbb{1}_{[f(x) \neq y]} \right] + \rho \left( 1 - \mathbb{E}_{(x,y) \sim \mathcal{D}} \left[ \mathbb{1}_{[f(x) \neq y]} \right] \right) \\
&= (1 - \rho) L_\mathcal{D}(f) + \rho (1 - L_\mathcal{D}(f)) \\
&= (1 - 2\rho) L_\mathcal{D}(f) + \rho.
\end{aligned}
$$

Note, following the technique above, a similar statement is true about dataset $S$ instead of true distribution $\mathcal{D}$, meaning that for a given $f \in \mathcal{F}$,

$$\mathbb{E}_{S_\rho} \hat{L}_{S_\rho}(f) = (1 - 2\rho)\hat{L}_S(f) + \rho. \tag{3}$$

Recall that we take expectation with respect to different ways of adding noise to labels, therefore $S_\rho$ and $S$ have the same $x$, but different $y$. We do not use (3) for the proof of Theorem 2, but use it in Appendix J.

For true distribution $\mathcal{D}$, since $l$ is 0-1 loss, then for a given model $f$, $l(f, z)$ is Bernoulli distributed with mean $p_{Ber} = \mathbb{E}_{z \sim \mathcal{D}} l(f, z) = L_\mathcal{D}(f)$ and variance $\sigma_f^2 = p_{Ber}(1 - p_{Ber}) = L_\mathcal{D}(f)(1 - L_\mathcal{D}(f))$. Therefore, the expected variance for a given model $f \in R_{set}(\mathcal{F}, \gamma)$ on distribution $\mathcal{D}_\rho$ is:

$$
\begin{aligned}
Var_{z \sim \mathcal{D}_\rho}[l(f,z)] &= \mathbb{E}_{\mathcal{D}_\rho} L_{\mathcal{D}_\rho}(f)(1 - L_{\mathcal{D}_\rho}(f)) \\
&= \mathbb{E}_{\mathcal{D}_\rho} L_{\mathcal{D}_\rho}(f) - \mathbb{E}_{\mathcal{D}_\rho}(L_{\mathcal{D}_\rho}(f))^2 \\
&= \mathbb{E}_{\mathcal{D}_\rho} L_{\mathcal{D}_\rho}(f) - \mathbb{E}_{\mathcal{D}_\rho}(L_{\mathcal{D}_\rho}(f))^2 \\
&= \mathbb{E}_{\mathcal{D}_\rho} L_{\mathcal{D}_\rho}(f) - (\mathbb{E}_{\mathcal{D}_\rho} L_{\mathcal{D}_\rho}(f))^2 - Var_{\mathcal{D}_\rho}[L_{\mathcal{D}_\rho(f)}] \\
&= L_\mathcal{D}(f)(1 - 2\rho) + \rho - (L_\mathcal{D}(f)(1 - 2\rho) + \rho)^2 - Var_{\mathcal{D}_\rho}[L_{\mathcal{D}_\rho(f)}] \\
&= L_\mathcal{D}(f)((1 - 2\rho) - 2\rho(1 - 2\rho)) - L_\mathcal{D}^2(f)(1 - 2\rho)^2 + \rho - \rho^2 - Var_{\mathcal{D}_\rho}[L_{\mathcal{D}_\rho(f)}] \\
&= (1 - 2\rho)^2(L_\mathcal{D}(f) - L_\mathcal{D}^2(f)) + \rho - \rho^2 - Var_{\mathcal{D}_\rho}[L_{\mathcal{D}_\rho(f)}] \\
&= (1 - 2\rho)^2 (L_\mathcal{D}(f)(1 - L_\mathcal{D}(f))) + \rho(1 - \rho) - Var_{\mathcal{D}_\rho}[L_{\mathcal{D}_\rho(f)}] \\
&= (1 - 2\rho)^2 (L_\mathcal{D}(f)(1 - L_\mathcal{D}(f))) + \rho(1 - \rho),
\end{aligned}
$$

Note that, by our assumption, there exists $\bar{f}$ such that $L_{\mathcal{D}}(\bar{f}) < \frac{1}{2} - \gamma$, so $L_{\mathcal{D}}(f^*) < \frac{1}{2} - \gamma$, where $f^*$ is optimal model. Then for any fixed $f \in \mathcal{F}$, we get $L_{\mathcal{D}}(f) \leq L_{\mathcal{D}}(f^*) + \gamma < \frac{1}{2}$ which implies that $L_{\mathcal{D}}(f)(1 - L_{\mathcal{D}}(f)) < \frac{1}{4}$.

For $\rho \in (0, \frac{1}{2})$, $Var_{z \sim \mathcal{D}_\rho} [l(f, z)]$ is monotonically increasing in $\rho$, since:

$$
\begin{aligned}
\frac{\partial}{\partial \rho} \left[ Var_{z \sim \mathcal{D}_\rho} [l(f, z)] \right] &= \frac{\partial}{\partial \rho} \left[ (1 - 2\rho)^2 \left( L_{\mathcal{D}}(f)(1 - L_{\mathcal{D}}(f)) \right) + \rho(1 - \rho) \right] \\
&= -4(1 - 2\rho) \left( L_{\mathcal{D}}(f)(1 - L_{\mathcal{D}}(f)) \right) + (1 - 2\rho) \\
&= (1 - 2\rho) \left( 1 - 4L_{\mathcal{D}}(f)(1 - L_{\mathcal{D}}(f)) \right) \\
&> \left( 1 - 2 \times \frac{1}{2} \right) \left( 1 - 4 \times \frac{1}{4} \right) = 0.
\end{aligned}
$$

Consider $\rho_1 < \rho_2$. Since $Var_{z \sim \mathcal{D}_\rho} [l(f, z)]$ is monotonically increasing in $\rho$ for a fixed $f$, then $\sigma^2(f, \mathcal{D}_{\rho_1}) < \sigma^2(f, \mathcal{D}_{\rho_2})$, and we proved that variance increases with random uniform label noise.

$\square$

In Theorem 2, the statement of the theorem is correct for any fixed $f \in \mathcal{F}$. Corollary 3 follows directly from Theorem 2. Here, instead of a fixed model $f \in \mathcal{F}$, we consider models in the Rashomon sets that maximize expected variance.

**Corollary 3** (Maximum variance increases with label noise). *Under the same assumptions as in Theorem 2, we have that*

$$
\sup_{f \in R_{set_{\mathcal{D}_{\rho_1}}} (\mathcal{F}, \gamma)} \sigma^2(f, \mathcal{D}_{\rho_1}) < \sup_{f \in R_{set_{\mathcal{D}_{\rho_2}}} (\mathcal{F}, \gamma)} \sigma^2(f, \mathcal{D}_{\rho_2}).
$$

*Proof.* Let $f_1^{\text{sup}}$ and $f_2^{\text{sup}}$ be maximizers of the variance of the loss in their respective Rashomon sets:

$$
f_1^{\text{sup}} \in \arg \sup_{f \in R_{set_{\mathcal{D}_{\rho_1}}} (\mathcal{F}, \gamma)} Var_{z \sim \mathcal{D}_{\rho_1}} [l(f, z)],
$$

$$
f_2^{\text{sup}} \in \arg \sup_{f \in R_{set_{\mathcal{D}_{\rho_2}}} (\mathcal{F}, \gamma)} Var_{z \sim \mathcal{D}_{\rho_2}} [l(f, z)].
$$

Given that for any $f \in R_{set_{\mathcal{D}_{\rho_2}}} (\mathcal{F}, \gamma)$, $Var_{z \sim \mathcal{D}_{\rho_2}} [l(f, z)] \leq Var_{z \sim \mathcal{D}_{\rho_2}} [l(f_2^{\text{sup}}, z)]$ and since $Var_{z \sim \mathcal{D}_\rho} [l(f, z)]$ is monotonically increasing in $\rho$, we have that:

$$
\begin{aligned}
\sup_{f \in R_{set_{\mathcal{D}_{\rho_1}}} (\mathcal{F}, \gamma)} \sigma^2(f, \mathcal{D}_{\rho_1}) &= Var_{z \sim \mathcal{D}_{\rho_1}} [l(f_1^{\text{sup}}, z)] \\
&< Var_{z \sim \mathcal{D}_{\rho_2}} [l(f_1^{\text{sup}}, z)] \\
&\leq Var_{z \sim \mathcal{D}_{\rho_2}} [l(f_2^{\text{sup}}, z)] \\
&= \sup_{f \in R_{set_{\mathcal{D}_{\rho_2}}} (\mathcal{F}, \gamma)} \sigma^2(f, \mathcal{D}_{\rho_2}).
\end{aligned}
$$

$\square$

Next, we generalize the statement of Theorem 2 to the non-uniform label noise case, where each sample $z = (x, y)$ is flipped with probability $\rho_x$ that depends on $x$. We show that under this non-uniform label noise, the variance of the loss increases in Theorem 12.

**Theorem 12** (Variance increases with non-uniform label noise). *Consider 0-1 loss $l$, infinite true data distribution $\mathcal{D}$, and a hypothesis space $\mathcal{F}$. Assume that there exists at least one function $\bar{f} \in \mathcal{F}$ such that $L_{\mathcal{D}}(\bar{f}) < \frac{1}{2} - \gamma$. For a fixed $f \in \mathcal{F}$, let $\sigma^2(f, \mathcal{D})$ be the variance of the loss: $\sigma^2(f, \mathcal{D}) = Var_{z \sim \mathcal{D}} l(f, z)$ on data distribution $\mathcal{D}$. Consider non-uniform label noise, where each label $y$ is flipped independently with probability $\rho_x$, $(x, y) \sim \mathcal{D}$. Let $\mathcal{D}_\rho$ denote the noisy version of $\mathcal{D}$. For any $\delta > 0$, let $\mathcal{D}_{\rho^\delta}$ be a noisier data distribution than $\mathcal{D}_\rho$, meaning that for every sample*

$(x, y)$ the probabilities of labels being flipped are higher by $\delta$: $\rho_x^\delta = \rho_x + \delta$. If for a fixed model $f \in \mathcal{F}$, $L_{\mathcal{D}_{\rho^\delta}}(f) < 0.5$, then

$$\sigma^2(f, \mathcal{D}_\rho) < \sigma^2(f, \mathcal{D}_{\rho^\delta}).$$

*Proof.* Recall that the true risk for 0-1 loss $L_{\mathcal{D}}(f) = \mathbb{E}_{z=(x,y)\sim\mathcal{D}}[l(f,z)] = \mathbb{E}_{(x,y)\sim\mathcal{D}}[\mathbb{1}_{[f(x)\neq y]}]$. Without loss of generality, let $y \in \{0, 1\}$. Drawing from $z_\rho \sim \mathcal{D}_\rho$ is equivalent to drawing $z \sim \mathcal{D}$ and changing label $y$ to $1-y$ with probability $\rho_x$. More explicitly, let $\eta \sim$ Bernoulli$(\rho_x)$, then the flipped label is $XOR(y, \eta) = \mathbb{1}_{[y\neq\eta]}$. For any given $f \in \mathcal{F}$ we have that:

$$
\begin{aligned}
L_{\mathcal{D}_\rho}(f) &= \mathbb{E}_{(x,y)\sim\mathcal{D}} \, \mathbb{E}_{\eta\sim Ber(\rho_x)} \left[ \mathbb{1}_{[f(x)\neq XOR(y,\eta)]} \right] \\
&= \mathbb{E}_{(x,y)\sim\mathcal{D}} \, \mathbb{E}_{\eta\sim Ber(\rho_x)} \left[ \mathbb{1}_{[f(x)\neq y]}(1-\eta) \right] + \mathbb{E}_{(x,y)\sim\mathcal{D}} \, \mathbb{E}_{\eta\sim Ber(\rho_x)} \left[ \mathbb{1}_{[f(x)=y]}\eta \right] \\
&= \mathbb{E}_{(x,y)\sim\mathcal{D}} \left[ \mathbb{1}_{[f(x)\neq y]} \, \mathbb{E}_{\eta\sim Ber(\rho_x)} \left[ (1-\eta) \right] \right] + \mathbb{E}_{(x,y)\sim\mathcal{D}} \left[ \mathbb{1}_{[f(x)=y]} \, \mathbb{E}_{\eta\sim Ber(\rho_x)} \left[ \eta \right] \right] \\
&= \mathbb{E}_{(x,y)\sim\mathcal{D}} \left[ \mathbb{1}_{[f(x)\neq y]} \, \mathbb{E}_{\eta\sim Ber(\rho_x)} \left[ (1-\eta) \right] \right] + \mathbb{E}_{(x,y)\sim\mathcal{D}} \left[ \left( 1 - \mathbb{1}_{[f(x)\neq y]} \right) \mathbb{E}_{\eta\sim Ber(\rho_x)} \left[ \eta \right] \right] \\
&= \mathbb{E}_{(x,y)\sim\mathcal{D}} \left[ \mathbb{1}_{[f(x)\neq y]} \, \mathbb{E}_{\eta\sim Ber(\rho_x)} \left[ (1-\eta) \right] \right] + \mathbb{E}_{(x,y)\sim\mathcal{D}} \left[ \mathbb{E}_{\eta\sim Ber(\rho_x)} \left[ \eta \right] - \mathbb{1}_{[f(x)\neq y]} \, \mathbb{E}_{\eta\sim Ber(\rho_x)} \left[ \eta \right] \right] \\
&= \mathbb{E}_{(x,y)\sim\mathcal{D}} \left[ \mathbb{1}_{[f(x)\neq y]} \, \mathbb{E}_{\eta\sim Ber(\rho_x)} \left[ (1-2\eta) \right] \right] + \mathbb{E}_{(x,y)\sim\mathcal{D}} \left[ \mathbb{E}_{\eta\sim Ber(\rho_x)} \left[ \eta \right] \right] \\
&= \mathbb{E}_{(x,y)\sim\mathcal{D}} \left[ \mathbb{1}_{[f(x)\neq y]} \left( 1-2\rho_x \right) \right] + \mathbb{E}_{(x,y)\sim\mathcal{D}} \, \rho_x.
\end{aligned}
$$

Now we will show that $L_{\mathcal{D}_{\rho^\delta}}(f) > L_{\mathcal{D}_\rho}(f)$:

$$
\begin{aligned}
L_{\mathcal{D}_{\rho^\delta}}(f) - L_{\mathcal{D}_\rho}(f) &= \mathbb{E}_{(x,y)\sim\mathcal{D}} \left[ \mathbb{1}_{[f(x)\neq y]} \left( 1-2\rho_x^\delta \right) \right] + \mathbb{E}_{(x,y)\sim\mathcal{D}} \, \rho_x^\delta \\
&\quad - \mathbb{E}_{(x,y)\sim\mathcal{D}} \left[ \mathbb{1}_{[f(x)\neq y]} \left( 1-2\rho_x \right) \right] + \mathbb{E}_{(x,y)\sim\mathcal{D}} \, \rho_x \\
&= \mathbb{E}_{(x,y)\sim\mathcal{D}} \left[ \mathbb{1}_{[f(x)\neq y]} \left( -2\rho_x^\delta + 2\rho_x \right) \right] + \mathbb{E}_{(x,y)\sim\mathcal{D}} \left( \rho_x^\delta - \rho_x \right) \\
&= \mathbb{E}_{(x,y)\sim\mathcal{D}} \left[ \mathbb{1}_{[f(x)\neq y]} \left( -2\delta \right) \right] + \mathbb{E}_{(x,y)\sim\mathcal{D}} \left( \delta \right) \\
&= (-2\delta) \, \mathbb{E}_{(x,y)\sim\mathcal{D}} \left[ \mathbb{1}_{[f(x)\neq y]} \right] + \delta \\
&= \delta(1 - 2L_{\mathcal{D}}(f)) > 0.
\end{aligned}
$$

Note that, by our assumption, there exists $\bar{f}$ such that $L_{\mathcal{D}}(\bar{f}) < \frac{1}{2} - \gamma$, so $L_{\mathcal{D}}(f^*) < \frac{1}{2} - \gamma$, where $f^*$ is an optimal model. Then for any fixed $f \in R_{set}(\mathcal{F}, \gamma)$, we get $L_{\mathcal{D}}(f) \leq L_{\mathcal{D}}(f^*) + \gamma < \frac{1}{2}$, and then $1 - 2L_{\mathcal{D}}(f) > 0$. Since $\delta > 0$, we have shown that $L_{\mathcal{D}_{\rho^\delta}}(f) > L_{\mathcal{D}_\rho}(f)$.

For true distribution $\mathcal{D}$, since $l$ is 0-1 loss, then for a given model $f$, $l(f, z)$ is Bernoulli distributed with mean $p_{Ber} = \mathbb{E}_{z\sim\mathcal{D}} \, l(f, z) = L_{\mathcal{D}}(f)$ and variance $\sigma_f^2 = p_{Ber}(1-p_{Ber}) = L_{\mathcal{D}}(f)(1-L_{\mathcal{D}}(f))$. Therefore, the expected variance for a given model $f \in \mathcal{F}$ on distributions $\mathcal{D}_\rho$ and $\mathcal{D}_{\rho^\delta}$ is:

$$
\begin{aligned}
Var_{z\sim\mathcal{D}_\rho} \left[ l(f, z) \right] &= L_{\mathcal{D}_\rho}(f)(1 - L_{\mathcal{D}_\rho}(f)) \\
&< L_{\mathcal{D}_{\rho^\delta}}(f)(1 - L_{\mathcal{D}_{\rho^\delta}}(f)) \\
&= Var_{z\sim\mathcal{D}_{\rho^\delta}} \left[ l(f, z) \right],
\end{aligned}
$$

where the inequality arises from the fact that the parabola $x(1-x)$ is monotonic along the interval $x \in [0, 0.5]$. This implies that $\sigma^2(f, \mathcal{D}_\rho) < \sigma^2(f, \mathcal{D}_{\rho^\delta})$.

$\square$

We also show that the variance of losses increases under margin noise for data that come from Gaussian distributions (in Theorem 15). We model margin noise by moving two Gaussians closer together along the vector that connects the two means. Before stating and proving the theorem we discuss two lemmas that are helpful for the proof.

**Lemma 13.** *Consider distribution $\mathcal{X} \in \mathbb{R}^m$ and a linear model $f = \omega^T x + b$, where $\omega \in \mathbb{R}^m$, $\omega \neq \bar{0}$ and $b \in \mathbb{R}$. Let $x \mapsto Ax + c$ be a bijective affine transformation, where $A \in \mathbb{R}^{m\times m}$ and $c \in \mathbb{R}^m$. For the linear model $g(x) = f(A^{-1}(x - c))$ and the distribution $\mathcal{Z} = A\mathcal{X} + c$, we have that:*

$$P_{x\sim\mathcal{X}}(f(x) > 0) = P_{z\sim\mathcal{Z}}(g(z) > 0).$$

*Proof.* The proof follows from the lemma's statement and the assumption that $A$ is a bijective affine transformation, and thus is invertible:

$$P_{z \sim \mathcal{Z}}(g(z) > 0) = P_{x \sim \mathcal{X}}(g(Ax+c) > 0) = P_{x \sim \mathcal{X}}(f(A^{-1}(Ax+c-c)) > 0) = P_{x \sim \mathcal{X}}(f(x) > 0).$$

$\square$

**Lemma 14.** *Consider a Gaussian distribution $\mathcal{X} \sim \mathcal{N}(\mu, I)$, where $\mu \in \mathbb{R}^m$, and a linear model $f = \omega^T x + b$, where $\omega \in \mathbb{R}^m$, $\omega \neq \bar{0}$ and $b \in \mathbb{R}$. Let $r = \frac{\omega^T \mu + b}{\|\omega\|}$ be the signed distance from $\mu$ to the decision boundary of $f$. Then,*

$$P_{x \sim \mathcal{X}}(f(x) > 0) = \Phi(r),$$

*where $\Phi$ is the CDF of the univariate normal distribution $\mathcal{N}(0, 1)$.*

*Proof.* Let $O \in \mathbb{R}^{m \times m}$ be a matrix with the first row equal to $\frac{\omega}{\|\omega\|}$, and let the other rows be chosen so that the rows of $O$ form an orthonormal basis of $\mathbb{R}^m$. Note that $O$ is an orthogonal matrix, so $O$ is bijective and $O^T O = O O^T = I$. Let $g(t) = f(O^{-1}(t + O\mu))$ and $e_1$ be a unit vector $e_1 = \{1, 0, ..., 0\}$, then:

$$\begin{aligned}
g(t) = f(O^{-1}(t + O\mu)) &= f(O^{-1}t + \mu) = f(O^T t + \mu) \\
&= \omega^T O^T t + \omega^T \mu + b = \|\omega\| (e_1^T t) + \omega^T \mu + b \\
&= \|\omega\| t_1 + \omega^T \mu + b,
\end{aligned}$$

where $t_1$ is the first element of $t$, and $\omega^T O^T = \|\omega\| e_1^T$ comes from the fact that $\omega$ is orthogonal to every row of $O$ except for the first row. Note that $g(t) > 0$ when $\|\omega\| t_1 + \omega^T \mu + b > 0$, which leads to $t_1 > -\frac{\omega^T \mu + b}{\|\omega\|} = -r$. Correspondingly, $g(t) < 0$ when $t_1 < -r$.

Now, let $\mathcal{Z} = O(\mathcal{X} - \mu)$. From the properties of the normal distribution, $\mathcal{Z} \sim \mathcal{N}(\bar{0}, I)$ since:

$$\mathcal{Z} = O(\mathcal{X} - \mu) \sim \mathcal{N}(O(\mu - \mu), O I O^T) = \mathcal{N}(\bar{0}, I).$$

Moreover, since the standard multivariate normal distribution is the joint distribution of independent univariate normal distributions, $z_1 \sim \mathcal{N}(0, 1)$.

From Lemma 13 and definitions of $O$, $g$, $\mathcal{Z}$, we get that $P_{x \sim \mathcal{X}}(f(x) > 0) = P_{z \sim \mathcal{Z}}(g(z) > 0)$. Therefore:

$$\begin{aligned}
P_{x \sim \mathcal{X}}(f(x) > 0) = P_{z \sim \mathcal{Z}}(g(z) > 0) &= P_{z \sim \mathcal{Z}}(z_1 > -r) \\
&= P_{z_1 \sim \mathcal{N}(0,1)}(z_1 > -r) = P_{z_1 \sim \mathcal{N}(0,1)}(z_1 \leq r) \\
&= \Phi(r),
\end{aligned}$$

where the strict inequality becomes non-strict since for the Gaussian distribution, the probability $P_{z_1 \sim \mathcal{N}(0,1)}(z_i = r) = 0$. Thus, $P_{x \sim \mathcal{X}}(f(x) > 0) = \Phi(r)$ as desired. $\square$

Now, we show that the variance of losses increases under margin noise in the theorem below:

**Theorem 15.** *Consider data distribution $\mathcal{D} = \mathcal{X} \times \mathcal{Y}$, where, $\mathcal{X} \in \mathbb{R}^m$, $\mathcal{Y} \in \{-1, 1\}$, classes are balanced $P(Y = -1) = P(Y = 1)$ and generated by Gaussian distributions $P(X|Y = -1) = \mathcal{N}(\bar{0}, \Sigma)$, $P(X|Y = 1) = \mathcal{N}(\mu, \Sigma)$, where $\Sigma$ is a diagonal matrix with non-zero elements. Let the hypothesis space $\mathcal{F}$ be the set of linear models, $f = \omega^T x + b$, where $\omega \in \mathbb{R}^m$. $\omega \neq 0$ and $b \in \mathbb{R}$. We add margin noise by moving the means of the Gaussians towards each other by a factor of $k$, where $0 < k < 1$, meaning that the mean of the positive class becomes $\mu_k = k \cdot \mu$. For a fixed $f \in \mathcal{F}$, if $L_\mu(f) < 0.5$, we get that the variance of losses increases with more noise,*

$$\sigma(f, \mu) < \sigma(f, \mu_k).$$

*Proof.* Without loss of generality, we will show that the variance of the losses increases for data generated from two Gaussian distributions $P(X|Y = -1) = \mathcal{N}(\bar{0}, I)$ and $P(X|Y = 1) = \mathcal{N}(\mu, I)$ (where $I$ is the identity matrix) when we move them towards each other. More specifically, since normalization by variance $\left(\frac{1}{\Sigma_{i,i}}\right)$ is a bijective linear transformation, by Lemma 13 we can work

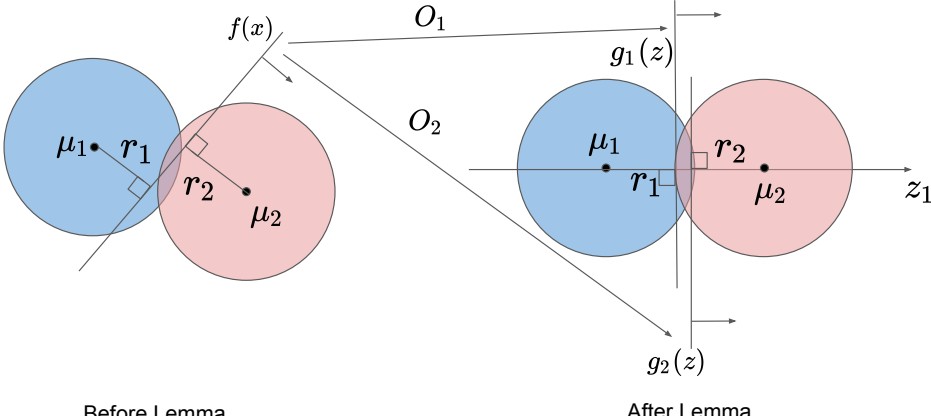

Before Lemma                    After Lemma

Figure 4: An illustration of how Lemma 14 rotates each of the Gaussians $\mathcal{N}(\mu_1, I), \mathcal{N}(\mu_2, I)$ and the decision boundary $f(x)$ in order to compute loss as CDF of the signed distances $(r_1, r_2)$ from means $(\mu_1, \mu_2)$ to the rotated boundaries $(g_1(z), g_2(z))$. Note that we apply Lemma 14 separately to each Gaussian, thus there are two rotation operators $O_1$, and $O_2$.

with $P(X|Y = -1) = \mathcal{N}(\bar{0}, I)$ and $P(X|Y = 1) = \mathcal{N}(\mu, I)$ instead of $P(X|Y = -1) = \mathcal{N}(\bar{0}, \Sigma)$ and $P(X|Y = 1) = \mathcal{N}(\mu, \Sigma)$.

Let $r_1 = \frac{b}{\|\omega\|}$ and $r_2 = \frac{\omega^T \mu + b}{\|\omega\|}$ be the signed distances from the centers of the two Gaussians to the decision boundary. Then, from Lemma 14 (see illustration in Figure 4), the loss can be computed using the CDFs based on the signed distance:

$$L_\mu(f) = P(f(x) > 0|Y = -1)P(Y = -1) + P(f(x) \leq 0|Y = 1)P(Y = 1)$$
$$= \frac{1}{2}P(f(x) > 0|Y = -1) + \frac{1}{2}(1 - P(f(x) > 0|Y = 1))$$
$$= \frac{1}{2}[(\Phi(r_1) + (1 - \Phi(r_2))].$$

Next, we will show that $\omega^T \mu > 0$. If $L_\mu(f) < \frac{1}{2}$, then we get that $\frac{1}{2}[(\Phi(r_1) + (1 - \Phi(r_2))] < \frac{1}{2}$, which means that $\Phi(r_2) > \Phi(r_1)$. Since the CDF of the Gaussian distribution $\mathcal{N}(\bar{0}, I)$ is strictly increasing, we have that $r_2 > r_1$, which means that $\frac{\omega^T \mu + b}{\|\omega\|} > \frac{b}{\|\omega\|}$, and so $\omega^T \mu > 0$.

Recall that we induce noise by moving the Gaussians towards each other by decreasing $k$. Now we will show that loss is monotonically decreasing with respect to increasing values of $k$, or equivalently that $\frac{\partial}{\partial k} L_{\mu_k}(f) < 0$:

$$\frac{\partial}{\partial k} L_{\mu_k}(f) = \frac{\partial}{\partial k} \left( \frac{1}{2}[(\Phi(r_1) + (1 - \Phi(r_2))] \right)$$
$$= \frac{\partial}{\partial k} \left( \frac{1}{2} \left[ (\Phi \left( \frac{b}{\|\omega\|} \right) + 1 - \Phi \left( \frac{k\omega^T \mu + b}{\|\omega\|} \right) \right] \right)$$
$$= -\frac{1}{2} \left[ \frac{\partial}{\partial k} \Phi \left( \frac{k\omega^T \mu + b}{\|\omega\|} \right) \right] = -\frac{1}{2} \frac{\omega^T \mu}{\|\omega\|} \phi \left( \frac{k\omega^T \mu + b}{\|\omega\|} \right) < 0,$$

since as we showed above, $\omega^T \mu > 0$, and $\phi$ is the PDF of normal distribution $\mathcal{N}(\bar{0}, I)$ which is always positive. Therefore, $L_{\mu_k}(f)$ is monotonically decreasing with respect to $k$, and we have that $L_\mu(f) < L_{\mu_k}(f)$ for all $0 < k < 1$.

For the true distribution $\mathcal{D}$, since $l$ is 0-1 loss, then for a given model $f$, $l(f, z)$ is Bernoulli distributed with mean $p_{Ber} = \mathbb{E}_{z \sim \mathcal{D}} l(f, z) = L_\mathcal{D}(f)$ and variance $\sigma_f^2 = p_{Ber}(1 - p_{Ber}) = L_\mathcal{D}(f)(1 - L_\mathcal{D}(f))$. Therefore, the expected variance for a given model $f \in R_{set}(\mathcal{F}, \gamma)$ on distributions $\mathcal{D}_\mu$ and $\mathcal{D}_{\mu_k}$ obeys:

$$\sigma^2(f, \mu) = L_\mu(f)(1 - L_\mu(f))$$

$$< L_{\mu_k}(f)(1 - L_{\mu_k}(f))$$
$$= \sigma^2(f, \mu_k),$$

where the inequality arises from the fact that the parabola $x(1-x)$ is monotonically increasing along the interval $x \in [0, 0.5]$, and $\mu_k = k\mu$ is closer to $\bar{0}$ than $\mu$. $\qquad\square$

Note, that we can generalize Theorem 15 to the case when $\Sigma$ is any positive-definite matrix that is not necessarily diagonal (covariance matrices are always positive semi-definite, and we now additionally assume that $\Sigma$ does not have zero eigenvalues). Since $\Sigma$ is real and symmetric, by the spectral theorem, there exists an orthogonal matrix $Q \in \mathbb{R}^{m \times m}$ such that $D = Q\Sigma Q^T$ where $D$ is diagonal and contains eigenvalues of $\Sigma$. The diagonal elements of $D$ must be real and positive since $\Sigma$ is positive-definite. Then, consider the data distribution $(Q\mathcal{X}) \times \mathcal{Y}$. From the properties of the Gaussian distribution, $Q\mathcal{X}$ is Gaussian with mean $Q\mu$ and covariance matrix $Q\mathcal{X}Q^T = D$. Thus, we can generalize the results of Theorem 15 to apply to positive-definite non-diagonal matrices $\Sigma$.

For a fixed model, we additionally verify the results of Theorem 15 empirically, by generating Gaussian distributions and introducing margin noise by moving the Gaussians closer together (see Figure 5(b).) The variance of losses increases with additive and uniform random attribute noise as well, as we show empirically in Figure 5(c)-(d).

While the results of Theorems 12, 15 are for a given and fixed model $f$, they hold for the $f$ that achieves the maximum variance in the Rashomon set as well, meaning that Corollary 3 extends to Theorems 12, 15.

## B    Bernstein's and Hoeffding's inequalities

In this section, we compare Bernstein's and Hoeffding's inequalities and show that, under certain assumptions on variance, Bernstein's inequality is tighter than Hoeffding's inequality. We provide Bernstein's inequality in Lemma 16 and Hoeffding's inequality in Lemma 17.

**Lemma 16** (Bernstein's inequality for loss class). *Consider a hypothesis space $\mathcal{F}$. For a fixed $f \in \mathcal{F}$, let loss $l$ be bounded by $C > 0$ such that $|l(f, z)| \leq C$ for every $z \in \mathcal{Z}$. For any $\varepsilon > 0$,*

$$P\left(L(f) - \hat{L}(f) > \varepsilon\right) \leq e^{\frac{-n\varepsilon^2}{2\sigma_f^2 + 2C\varepsilon/3}}, \tag{4}$$

*where $\sigma_f^2 = \mathrm{Var}_{z \sim \mathcal{D}}\, l(f, z)$, and $n$ is number of samples in $S = \{z_i\}_{i=1}^n \sim \mathcal{D}$.*

**Lemma 17** (Hoeffding's inequality for loss class). *Consider a hypothesis space $\mathcal{F}$. For a fixed $f \in \mathcal{F}$, let loss $l$ be bounded by $a, b \geq 0$ such that $a \leq l(f, z) \leq b$ for every $z \in \mathcal{Z}$. For any $\varepsilon > 0$,*

$$P\left(L(f) - \hat{L}(f) > \varepsilon\right) \leq e^{\frac{-2n\varepsilon^2}{(b-a)^2}}, \tag{5}$$

*where $n$ is the number of samples in $S = \{z_i\}_{i=1}^n \sim \mathcal{D}$.*

Note that for 0-1 loss in the lemmas above, $a = 0$, $b = 1$, and $C = 1$. Now we show that Bernstein's inequality is stronger than Hoeffding's if variance is lower than $\frac{(b-a)^2}{12}$.

**Theorem 18** (Bernstein's inequality is stronger than Hoeffding's for lower variance). *For a fixed $f \in \mathcal{F}$, let loss $l \in [a, b]$ so that $a \leq l(f, z) \leq b$ for every $z \in \mathcal{Z}$. Then, Bernstein's inequality is stronger than Hoeffding's inequality for all $\varepsilon \in (0, b - a)$ if $\sigma_f^2 \leq \frac{(b-a)^2}{12}$ or if $\left|L(f) - \frac{a+b}{2}\right| > \frac{b-a}{\sqrt{6}}$ where $\sigma_f^2 = \mathrm{Var}_{z \sim \mathcal{D}}\, l(f, z)$.*

Note that since the true risk and empirical risk can only differ by at most $b - a$, $\epsilon$ is not meaningful if $\epsilon \geq b - a$.

*Proof.* According to Hoeffding's inequality (5), we have that

$$P\left(\left|L(f) - \hat{L}(f)\right| > \varepsilon\right) \leq 2e^{\frac{-2n\varepsilon^2}{(b-a)^2}}.$$

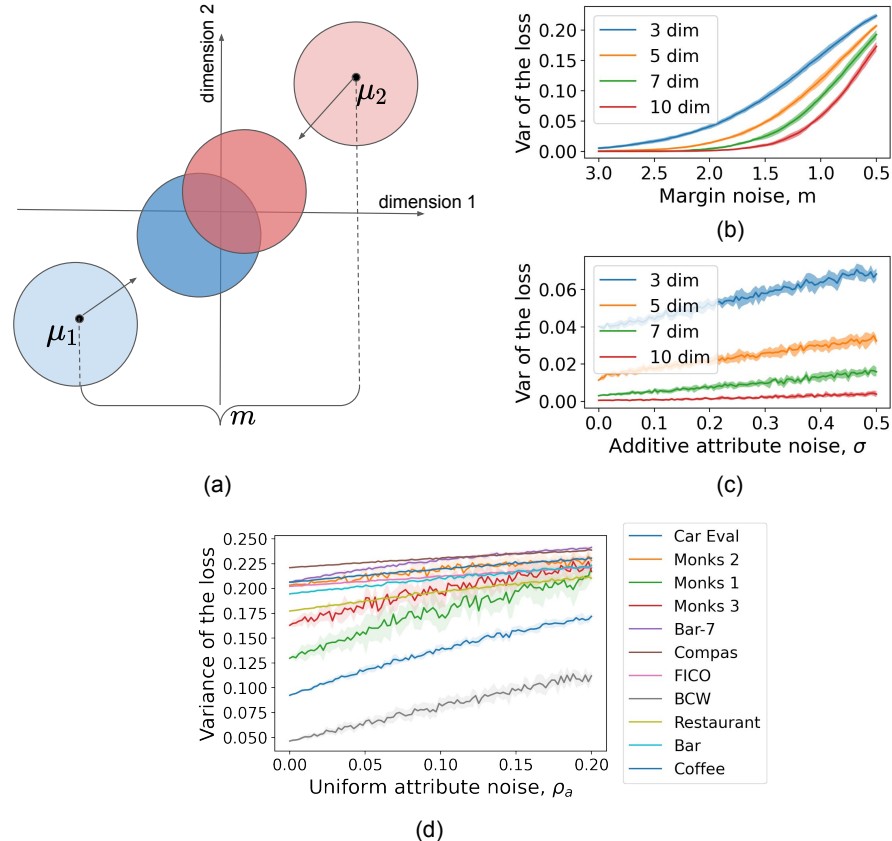

(a)   (b)   (c)

(d)

Figure 5: The variance of losses increases with margin (b) and additive attribute (c, d) noise. For (b) and (c) we generated data from Gaussians in 3, 5, 7, and 10 dimensions. For margin noise (b), as illustrated in (a), the negative class is generated from $\mathcal{N}(\bar{\mu_1}, I)$ and positive from $\mathcal{N}(\bar{\mu_2}, I)$, where $I$ is the identity matrix, $\bar{\mu}_1 = -m/2 \times \bar{1}$, $\bar{\mu}_2 = m/2 \times \bar{1}$, and $m$ controls the distance between Gaussians that determines the amount of margin noise. For additive noise, data is generated from $\mathcal{N}(\bar{0}, I)$ and $\mathcal{N}(\bar{2}, I)$. The noise model is $x' = x + \epsilon$, where $\epsilon \sim \mathcal{N}(\bar{0}, \sigma I)$ is the noise vector added to every sample and $\sigma$ determines how much noise is added to the data. For evaluation, as a fixed model we consider a random linear model from the Rashomon set. For (d), we chose 3 features with the highest AUC value and introduced uniform noise by negating the attribute values with probability $\rho_a$. As a fixed model, we consider a tree generated by the CART algorithm that uses at least one of the features to which noise was applied (this is because if the model does not use these features, the variance of losses for that model will not change). All plots are based on 0-1 loss and are averaged over 10 iterations.

Recall that Bernstein's inequality (4) states

$$P\left(\left|L(f) - \hat{L}(f)\right| > \varepsilon\right) \leq 2e^{\frac{-n\varepsilon^2}{2\sigma_f^2 + 2C\varepsilon/3}}$$

where $C = \frac{b-a}{2}$. Without loss of generality, let $l'(f, z) = l(f, z) - \frac{a+b}{2}$ so that $l' \in [-C, C]$. Then, we get that $L'(f) = L(f) - \frac{a+b}{2}$, $\text{Var}_{z \sim \mathcal{D}} l'(f, z) = \text{Var}_{z \sim \mathcal{D}} l(f, z)$, and $\hat{L}'(f) = \hat{L}(f) - \frac{a+b}{2}$. Therefore, we can rewrite Bernstein's inequality as

$$P\left(\left|L(f) - \hat{L}(f)\right| > \varepsilon\right) = P\left(\left|L'(f) - \hat{L}'(f)\right| > \varepsilon\right) \leq 2e^{\frac{-2n\varepsilon^2}{4\sigma_f^2 + 2(b-a)\varepsilon/3}}.$$

Consider $\sigma_f^2 \leq \frac{(b-a)^2}{12}$. Then, we can upper-bound the right side of Bernstein's inequality by

$$2e^{-\frac{2n\varepsilon^2}{4\sigma_f^2 + 2(b-a)\varepsilon/3}} < 2e^{-\frac{2n\varepsilon^2}{(b-a)^2/3 + 2(b-a)^2/3}} = 2e^{\frac{-2n\varepsilon^2}{(b-a)^2}},$$

where $2e^{\frac{-2n\varepsilon^2}{(b-a)^2}}$ is the bound given by Hoeffding's inequality. Therefore, we showed that, if $\sigma_f^2 \leq \frac{(b-a)^2}{12}$, then Bernstein's inequality is stronger than Hoeffding's inequality for all $\varepsilon \in (0, b-a)$.

We now consider $\left|L(f) - \frac{a+b}{2}\right| > \frac{b-a}{\sqrt{6}}$. Recall that $L'(f) = L(f) - \frac{a+b}{2}$, so we can rewrite this as $|L'(f)| > \frac{b-a}{\sqrt{6}}$. Since $-C \leq l'(f, z) \leq C$, we know that

$$
\begin{aligned}
\mathrm{Var}_{z\sim\mathcal{D}}(l'(f, z)) &= E_{z\sim\mathcal{D}}((l'(f, z))^2) - (E_{z\sim\mathcal{D}}(l'(f, z)))^2 \\
&\leq C^2 - (L'(f))^2 \\
&\leq \frac{(b-a)^2}{4} - \frac{(b-a)^2}{6} \\
&= \frac{(b-a)^2}{12}.
\end{aligned}
$$

Then, we can follow the same argument as in the previous case to conclude that Bernstein's inequality is stronger than Hoeffding's inequality for all $\varepsilon \in (0, b-a)$. $\qquad\square$

## C  Proof for Theorem 4

For a discrete hypothesis space, Theorem 4 bounds generalization error with a term that depends on the size of the true Rashomon set and maximum variance of the loss. Recall Theorem 4.

**Theorem 4** (Variance-based "generalization bound"). *Consider dataset S, 0-1 loss l , and finite hypothesis space $\mathcal{F}$. With probability at least $1 - \delta$, we have that for every $f \in R_{set}(\mathcal{F}, \gamma)$:*

$$
L(f) - \hat{L}(f) \leq \frac{2}{3n} \log\left(\frac{|R_{set}(\mathcal{F}, \gamma)|}{\delta}\right) + \sqrt{\frac{2\sigma^2}{n} \log\left(\frac{|R_{set}(\mathcal{F}, \gamma)|}{\delta}\right)},
$$

*where $\sigma^2 = \sup_{f \in R_{set}(\mathcal{F}, \gamma)} \mathrm{Var}_{z\sim\mathcal{D}} l(f, z)$, and n is number of samples in $S = \{z_i\}_{i=1}^n \sim \mathcal{D}$.*

*Proof.* For each fixed model $f \in R_{set}(\mathcal{F}, \gamma)$ in the true Rashomon set, from Bernstein's inequality, using that the maximum value for the 0-1 loss is 1, we have that

$$
P(L(f) - \hat{L}(f) > \varepsilon) \leq e^{\frac{-n\varepsilon^2}{2\sigma_f^2 + 2\varepsilon/3}}.
$$

According to the union bound:

$$
\begin{aligned}
P\left(\exists f \in R_{set}(\mathcal{F}, \gamma) : L(f) - \hat{L}(f) > \varepsilon\right) &\leq \sum_{f \in R_{set}(\mathcal{F}, \gamma)} P\left(L(f) - \hat{L}(f) > \varepsilon\right) \\
&\leq \sum_{f \in R_{set}(\mathcal{F}, \gamma)} e^{\frac{-n\varepsilon^2}{2\sigma_f^2 + 2\varepsilon/3}} \\
&\leq \sum_{f \in R_{set}(\mathcal{F}, \gamma)} e^{\frac{-n\varepsilon^2}{2\sigma^2 + 2\varepsilon/3}} \\
&= |R_{set}(\mathcal{F}, \gamma)| \cdot e^{\frac{-n\varepsilon^2}{2\sigma^2 + 2\varepsilon/3}},
\end{aligned}
$$

where we used the fact that $e^{-\frac{1}{\sigma_f^2}} \leq e^{-\frac{1}{\sup_{f \in R_{set}(\mathcal{F}, \gamma)} \sigma_f^2}} = e^{-\frac{1}{\sigma^2}}$, since the exponential function is monotonic.

Let $\delta = |R_{set}(\mathcal{F}, \gamma)| e^{\frac{-n\varepsilon^2}{2\sigma^2 + 2\varepsilon/3}}$, then we have the following quadratic equation to find $\varepsilon$:

$$
\varepsilon^2 - \frac{2}{3n} \log\left(\frac{|R_{set}(\mathcal{F}, \gamma)|}{\delta}\right) \varepsilon - \frac{2\sigma^2}{n} \log\left(\frac{|R_{set}(\mathcal{F}, \gamma)|}{\delta}\right) = 0.
$$

Setting $a = \frac{2}{n} \log\left(\frac{|R_{set}(\mathcal{F}, \gamma)|}{\delta}\right)$, we find that the roots of the quadratic equation with respect to $\varepsilon$ are:

$$
\varepsilon = \frac{a}{2 \cdot 3} \pm \frac{1}{2}\sqrt{\left(\frac{a}{3}\right)^2 + 4a\sigma^2}.
$$

Since $4a\sigma^2 \geq 0$, we see that $\frac{a}{2\cdot3} - \frac{1}{2}\sqrt{\left(\frac{a}{3}\right)^2 + 4a\sigma^2} < 0$ which is not a valid root as $\varepsilon > 0$. Thus,

$$\varepsilon = \frac{1}{3n}\log\left(\frac{|R_{set}(\mathcal{F},\gamma)|}{\delta}\right) + \sqrt{\left(\frac{1}{3n}\log\left(\frac{|R_{set}(\mathcal{F},\gamma)|}{\delta}\right)\right)^2 + \frac{2\sigma^2}{n}\log\left(\frac{|R_{set}(\mathcal{F},\gamma)|}{\delta}\right)}$$

$$\leq \frac{2}{3n}\log\left(\frac{|R_{set}(\mathcal{F},\gamma)|}{\delta}\right) + \sqrt{\frac{2\sigma^2}{n}\log\left(\frac{|R_{set}(\mathcal{F},\gamma)|}{\delta}\right)},$$

where the latter inequality arises from the inequality $\sqrt{a+b} \leq \sqrt{a} + \sqrt{b}$. Therefore, we get that with probability at least $1 - \delta$:

$$\forall f \in R_{set}(\mathcal{F},\gamma): L(f) - \hat{L}(f) \leq \varepsilon = \frac{2}{3n}\log\left(\frac{|R_{set}(\mathcal{F},\gamma)|}{\delta}\right) + \sqrt{\frac{2\sigma^2}{n}\log\left(\frac{|R_{set}(\mathcal{F},\gamma)|}{\delta}\right)}.$$

$\square$

## D  Proof for Proposition 5

We recall and provide proof for Proposition 5 below.

**Proposition 5** (ERM can be close to the true Rashomon set)**.** *Assume that through the cross-validation process, we can assess $\xi$ such that $L(\hat{f}) - \hat{L}(\hat{f}) \leq \xi$ with high probability (at least $1 - \epsilon_\xi$) with respect to the random draw of data. Then, for any $\epsilon > 0$, with probability at least $1 - e^{-2n\epsilon^2} - \epsilon_\xi$ with respect to the random draw of training data, when $\xi + \epsilon \leq \gamma$, then $\hat{f} \in R_{set}(\mathcal{F},\gamma)$.*

*Proof.* For a fixed $f \in \mathcal{F}$ for 0-1 loss by Hoeffding's inequality (5):

$$P\left[\hat{L}(f) - L(f) > \epsilon\right] \leq e^{-2n\epsilon^2}.$$

Therefore, with probability at least $1 - e^{-2n\epsilon^2}$ with respect to the random draw of data, $\hat{L}(f) - L(f) \leq \epsilon$. This is true for the optimal model as well, thus with high probability $\hat{L}(f^*) - L(f^*) \leq \epsilon$.

Since $\hat{f}$ is the empirical risk minimizer, and $\epsilon + \xi \leq \gamma$ by assumption, we have that $\hat{L}(\hat{f}) \leq \hat{L}(f^*)$. We use that for two events $A$ and $B$, $P(\neg(A \cup B)) = 1 - P(A \cup B) \geq 1 - P(A) - P(B)$, where $A$ is the event that cross-validation gives us an incorrect generalization bound, and $B$ is the event that $f^*$ does not generalize. Thus, $P(A) \leq e^{2n\epsilon^2}$ and $P(B) \leq \epsilon_\xi$. Thus, with probability at least $1 - e^{-2n\epsilon^2} - \epsilon_\xi$,

$$L(\hat{f}) \leq \hat{L}(\hat{f}) + \xi \leq \hat{L}(f^*) + \xi \leq L(f^*) + \epsilon + \xi \leq L(f^*) + \gamma.$$

Therefore $\hat{f} \in R_{set}(\mathcal{F},\gamma)$.

$\square$

## E  Proof for Proposition 6

For the hypothesis space of decision trees, the number of possible decision trees in the hypothesis space grows exponentially fast with the depth of the tree and the number of features. In Proposition 6 we show that the Rashomon set growth rate is smaller than the growth rate of the hypothesis space, and thus this leads to larger Rashomon ratios for simpler hypothesis spaces. We recall Proposition 6 below.

**Proposition 6** (Rashomon ratio is larger for decision trees of smaller depth)**.** *For a dataset $S = X \times Y$ with binary feature matrix $X \in \{0,1\}^{n \times m}$, consider a hypothesis space $\mathcal{F}_d$ of fully grown trees of depth $d$. Let the number of dimensions $m < 2^{2^d}$. Assume: (Leaves are correct) all leaves in all trees in the Rashomon set have at least $\lceil\theta n\rceil$ more correctly classified points than incorrectly classified points; (Bad features) there is a set of $m_{bad} \geq d$ "bad" features such that the empirical risk minimizer of models using only the bad features is not in the Rashomon set. Then $\hat{R}_{ratio}(\mathcal{F}_{d+1},\theta) < \hat{R}_{ratio}(\mathcal{F}_d,\theta)$.*

*Proof.* The hypothesis space of fully-grown trees of depth $d$ contains

$$|\mathcal{F}_d| = 2^{2^d} \prod_{k=1}^{d} (m - k + 1)^{2^{k-1}}$$

trees, where 2 is the number of label options each leaf can have, $2^d$ is the number of leaves we have, $\prod_{k=1}^{d}$ is the product over all depth levels in a tree, $2^{k-1}$ is the number of nodes we have at that level, and $(m - (k - 1))$ is the number of options we have to choose from given that the previous features were used in the path from the root. We do not count symmetric trees, meaning that we always assume that split $= 0$ is on the left and $= 1$ is on the right.

Now let's compute the size of the Rashomon set. First, since each leaf of every tree in the Rashomon set has correctly classified $\lceil \theta n \rceil$ points more than misclassified, flipping the label of this leaf will add more than $\theta$ to the loss and thus will push the tree out of the Rashomon set. Therefore, for every tree, every leaf label is determined by the data.

Second, since the empirical risk minimizer of models using only the bad features $m_{bad}$ is not in the Rashomon set, every tree that has only features from the set $m_{bad}$ is not in the Rashomon set. Therefore, trees in the Rashomon set must have at least one "good" feature at some node, where good means that the feature is not in $m_{bad}$. The cardinality of the set of good features is $\bar{m} = m - |m_{bad}|$, then the cardinality of the Rashomon set is:

$$\hat{R}_{set}(\mathcal{F}_d, \theta) = \prod_{k=1}^{d} (m - k + 1)^{2^{k-1}} - \prod_{k=1}^{d} (m - \bar{m} - k + 1)^{2^{k-1}},$$

meaning that among all models, we do not consider those that consist of bad features only (since $m_{bad} \geq d$, there exists at least one such tree). Then the Rashomon ratio is:

$$\hat{R}_{ratio}(\mathcal{F}_d, \theta) = \frac{|\hat{R}_{set}(\mathcal{F}_d, \theta)|}{|\mathcal{F}_d|} = \frac{\prod_{k=1}^{d}(m - k + 1)^{2^{k-1}} - \prod_{k=1}^{d}(m - \bar{m} - k + 1)^{2^{k-1}}}{2^{2^d} \prod_{k=1}^{d}(m - k + 1)^{2^{k-1}}}$$

$$= \frac{1}{2^{2^d}} \left( 1 - \frac{\prod_{k=1}^{d}(m - \bar{m} - k + 1)^{2^{k-1}}}{\prod_{k=1}^{d}(m - k + 1)^{2^{k-1}}} \right)$$

$$= \frac{1}{2^{2^d}} \left( 1 - \prod_{k=1}^{d} \left( 1 - \frac{\bar{m}}{m - k + 1} \right)^{2^{k-1}} \right)$$

$$= \frac{1 - \alpha(d)}{2^{2^d}},$$

where $\alpha(d) = \prod_{k=1}^{d} \left( 1 - \frac{\bar{m}}{m - k + 1} \right)^{2^{k-1}}$. Since $d > 1, \bar{m} > 1$, and $\frac{\bar{m}}{m-k+1} > \frac{\bar{m}}{m}$ for $k > 2$, we get that

$$\alpha(d) = \prod_{k=1}^{d} \left( 1 - \frac{\bar{m}}{m - k + 1} \right)^{2^{k-1}} < \prod_{k=1}^{d} \left( 1 - \frac{\bar{m}}{m} \right)^{2^{k-1}} < 1 - \frac{\bar{m}}{m}.$$

Note as well that $\alpha(d) < 1$ for any $d$. Recall that $m < 2^{2^d}$, then for the ratio of ratios:

$$\frac{\hat{R}_{ratio}(\mathcal{F}_d, \theta)}{\hat{R}_{ratio}(\mathcal{F}_{d+1}, \theta)} = \frac{|\hat{R}_{set}(\mathcal{F}_d, \theta)|}{|\mathcal{F}_d|} \frac{|\mathcal{F}_{d+1}|}{|\hat{R}_{set}(\mathcal{F}_{d+1}, \theta)|}$$

$$= \frac{1 - \alpha(d)}{2^{2^d}} \frac{2^{2^{d+1}}}{1 - \alpha(d+1)} = 2^{2^d} \frac{1 - \alpha(d)}{1 - \alpha(d+1)}$$

$$> 2^{2^d}(1 - \alpha(d)) > 2^{2^d} \left( 1 - \left( 1 - \frac{\bar{m}}{m} \right) \right)$$

$$= 2^{2^d} \frac{\bar{m}}{m} > 2^{2^d} \frac{1}{2^{2^d}} = 1.$$

Thus we showed that $\hat{R}_{ratio}(\mathcal{F}_d, \theta) > \hat{R}_{ratio}(\mathcal{F}_{d+1}, \theta)$, meaning that the Rashomon ratio grows as we consider less deep trees.

$\square$

# F  Proof for Theorem 7

Recall Theorem 7:

**Theorem 7** (Rashomon ratio increases with noise for ridge regression). *Consider dataset $S = X \times Y$, $X$ is a non-zero matrix, and a hypothesis space of linear models $\mathcal{F} = \{f = \omega^T x, \omega \in \mathbb{R}^m, \omega^T \omega \leq \hat{L}_{\max}/C\}$. Let $\epsilon_i$, such that $\epsilon_i \sim \mathcal{N}(\bar{0}, \lambda I)$ ($\lambda > 0$, $I$ is identity matrix), be i.i.d. noise vectors added to every sample: $x_i' = x_i + \epsilon_i$. Consider options $\lambda_1 > 0$ and $\lambda_2 > 0$ that control how much noise we add to the dataset. For ridge regression, if $\lambda_1 < \lambda_2$, then the Rashomon ratios obey $\hat{R}_{ratio_{\lambda_1}}(\mathcal{F}, \theta)) < \hat{R}_{ratio_{\lambda_2}}(\mathcal{F}, \theta))$.*

*Proof.* For simplicity denote $\mathbb{E}_{\epsilon_1,\ldots,\epsilon_n \sim \mathcal{N}(\bar{0}, \lambda I)}$ as $\mathbb{E}_\epsilon$. To find the optimal solution, under added noise, we would like to minimize expected regularized least squares:

$$
\mathbb{E}_\epsilon \hat{L}(\omega) = \mathbb{E}_\epsilon \left[ \frac{1}{n} \sum_{i=1}^n ((x_i + \epsilon_i)^T \omega - y_i)^2 + C\omega^T \omega \right]
$$

$$
= \mathbb{E}_\epsilon \left[ \frac{1}{n} \sum_{i=1}^n \left( (x_i^T \omega - y_i)^2 + 2\epsilon^T \omega (x_i^T \omega - y_i) + \omega^T \epsilon_i \epsilon_i^T \omega \right) \right] + C\omega^T \omega
$$

$$
= \frac{1}{n} \sum_{i=1}^n \left( (x_i^T \omega - y_i)^2 + 2 \mathbb{E}_\epsilon \left[ \epsilon_i \right]^T \omega (x_i^T \omega - y_i) + \omega^T \mathbb{E}_\epsilon \left[ \epsilon_i \epsilon_i^T \right] \omega \right) + C\omega^T \omega
$$

$$
= \frac{1}{n} \sum_{i=1}^n \left( (x_i^T \omega - y_i)^2 + \omega^T (\lambda I) \omega \right) + C\omega^T \omega
$$

$$
= \frac{1}{n} \sum_{i=1}^n (x_i^T \omega - y_i)^2 + (C + \lambda)\omega^T \omega,
$$

where $\mathbb{E}_\epsilon \left[ \epsilon_i \epsilon_i^T \right] = \lambda I$, $I$ is identity matrix, and $E_\epsilon \left[ \epsilon_i \right] = \bar{0}$.

Therefore, adding attribute noise to the training data becomes equivalent to $\ell_2$-regularization, and the new regularization parameter is $C + \lambda$. According to Theorem 10 in Semenova et al. [39], the Rashomon volume can be computed as:

$$
\mathcal{V}(\hat{R}_{set_\lambda}(\mathcal{F}, \theta)) = \frac{(\pi\theta)^{\frac{m}{2}}}{\Gamma(\frac{m}{2} + 1)} \prod_{i=1}^m \frac{1}{\sqrt{\sigma_i^2 + C + \lambda}},
$$

where $\sigma_i$ are singular values of matrix $X$, and $\Gamma(\cdot)$ is the Gamma-function.

On the other hand, for the regularization parameter $C + \lambda$, the hypothesis space is defined as $(C + \lambda)w^T w \leq \hat{L}_{\max}$, meaning that $w^T w \leq \frac{\hat{L}_{\max}}{C+\lambda}$. The volume of the ball defined by the $\ell_2$-norm in $m$-dimensional space with radius $R$, $\|x\|_2 = \left( \sum_{i=1}^m |x_i|^2 \right)^{\frac{1}{2}} \leq R$, can be computed as:

$$
\mathcal{V}_m^2(R) = \frac{\pi^{\frac{m}{2}}}{\Gamma(\frac{m}{2} + 1)} R^m.
$$

Since for $\|\omega\|_2^2 = w^T w \leq \frac{\hat{L}_{\max}}{C+\lambda}$, we have radius $R_\lambda = \sqrt{\frac{\hat{L}_{\max}}{C+\lambda}}$, we get that the Rashomon ratio obeys:

$$
\hat{R}_{ratio_\lambda}(\mathcal{F}, \theta)) = \frac{\mathcal{V}(\hat{R}_{set_\lambda}(\mathcal{F}, \theta))}{\mathcal{V}_m^2(R_\lambda)}
$$

$$
= \frac{(\pi\theta)^{\frac{m}{2}}}{\Gamma(\frac{m}{2} + 1)} \left[ \prod_{i=1}^m \frac{1}{\sqrt{\sigma_i^2 + C + \lambda}} \right] \frac{\Gamma(\frac{m}{2} + 1)}{\pi^{\frac{m}{2}}} \frac{(C + \lambda)^{\frac{m}{2}}}{(\hat{L}_{\max})^{\frac{m}{2}}}
$$

$$
= \left( \frac{\theta}{\hat{L}_{\max}} \right)^{\frac{m}{2}} \prod_{i=1}^m \sqrt{\frac{C + \lambda}{\sigma_i^2 + C + \lambda}}.
$$

Since $0 < \lambda_1 < \lambda_2, C > 0$, without loss of generality, let $\lambda_C = \lambda_1 + C$, and $\lambda_C + \delta = \lambda_2 + C$, where $\delta = \lambda_2 - \lambda_1 > 0$. Consider function $\frac{x}{a+x}$, where $a > 0$. This function is monotonically increasing for all $x > 0$, since $\frac{\partial}{\partial x}\left(\frac{x}{a+x}\right) = \frac{a}{(a+x)^2} > 0$. Therefore, for all non-zero $\sigma_i^2$:

$$\frac{\lambda_C}{\sigma_i^2 + \lambda_C} < \frac{\lambda_C + \delta}{\sigma_i^2 + \lambda_C + \delta}.$$

Since $X$ is a non-zero matrix, there is at least one non-zero singular value $\sigma_i^2$. Given monotonicity of the square root function, we have that for the Rashomon ratios for noise levels $\lambda_1$ and $\lambda_2$:

$$\hat{R}_{ratio_{\lambda_1}}(\mathcal{F}, \theta)) = \hat{R}_{ratio_{\lambda_C}}(\mathcal{F}, \theta)) = \left(\frac{\theta}{\hat{L}_{\max}}\right)^{\frac{m}{2}} \prod_{i=1}^{m} \sqrt{\frac{\lambda_C}{\sigma_i^2 + \lambda_C}}$$

$$< \left(\frac{\theta}{\hat{L}_{\max}}\right)^{\frac{m}{2}} \prod_{i=1}^{m} \sqrt{\frac{\lambda_C + \delta}{\sigma_i^2 + \lambda_C + \delta}}$$

$$= \hat{R}_{ratio_{\lambda_C + \delta}}(\mathcal{F}, \theta)) = \hat{R}_{ratio_{\lambda_2}}(\mathcal{F}, \theta)).$$

Therefore we proved that with the additive attribute noise, the Rashomon ratio increases.

$\square$

Compared to the Rashomon ratio, the relationship between the regularization parameter and the Rashomon volume is inverted: the stronger the regularization, the smaller the Rashomon volume. This means that adding more noise leads to stronger regularization and smaller Rashomon volume. In some ways, this is consistent with what we saw in Figure 1(b), where CART preferred shorter trees in the presence of noise.

Next we show that the variance of losses (recall notation $\sigma^2(f, \mathcal{D}) = \text{Var}_{z \sim \mathcal{D}} \, l(f, z)$) increases for the least squares loss function under additive attribute noise, as in Step 1 of the path in Section 4:

**Theorem 19** (Variance of least squares loss increases with noise). *Consider dataset $S = X \times Y$, where $X \in \mathbb{R}^{n \times m}$, $Y \in \mathbb{R}^n$, $z_i = (x_i, y_i)$. Let $\epsilon_i = \{\epsilon_{ij}\}_{j=1}^m$, such that $\epsilon_{ij} \sim \mathcal{N}(0, \sigma_{\mathcal{N}}^2)$, be i.i.d. noise vectors added to every sample: $x_i' = x_i + \epsilon_i$. Consider $\sigma_{\mathcal{N}_1}^2 > 0$, $\sigma_{\mathcal{N}_2}^2 > 0$ that control how much noise is added to the dataset. For the least squares loss $l(z_i) = r_i^2 = (w^T x_i - y_i)^2$ and a fixed model $f(x) = \omega^T x$, where $\omega \in \mathbb{R}^m$, $\omega \neq \bar{0}$, the variance of losses increases with more noise: if $\sigma_{\mathcal{N}_1}^2 < \sigma_{\mathcal{N}_2}^2$, then: $\sigma^2(f, S_{\sigma_{\mathcal{N}_1}}) < \sigma^2(f, S_{\sigma_{\mathcal{N}_2}})$.*

*Proof.* For simplicity, denote $\mathbb{E}_{\epsilon_{11},\ldots,\epsilon_{1m},\ldots,\epsilon_{n1},\ldots,\epsilon_{nm}}$ as $\mathbb{E}_{\bar{\epsilon}}$, and $\mathbb{E}_{x_i, y_i}$ as $\mathbb{E}_z$. The variance of losses for the least squares loss under the additive normal noise is: $\sigma^2(f, S_{\sigma_{\mathcal{N}}}) = Var_{z, \bar{\epsilon}}\left[\left((x_i + \epsilon_i)^T \omega - y_i\right)^2\right]$. Also, for simplicity, we will omit index $i$ over samples (but keep index $j$ over the dimensions). Recall that $r = x^T \omega - y$. From the definition of the variance we have that:

$$Var_{z,\bar{\epsilon}}\left[\left((x + \epsilon)^T \omega - y\right)^2\right] = Var_{z,\bar{\epsilon}}\left[\left((x^T \omega - y) + \epsilon^T \omega\right)^2\right] = Var_{z,\bar{\epsilon}}\left[\left(r + \epsilon^T \omega\right)^2\right]$$

$$= \mathbb{E}_{z,\bar{\epsilon}}\left[\left(r + \epsilon^T \omega\right)^4\right] - \left(\mathbb{E}_{z,\bar{\epsilon}}\left[\left(r + \epsilon^T \omega\right)^2\right]\right)^2. \tag{6}$$

Since $\epsilon_j \sim \mathcal{N}(0, \sigma_{\mathcal{N}}^2)$, we have that $\mathbb{E}_{\epsilon_j}[\epsilon_j] = 0$, $\mathbb{E}_{\epsilon_j}\left[(\epsilon_j)^2\right] = \sigma_{\mathcal{N}}^2$, $\mathbb{E}_{\epsilon_j}\left[(\epsilon_j)^3\right] = 0$ (this is a property of Gaussian random variables), and $\mathbb{E}_{\epsilon_j}\left[(\epsilon_j)^4\right] = 3\sigma_{\mathcal{N}}^4$. Also recall that the multinomial theorem states:

$$\left(\sum_{j=1}^m a_j\right)^t = \sum_{k_1 + k_2 + \ldots + k_m = t} \frac{t!}{k_1! \cdot k_2! \cdot \ldots \cdot k_m!} \cdot a_1^{k_1} \cdot a_2^{k_2} \cdot \ldots \cdot a_m^{k_m}.$$

The multinomial theorem helps us to compute coefficients of the first four moments for $\epsilon^T \omega$. More specifically, for the first and second moment:

$$\mathbb{E}_{\bar{\epsilon}}\left[\epsilon^T\omega\right] = 0,$$

$$\mathbb{E}_{\bar{\epsilon}}\left[(\epsilon^T\omega)^2\right] = \mathbb{E}_{\bar{\epsilon}}\left[\left(\sum_{j=1}^{m}\epsilon_j\omega_j\right)^2\right] = \mathbb{E}_{\bar{\epsilon}}\left[\sum_{j=1}^{m}(\epsilon_j\omega_j)^2 + \sum_{j=1..m,k=1..m,j\neq k}\epsilon_j\omega_j\epsilon_k\omega_k\right]$$

$$= \sum_{j=1}^{m}\mathbb{E}_{\bar{\epsilon}}\left[\epsilon_j^2\right]\omega_j^2 = \sigma_{\mathcal{N}}^2\omega^T\omega.$$

For the third moment, notice that from the multinomial theorem, $k_1 + \cdots + k_m = 3$. Then there are three possible combinations of the values of $k_j$: some $k_a = 3$ and the rest are 0, some $k_a = 2$, $k_b = 1$, and the rest are 0, and finally some $k_a = 1$, $k_b = 1$, $k_c = 1$ and the rest are 0. All of these cases will lead to the presence of either $\epsilon_j^3$ or $\epsilon_j$ in the product. Since $\mathbb{E}_{\epsilon_j}\left[\epsilon_j\right] = 0$, and $\mathbb{E}_{\epsilon_j}\left[\epsilon_j^3\right] = 0$, we have that

$$\mathbb{E}_{\bar{\epsilon}}\left[(\epsilon^T\omega)^3\right] = 0.$$

Similarly, for $\mathbb{E}_{\bar{\epsilon}}\left[\left(\sum_{j=1}^{m}\epsilon_j\omega_j\right)^4\right]$ we get non-zero terms for some of the combinations and the others are 0. In particular, non-zero terms arise when some $k_a = 4$ and the rest are 0, and some $k_a = 2$, $k_b = 2$, and the rest are 0s. This gives us:

$$\mathbb{E}_{\bar{\epsilon}}\left[(\epsilon^T\omega)^4\right] = \mathbb{E}_{\bar{\epsilon}}\left[\left(\sum_{j=1}^{m}\epsilon_j\omega_j\right)^4\right]$$

$$= \sum_{j=1}^{m}\mathbb{E}_{\bar{\epsilon}}\left[\epsilon_j^4\right]\omega_j^4 + 6\sum_{j=1..m,k=1..m,j\neq k}\mathbb{E}_{\bar{\epsilon}}\left[\epsilon_j^2\right]\omega_j^2\mathbb{E}_{\bar{\epsilon}}\left[\epsilon_k^2\right]\omega_k^2$$

$$= 3\sigma_{\mathcal{N}}^4\sum_{j=1}^{m}\omega_j^4 + 6\sigma_{\mathcal{N}}^4\sum_{j=1..m,k=1..m,j\neq k}\omega_j^2\omega_k^2$$

$$= 3\sigma_{\mathcal{N}}^4(\omega^T\omega)^2.$$

Let's focus on the first term of the variance equation (6):

$$\mathbb{E}_{z,\bar{\epsilon}}\left[\left(r + \epsilon^T\omega\right)^4\right] = \mathbb{E}_{z,\bar{\epsilon}}\left[r^4 + 4r^3\epsilon^T\omega + 6r^2(\epsilon^T\omega)^2 + 4r(\epsilon^T\omega)^3 + (\epsilon^T\omega)^4\right]$$

$$= \mathbb{E}_z\left[r^4\right] + 4\mathbb{E}_z\left[r^3\right]\mathbb{E}_{\bar{\epsilon}}\left[\epsilon^T\omega\right] + 6\mathbb{E}_z\left[r^2\right]\mathbb{E}_{\bar{\epsilon}}\left[(\epsilon^T\omega)^2\right]$$

$$+ 4\mathbb{E}_z\left[r\right]\mathbb{E}_{\bar{\epsilon}}\left[(\epsilon^T\omega)^3\right] + \mathbb{E}_{\bar{\epsilon}}\left[(\epsilon^T\omega)^4\right]$$

$$= \mathbb{E}_z\left[r^4\right] + 6\sigma_{\mathcal{N}}^2\omega^T\omega\mathbb{E}_z\left[r^2\right] + 3\sigma_{\mathcal{N}}^4(\omega^T\omega)^2.$$

Now, we focus of the second term of the variance equation (6):

$$\left(\mathbb{E}_{z,\bar{\epsilon}}\left[\left(r + \epsilon^T\omega\right)^2\right]\right)^2 = \left(\mathbb{E}_{z,\bar{\epsilon}}\left[r^2 + 2r\epsilon^T\omega + (\epsilon^T\omega)^2\right]\right)^2$$

$$= \left(\mathbb{E}_z\left[r^2\right] + \mathbb{E}_z\left[2r\right]\mathbb{E}_{\bar{\epsilon}}\left[\epsilon^T\omega\right] + \mathbb{E}_{\bar{\epsilon}}\left[(\epsilon^T\omega)^2\right]\right)^2$$

$$= \left(\mathbb{E}_z\left[r^2\right] + \sigma_{\mathcal{N}}^2\omega^T\omega\right)^2$$

$$= \left(\mathbb{E}_z\left[r^2\right]\right)^2 + 2\sigma_{\mathcal{N}}^2\omega^T\omega\mathbb{E}_z\left[r^2\right] + \sigma_{\mathcal{N}}^4(\omega^T\omega)^2.$$

Therefore, for the variance we get that:

$$Var_{z,\bar{\epsilon}}\left[\left(r + \epsilon^T\omega\right)^2\right] = \mathbb{E}_z\left[r^4\right] + 6\sigma_{\mathcal{N}}^2\omega^T\omega\mathbb{E}_z\left[r^2\right] + 3\sigma_{\mathcal{N}}^4(\omega^T\omega)^2$$

$$- \left( \mathbb{E}_z \left[ r^2 \right] \right)^2 - 2\sigma_{\mathcal{N}}^2 \omega^T \omega \mathbb{E}_z \left[ r^2 \right] - \sigma_{\mathcal{N}}^4 (\omega^T \omega)^2$$
$$= 2\sigma_{\mathcal{N}}^4 (\omega^T \omega)^2 + 4\sigma_{\mathcal{N}}^2 \omega^T \omega \mathbb{E}_z \left[ r^2 \right] + 2 \left( \mathbb{E}_z \left[ r^2 \right] \right)^2 + \mathbb{E}_z \left[ r^4 \right] - 3 \left( \mathbb{E}_z \left[ r^2 \right] \right)^2$$
$$= 2 \left( \sigma_{\mathcal{N}}^2 \omega^T \omega + \mathbb{E}_z \left[ r^2 \right] \right)^2 + \mathbb{E}_z \left[ r^4 \right] - 3 \left( \mathbb{E}_z \left[ r^2 \right] \right)^2 .$$

Next, we will take the derivative of the variance with respect to $\sigma_{\mathcal{N}}^2$:

$$\frac{\partial}{\partial \sigma_{\mathcal{N}}^2} \left( Var_{z,\bar{\epsilon}} \left[ \left( r + \epsilon^T \omega \right)^2 \right] \right) = \frac{\partial}{\partial \sigma_{\mathcal{N}}^2} \left( 2 \left( \sigma_{\mathcal{N}}^2 \omega^T \omega + \mathbb{E}_z \left[ r^2 \right] \right)^2 + \mathbb{E}_z \left[ r^4 \right] - 3 \left( \mathbb{E}_z \left[ r^2 \right] \right)^2 \right)$$
$$= 4 \left( \sigma_{\mathcal{N}}^2 \omega^T \omega + \mathbb{E}_z \left[ r^2 \right] \right) \omega^T \omega > 0,$$

since $\sigma_{\mathcal{N}}^2 > 0$ by assumption, $\omega^T \omega > 0$ since $\omega \neq \bar{0}$, and the risk of the least squares loss $\mathbb{E}_z \left[ r^2 \right] \geq 0$. Therefore, the variance of losses for a fixed model $f = \omega^T x$ monotonically increases for $\sigma_{\mathcal{N}}^2 > 0$. Thus, for $\sigma_{\mathcal{N}_1}^2 < \sigma_{\mathcal{N}_2}^2$ we have that:

$$\sigma^2(f, S_{\sigma_{\mathcal{N}_1}}) < \sigma^2(f, S_{\sigma_{\mathcal{N}_2}}).$$

$\square$

As before, Corollary 3 is easily extendable to the results of Theorem 19, meaning that the maximum variance of losses, $\sigma^2 = \sup_{f \in \mathbb{R}_{set}(\mathcal{F}, \gamma)} Var_{z \sim \mathcal{D}} l(f, z)$, will also increase for the least squares loss under increasing additive attribute noise.

Next, we show that when the maximum variance $\sigma^2$ increases, the generalization bound becomes worse for the least squares loss and the continuous hypothesis space. Cucker and Smale [11] proved the generalization bound based on Bernstein's inequality for the least squares loss (Theorem B). We state and provide proof of the theorem for the true Rashomon set. To derive the generalization bound, we use the covering number over the true Rashomon set. Recall that for the functional space $\mathcal{F}$ and any $\epsilon > 0$, the $\ell_\infty$ *covering number* $N(\mathcal{F}, \epsilon)$ of $\mathcal{F}$ is the minimum number of balls of radius $\epsilon$, such that they can cover $\mathcal{F}$, meaning that there exist $h_1, ..., h_{N(\mathcal{F}, \epsilon)} \in \mathcal{F}$, such that for every $f \in \mathcal{F}$ there is $k \leq N(\mathcal{F}, \epsilon)$ such that $\|f - h_k\|_\infty = \max_{x \in \mathcal{X}} |f(x) - h_k(x)| \leq \epsilon$. Now we focus on the theorem.

**Theorem 20** (Variance-based "generalization bound" for least squares loss). *Consider data distribution $\mathcal{D}$ over $\mathcal{Z} = \mathcal{X} \times \mathcal{Y}$, dataset $S = \{z_i\}_{i=1}^n \sim \mathcal{D}$, hypothesis space $\mathcal{F}$, and the least squares loss $l(f, z) = (f(x) - y)^2$. Let the loss be bounded by $C^2 > 0$ such that $l(f, z) \leq C^2$ for every $z \in \mathcal{Z}$. For any $\epsilon > 0$:*

$$P \left( \sup_{f \in R_{set}(\mathcal{F}, \gamma)} L(f) - \hat{L}(f) > \varepsilon \right) \leq N \left( R_{set}(\mathcal{F}, \gamma), \frac{\epsilon}{8C} \right) e^{\frac{-n(\varepsilon/2)^2}{2\sigma^2 + C^2 \varepsilon/3}},$$

*where $\sigma^2 = \sup_{R_{set}(\mathcal{F}, \gamma)} Var_{z \in \mathcal{D}} l(f, z)$.*

*Proof.* For each fixed model $f \in R_{set}(\mathcal{F}, \gamma)$ in the true Rashomon set, from Bernstein's inequality and given that loss is bounded by $C^2$, we have that

$$P(L(f) - \hat{L}(f) > \varepsilon) \leq e^{\frac{-n\varepsilon^2}{2\sigma_f^2 + 2C^2 \varepsilon/3}}.$$

Let $B_1, \ldots B_{N\left(R_{set}(\mathcal{F}, \gamma), \frac{\epsilon}{8C}\right)}$ be an $\ell_\infty$ cover of radius $\frac{\epsilon}{8C}$ of the true Rashomon set, meaning that $R_{set}(\mathcal{F}, \gamma) \subseteq \bigcup_{k=1}^{N\left(R_{set}(\mathcal{F}, \gamma), \frac{\epsilon}{8C}\right)} B_k$, where $N \left( R_{set}(\mathcal{F}, \gamma), \frac{\epsilon}{8C} \right)$ is the covering number. Since the loss is bounded, $l(f, z) = (f(x) - y)^2 \leq C^2$, then $|f(x) - y| \leq C$. For every $f \in B_k$, we have that $\|f - h_k\|_\infty \leq \frac{\epsilon}{8C}$, where $h_k$ is the center of the ball $B_k$. Therefore:

$$(L(f) - \hat{L}(f)) - (L(h_k) - \hat{L}(h_k)) = (L(f) - L(h_k)) + (\hat{L}(h_k) - \hat{L}(f))$$
$$= \mathbb{E}_{z \sim \mathcal{D}} \left[ l(f, z) - l(h_k, z) \right] + \hat{\mathbb{E}}_{z_i \sim S} \left[ l(f, z_i) - l(h_k, z_i) \right]$$
$$= \mathbb{E}_{z \sim \mathcal{D}} \left[ (f(x) - y)^2 - (h_k(x) - y)^2 \right] + \hat{\mathbb{E}}_{z_i \sim S} \left[ (f(x_i) - y_i)^2 - (h_k(x_i) - y_i)^2 \right]$$

$$= \mathbb{E}_{z \sim \mathcal{D}} \left[ (f(x) - h_k(x)) \left( (f(x) - y) + (h_k(x) - y) \right) \right]$$
$$+ \hat{\mathbb{E}}_{z_i \sim S} \left[ (f(x_i) - h_k(x_i)) \left( (f(x_i) - y_i) + (h_k(x_i) - y_i) \right) \right]$$
$$\leq \mathbb{E}_{z \sim \mathcal{D}} \left[ \|f - h_k\|_\infty (C + C) \right] + \hat{\mathbb{E}}_{z_i \sim S} \left[ \|f - h_k\|_\infty (C + C) \right]$$
$$= 4C \|f - h_k\|_\infty \leq 4C \frac{\epsilon}{8C} = \frac{\epsilon}{2}.$$

Therefore, if $L(f) - \hat{L}(f) > \epsilon$, we have $L(h_k) - \hat{L}(h_k) \geq (L(f) - \hat{L}(f)) - \frac{\epsilon}{2} > \epsilon - \frac{\epsilon}{2} = \frac{\epsilon}{2}$. This holds for every $f \in B_k$, and thus for $\arg\sup_{f \in B_k}$ as well:

$$P \left( \sup_{f \in B_k} L(f) - \hat{L}(f) > \varepsilon \right) \leq P \left( L(h_k) - \hat{L}(h_k) > \frac{\varepsilon}{2} \right). \tag{7}$$

Since the exponential function is monotonic, $e^{-\frac{1}{\left( \sigma_{h_k}^2 \right)}} \leq e^{-\frac{1}{\sigma^2}}$. Based on the definition of the covering number, according to the union bound and (7) we have that:

$$P \left( \sup_{f \in R_{set}(\mathcal{F}, \gamma)} L(f) - \hat{L}(f) > \varepsilon \right) = P \left( \exists f \in R_{set}(\mathcal{F}, \gamma) : L(f) - \hat{L}(f) > \varepsilon \right)$$

$$\leq P \left( \bigcup_{k=1}^{N \left( R_{set}(\mathcal{F}, \gamma), \frac{\epsilon}{8C} \right)} \exists f \in B_k : L(f) - \hat{L}(f) > \varepsilon \right)$$

$$\leq \sum_{k=1}^{N \left( R_{set}(\mathcal{F}, \gamma), \frac{\epsilon}{8C} \right)} P \left( \exists f \in B_k : L(f) - \hat{L}(f) > \varepsilon \right)$$

$$\leq \sum_{k=1}^{N \left( R_{set}(\mathcal{F}, \gamma), \frac{\epsilon}{8C} \right)} P \left( L(h_k) - \hat{L}(h_k) > \frac{\varepsilon}{2} \right)$$

$$\leq \sum_{k=1}^{N \left( R_{set}(\mathcal{F}, \gamma), \frac{\epsilon}{8C} \right)} e^{\frac{-n(\varepsilon/2)^2}{2\sigma_{h_k}^2 + C^2 \varepsilon/3}}$$

$$\leq \sum_{k=1}^{N \left( R_{set}(\mathcal{F}, \gamma), \frac{\epsilon}{8C} \right)} e^{\frac{-n(\varepsilon/2)^2}{2\sigma^2 + C^2 \varepsilon/3}}$$

$$= N \left( R_{set}(\mathcal{F}, \gamma), \frac{\epsilon}{8C} \right) e^{\frac{-n(\varepsilon/2)^2}{2\sigma^2 + C^2 \varepsilon/3}}.$$

Therefore we obtained the desired bound. $\qquad\square$

Since $e^{-1/x}$ monotonically increases for $x > 0$, in Theorem 20, as the maximum variance of losses increases, the bound on the right side increases as well, and thus the generalization bound becomes worse.

## G  Pattern Diversity and Other Metrics of the Rashomon set

In this appendix, we discuss similarities and differences between pattern diversity and pattern Rashomon ratio as well as expected pairwise disagreement (as in [5]).

**Pattern Rashomon ratio**. Pattern Rashomon ratio measures how expressive the Rashomon set is compared to the whole hypothesis space. Interestingly, for the hypothesis space of linear models, for different datasets with the same number of samples and attributes, as long as no $m - 1$ points are collinear, the denominator of the pattern Rashomon ratio is the same and equal to $2 \sum_{i=0}^{m} \binom{n-1}{i}$ [10]. If we focus only on the numerator of the pattern Rashomon ratio, it is the number of distinct predictions, whereas the pattern diversity is the average Hamming distance between distinct predictions.

Intuitively, the more distinct prediction we have, the more different they are from each other, and the higher pattern diversity we should expect. However, it is not always the case, and there exists datasets such that we can have a large number of patterns with very small Hamming distance and a small number of patterns with larger Hamming distance.

Similarly to pattern diversity, we can upper-bound the number of patterns in the pattern Rashomon set by a bound that depends on the empirical risk of the empirical risk minimizer and the Rashomon parameter $\theta$. We discuss this bound in Lemma 21.

**Lemma 21.** *Given the dataset $S$ of size $n$, the pattern Rashomon set $\pi(\mathcal{F}, \theta)$, the empirical risk of the empirical risk minimizer $\hat{L}(\hat{f})$, and the Rashomon parameter $\theta$, the cardinality of the pattern Rashomon set obeys:*

$$|\pi(\mathcal{F}, \theta)| \leq \sum_{k=1}^{\lceil n\hat{L}(\hat{f}) + n\theta \rceil} \binom{n}{k}.$$

*Proof.* For every model from the Rashomon set $f$, $\hat{L}(f) \leq \hat{L}(\hat{f}) + \theta$, which means that, in the worst case, the Hamming distance between pattern $p^f$ and vector of true labels $Y = [y_i]_{i=1}^n$ is $\lceil n\hat{L}(\hat{f}) + n\theta \rceil$. Thus, patterns in the pattern Rashomon set can make one mistake, two mistakes, and so on up to $\lceil n\hat{L}(\hat{f}) + n\theta \rceil$ mistakes, which means there are at most $\sum_{k=1}^{\lceil n\hat{L}(\hat{f}) + n\theta \rceil} \binom{n}{k}$ patterns in the pattern Rashomon set. $\square$

**Expected pairwise disagreement (as in [5])**. Following [6] and [30], empirical expected pairwise disagreement $I(\hat{R}_{set}(\mathcal{F}, \theta))$ over the Rashomon set can be defined as $I(\hat{R}_{set}(\mathcal{F}, \theta)) = \mathbb{E}_{f_1, f_2 \sim \hat{R}_{set}(\mathcal{F}, \theta)} \hat{\mathbb{E}}_{z \sim S} \mathbb{1}_{[f_1(x) \neq f_2(x)]}$. Expected pairwise disagreement measures the average disagreement between every two hypotheses from the Rashomon set, while pattern diversity measures the average disagreement between two patterns from the Rashomon set. The expected pairwise disagreement is equivalent to pattern diversity when every pattern is achievable with the same probability by models from the Rashomon set. However, these metrics can be very different and we can have a small expected pairwise disagreement and larger pattern diversity as we show next.

**Proposition 22** (Same pattern diversity but different expected pairwise disagreement). *Consider finite Rashomon set $\hat{R}_{set}(\mathcal{F}, \theta)$ of size $d \geq 2$. Let $\pi(\mathcal{F}, \theta)$ be the pattern set of size $\Pi$, $2 \leq \Pi \leq d$. Assume that every pattern except $p_1$ is achievable by only one hypothesis in the Rashomon set, and thus $p_1$ is achievable by $d - \Pi + 1$ hypotheses. Let $d^*$ be the current value of $d$, then as $d \to \infty$ (for example, by replicating hypotheses that realize $p_1$ an infinite number of times), expected pairwise disagreement converges to zero, $I(\hat{R}_{set}(\mathcal{F}_d, \theta)) \to 0$, and pattern diversity does not change, $div(\hat{R}_{set}(\mathcal{F}_d, \theta)) = div(\hat{R}_{set}(\mathcal{F}_{d^*}, \theta))$.*

*Proof.* The proof proceeds in two steps.

**Pattern diversity.** As $d$ increases, the pattern set does not change, therefore for any $d \geq d^*$, $div(\hat{R}_{set}(\mathcal{F}_d, \theta)) = div(\hat{R}_{set}(\mathcal{F}_{d^*}, \theta))$.

**Expected pairwise disagreement.** Given a pattern $p \in \pi(\mathcal{F}, \theta)$, let $P_{f \sim \hat{R}_{set}(\mathcal{F}, \theta)} [p = p^f]$ be a probability with which this pattern is achieved by models from the Rashomon set. Since support for all patterns except $p_1$ is 1, then $P_k = P_{f \sim \hat{R}_{set}(\mathcal{F}, \theta)} [p_k = p^f] = \frac{1}{d}$ for $k = 2..\Pi$. And for $p_1$ we have $P_1 = P_{f \sim \hat{R}_{set}(\mathcal{F}, \theta)} [p_1 = p^f] = \frac{d - \Pi + 1}{d}$. Then expected pairwise disagreement:

$$I(\hat{R}_{set}(\mathcal{F}_d, \theta)) = \mathbb{E}_{f_1, f_2 \sim \hat{R}_{set}(\mathcal{F}, \theta)} \hat{\mathbb{E}}_{x \sim S} \mathbb{1}_{[f_1(x) \neq f_2(x)]}$$

$$= \sum_{k=1}^{\Pi} \left[ P_{f \sim \hat{R}_{set}(\mathcal{F}, \theta)} [p_k = p^f] \sum_{j=1}^{\Pi} P_{f \sim \hat{R}_{set}(\mathcal{F}, \theta)} [p_j = p^f] \frac{1}{n} H(p_k, p_j) \right]$$

$$= \left( P_{f \sim \hat{R}_{set}(\mathcal{F}, \theta)} [p_1 = p^f] \right)^2 \frac{1}{n} H(p_1, p_1)$$

$$+ 2P_{f \sim \hat{R}_{set}(\mathcal{F}, \theta)} \left[ p_1 = p^f \right] \sum_{j=2}^{\Pi} P_{f \sim \hat{R}_{set}(\mathcal{F}, \theta)} \left[ p_j = p^f \right] \frac{1}{n} H(p_1, p_j)$$

$$+ \sum_{k=2}^{\Pi} \left[ P_{f \sim \hat{R}_{set}(\mathcal{F}, \theta)} \left[ p_k = p^f \right] \sum_{j=2}^{\Pi} P_{f \sim \hat{R}_{set}(\mathcal{F}, \theta)} \left[ p_j = p^f \right] \frac{1}{n} H(p_k, p_j) \right]$$

$$= 0 + 2 \frac{d - \Pi + 1}{d^2} \sum_{j=2}^{\Pi} \frac{1}{n} H(p_1, p_j) + \frac{1}{d^2} \sum_{k=2}^{\Pi} \sum_{j=2}^{\Pi} \frac{1}{n} H(p_k, p_j)$$

$$= \frac{1}{d} \left( 2 \left( 1 - \frac{\Pi - 1}{d} \right) \sum_{j=2}^{\Pi} \frac{1}{n} H(p_1, p_j) + \frac{1}{d} \sum_{k=2}^{\Pi} \sum_{j=2}^{\Pi} \frac{1}{n} H(p_k, p_j) \right).$$

Therefore, as $d \to \infty$, $I(\hat{R}_{set}(\mathcal{F}_d, \theta)) \to 0$. $\square$

As we see from Proposition 22, we can change expected pairwise disagreement, for example, by adding multiple copies of the same functions $f$ to the hypothesis space. Expected pairwise disagreement measures predictive multiplicity [6], but it has the issue we showed above that it can depend on the weighting of hypotheses in the hypothesis space. In the case we described in Proposition 22, the multiplicity is small because one subset of hypotheses (which produce the same pattern) is weighted very heavily. Thus, expected pairwise disagreement can be influenced by overparameterization or poor choice of parameter space. We further illustrate the effect of re-parameterization on pairwise disagreement on a simple one-dimensional example in Figure 6. Pattern diversity does not depend on the parameter space and is computed in the pattern space. It is not impacted by any probability distribution or weighting on the hypotheses. Moreover, we can compute the pattern diversity by enumerating all possible patterns of the given finite dataset as described in Appendix K.3. We cannot do the same for the pairwise disagreement metric without additional assumptions on the patterns' support.

## H    Proof for Theorem 9

Recall that $a_i$ is sample agreement over the pattern Rashomon set. Then we can compute the pattern diversity based on sample agreement as in Theorem 9.

**Theorem 9** (Pattern diversity via sample agreement). *For 0-1 loss, dataset S, and pattern Rashomon set $\pi(\mathcal{F}, \theta)$, pattern diversity can be computed as $div(\hat{R}_{set}(\mathcal{F}, \theta)) = \frac{2}{n} \sum_{i=1}^{n} a_i (1 - a_i)$, where $a_i = \frac{1}{\Pi} \sum_{k=1}^{\Pi} \mathbb{1}_{[p_k^i = y_i]}$ is sample agreement over the pattern Rashomon set.*

*Proof.* Let $y \in \{0, 1\}$. We can transform $y \in \{-1, 1\}$ to $\{0, 1\}$, simply by adding one and dividing by two.

Recall that Hamming distance $H(p_j, p_k) = \sum_{i=1}^{n} \mathbb{1}_{[p_j^i \neq p_k^i]}$. Alternatively, we can rewrite logical XOR as $\mathbb{1}_{[p_j^i \neq p_k^i]} = p_j^i (1 - p_k^i) + p_k^i (1 - p_j^i)$. Denote $b_i = \frac{1}{\Pi} \sum_{j=1}^{\Pi} p_j^i$, then from the pattern diversity definition:

$$div(\hat{R}_{set}(\mathcal{F}, \theta)) = \frac{1}{n \Pi \Pi} \sum_{j=1}^{\Pi} \sum_{k=1}^{\Pi} \sum_{i=1}^{n} \mathbb{1}_{[p_j^i \neq p_k^i]} =$$

$$= \frac{1}{n \Pi \Pi} \sum_{j=1}^{\Pi} \sum_{k=1}^{\Pi} \sum_{i=1}^{n} \left[ p_j^i (1 - p_k^i) + p_k^i (1 - p_j^i) \right]$$

$$= \frac{1}{n \Pi \Pi} \sum_{j=1}^{\Pi} \sum_{k=1}^{\Pi} \sum_{i=1}^{n} \left[ p_j^i + p_k^i - 2 p_k^i p_j^i \right]$$

$$= \frac{1}{n \Pi} \sum_{i=1}^{n} \sum_{j=1}^{\Pi} \left[ \frac{1}{\Pi} \sum_{k=1}^{\Pi} p_j^i + \frac{1}{\Pi} \sum_{k=1}^{\Pi} p_k^i - 2 \frac{1}{\Pi} \sum_{k=1}^{\Pi} p_k^i p_j^i \right]$$

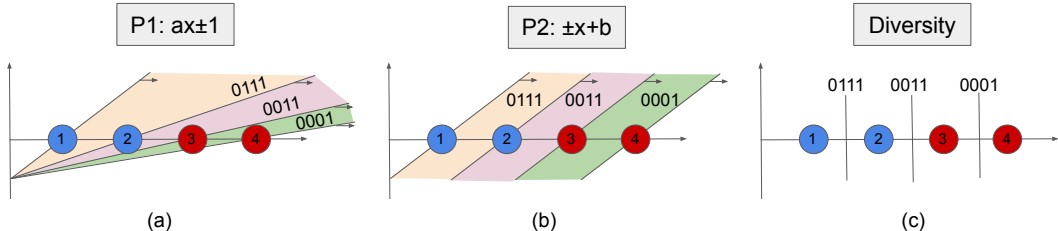

Figure 6: Illustration of how reparameterization changes pairwise disagreement metric.
Consider a separable dataset of four data points with a real-valued feature in one dimension: $S = \{(1,0),(2,0),(3,1),(4,1)\}$ and a hypothesis space of linear models. Let the Rashomon parameter be $\theta = 0.25$. There are three patterns in the Rashomon set: 0111, 0011, and 0001. The pattern diversity (c) is 0.444. Consider two different parameterizations for the hypothesis space of linear models: $ax \pm 1$ and $\pm x + b$. These two parameterizations produce the same decision boundaries for the dataset $S$. For the parameterization $\pm x + b$ (b), each pattern is achieved with the same number of models. For the parameterization $ax \pm 1$ (a), more models will support patterns that are closer to the origin. The support of each pattern is shown in a different color. The pairwise disagreement metric is 0.321 for $ax \pm 1$ and 0.444 for $\pm x + b$. (For the parameterization $ax \pm 1$, we see that the pattern 0001 occurs when $a \in (1, \frac{1}{2})$, the pattern 0011 occurs when $a \in (\frac{1}{2}, \frac{1}{3})$, and the pattern 0111 occurs when $a \in (\frac{1}{3}, \frac{1}{4})$. Therefore, the pattern 0001 has probability $w_1 = \frac{1 - \frac{1}{2}}{\frac{3}{4}} = 0.666$, the pattern 0011 has probability $w_2 = \frac{\frac{1}{2} - \frac{1}{3}}{\frac{3}{4}} = 0.222$, and the pattern 0111 has probability $w_3 = \frac{\frac{1}{3} - \frac{1}{4}}{\frac{3}{4}} = 0.111$. Recall that $H(cdot, \cdot)$ is the Hamming distance, then the pairwise disagreement metric is $w_1 w_2 H(0001, 0011) + w_1 w_3 H(0001, 0111) + w_2 w_3 H(0011, 0111) = w_1 w_2 + 2 w_1 w_3 + w_2 w_3 = 0.321$. For the parameterization $\pm x + b$, each pattern has equal probability $\frac{1}{3}$. We can then similarly calculate that the pairwise disagreement metric is 0.444). Note that if the data points are shifted together to the left, the difference in pairwise disagreement metrics for parameterizations in (a) and (b) will only grow.

$$= \frac{1}{n\Pi} \sum_{i=1}^{n} \sum_{j=1}^{\Pi} \left[ p_j^i + b_i - 2 b_i p_j^i \right]$$

$$= \frac{1}{n} \sum_{i=1}^{n} \left[ \frac{1}{\Pi} \sum_{j=1}^{\Pi} p_j^i + \frac{1}{\Pi} \sum_{j=1}^{\Pi} b_i - 2 \frac{1}{\Pi} \sum_{j=1}^{\Pi} b_i p_j^i \right]$$

$$= \frac{1}{n} \sum_{i=1}^{n} \left[ b_i + b_i - 2 b_i^2 \right]$$

$$= \frac{2}{n} \sum_{i=1}^{n} b_i (1 - b_i).$$

On the other hand, according to logical XNOR, we have that $\mathbb{1}_{[p_k^i = y_i]} = p_k^i y_i + (1 - p_k^i)(1 - y_i)$, therefore we can rewrite $a_i$ as:

$$a_i = \frac{1}{\Pi} \sum_{k=1}^{\Pi} \mathbb{1}_{[p_k^i = y_i]}$$

$$= \frac{1}{\Pi} \sum_{k=1}^{\Pi} \left[ p_k^i y_i + (1 - p_k^i)(1 - y_i) \right]$$

$$= \frac{1}{\Pi} \sum_{k=1}^{\Pi} \left[ 2 p_k^i y_i + 1 - y_i - p_k^i \right]$$

$$= 2 y_i \frac{1}{\Pi} \sum_{k=1}^{\Pi} p_k^i + 1 - y_i - \frac{1}{\Pi} \sum_{k=1}^{\Pi} p_k^i$$

$$= 2y_i b_i + 1 - y_i - b_i.$$

Since $y_i \in \{0, 1\}$, then $y_i^2 = y_i$ and we have that:

$$\frac{2}{n} \sum_{i=1}^{n} a_i(1 - a_i) = \frac{2}{n} \sum_{i=1}^{n} (2y_i b_i + 1 - y_i - b_i)(-2y_i b_i + y_i + b_i)$$

$$= \frac{2}{n} \sum_{i=1}^{n} (-4y_i b_i^2 + 2y_i b_i + 2y_i b_i^2 - 2y_i b_i + y_i + b_i$$

$$+ 2y_i b_i - y_i - y_i b_i + 2y_i b_i^2 - y_i b_i - b_i^2)$$

$$= \frac{2}{n} \sum_{i=1}^{n} (b_i - b_i^2)$$

$$= \frac{2}{n} \sum_{i=1}^{n} b_i(1 - b_i).$$

Therefore we get:

$$div(\hat{R}_{set}(\mathcal{F}, \theta)) = \frac{2}{n} \sum_{i=1}^{n} b_i(1 - b_i) = \frac{2}{n} \sum_{i=1}^{n} a_i(1 - a_i).$$

$\square$

# I    Proof for Theorem 10

Before providing the proof for Theorem 10, we show that average sample agreement over hypotheses that realize patterns in the pattern Rashomon set is negatively proportional to the average loss of these hypotheses. We use this intuition to derive an upper bound for average sample agreement and then discuss the upper bound for pattern diversity.

Let *hypothesis pattern set* $\mathcal{H}_{\pi(\mathcal{F},\theta)} \subset \hat{R}_{set}(\mathcal{F}, \theta)$ be a set of unique hypotheses corresponding to each pattern[1] in $\pi(\mathcal{F}, \theta)$, meaning that there is no $f_1^\pi, f_2^\pi \in \mathcal{H}_{\pi(\mathcal{F},\theta)}$, such that $f_1^\pi \neq f_2^\pi$, yet $p^{f_1^\pi} = p^{f_2^\pi}$.

**Theorem 23.** *Average sample agreement over the pattern Rashomon set is negatively proportional to the average loss of models in the hypothesis pattern Rashomon set $\mathcal{H}_\pi(\mathcal{F}, \theta)$,*

$$\frac{1}{n} \sum_{i=1}^{n} a_i = 1 - \hat{L}_{avg}(\mathcal{H}_\pi(\mathcal{F}, \theta)),$$

*where $\hat{L}_{avg}(\mathcal{H}_\pi(\mathcal{F}, \theta)) = \frac{1}{\Pi} \sum_{k=1}^{\Pi} \hat{L}(f_k^\pi)$. Moreover, when the Rashomon parameter $\theta = 0$, then*

$$\frac{1}{n} \sum_{i=1}^{n} a_i = 1 - \hat{L}(\hat{f}).$$

*Proof.* For a given $(x_i, y_i)$, when hypothesis $f_k^\pi$ realizes pattern $p^{f_k^\pi} = p_k$, we have that $p_k^i = f_k^\pi(x_i)$. Consider average sample agreement:

$$\frac{1}{n} \sum_{i=1}^{n} a_i = \frac{1}{n} \sum_{i=1}^{n} \frac{1}{\Pi} \sum_{k=1}^{\Pi} \mathbb{1}_{[p_k^i = y_i]}$$

$$= \frac{1}{n} \sum_{i=1}^{n} \left( 1 - \frac{1}{\Pi} \sum_{k=1}^{\Pi} \mathbb{1}_{[p_k^i \neq y_i]} \right)$$

---

[1]Since there could be many hypotheses that achieve the same pattern, $\mathcal{H}_{\pi(\mathcal{F},\theta)}$ is not unique. We can work with any of them, as $\mathcal{H}_{\pi(\mathcal{F},\theta)}$ is simply a representation of the pattern set in the hypothesis space.

$$= 1 - \frac{1}{n} \sum_{i=1}^{n} \frac{1}{\Pi} \sum_{k=1}^{\Pi} \mathbb{1}_{[p_k^i \neq y_i]}$$

$$= 1 - \frac{1}{\Pi} \sum_{k=1}^{\Pi} \frac{1}{n} \sum_{i=1}^{n} \mathbb{1}_{[f_k^\pi(x_i) \neq y_i]}$$

$$= 1 - \frac{1}{\Pi} \sum_{k=1}^{\Pi} \hat{L}(f_k^\pi)$$

$$= 1 - \hat{L}_{avg}(\mathcal{H}_\pi(\mathcal{F}, \theta)).$$

When $\theta = 0$, for any $k$, $\hat{L}(\hat{f}) = \hat{L}(f_k^\pi)$, therefore $\frac{1}{n} \sum_{i=1}^{n} a_i = 1 - \hat{L}(\hat{f})$. $\qquad\square$

Given the definition of models in the Rashomon set, we can derive an upper bound on average sample agreement in Corollary 24.

**Corollary 24.** *For any parameter $\theta > 0$, average sample agreement is upper and lower bounded by the empirical loss of the empirical risk minimizer,*

$$1 - \hat{L}(\hat{f}) - \theta \leq \frac{1}{n} \sum_{i=1}^{n} a_i \leq 1 - \hat{L}(\hat{f}).$$

*Proof.* Proof follows directly from Theorem 23 and the fact that for every model $f$ from the Rashomon set, $\hat{L}(\hat{f}) \leq \hat{L}(f) \leq \hat{L}(\hat{f}) + \theta$. $\qquad\square$

Finally, we provide proof for Theorem 10.

**Theorem 10** (Upper bound on pattern diversity). *Consider hypothesis space $\mathcal{F}$, 0-1 loss, and empirical risk minimizer $\hat{f}$. For any $\theta \geq 0$, pattern diversity can be upper bounded by*

$$div(\hat{R}_{set}(\mathcal{F}, \theta)) \leq 2(\hat{L}(\hat{f}) + \theta)(1 - (\hat{L}(\hat{f}) + \theta)) + 2\theta.$$

*Proof.* From the Cauchy–Schwarz inequality, we have that

$$\left( \sum_{i=1}^{n} a_i \right)^2 \leq \sum_{i=1}^{n} 1^2 \sum_{i=1}^{n} a_i^2 = n \sum_{i=1}^{n} a_i^2.$$

Given this and from the definition of pattern diversity and Corollary 24 we get that:

$$div(\hat{R}_{set}(\mathcal{F}, \theta)) = \frac{2}{n} \sum_{i=1}^{n} a_i(1 - a_i) = \frac{2}{n} \sum_{i=1}^{n} a_i - \frac{2}{n} \sum_{i=1}^{n} a_i^2$$

$$\leq \frac{2}{n} \sum_{i=1}^{n} a_i - \frac{2}{n^2} \left( \sum_{i=1}^{n} a_i \right)^2 = 2 \left( \frac{1}{n} \sum_{i=1}^{n} a_i \right) - 2 \left( \frac{1}{n} \sum_{i=1}^{n} a_i \right)^2$$

$$\leq 2(1 - \hat{L}(\hat{f})) - 2(1 - \hat{L}(\hat{f}) - \theta)^2$$

$$= 2 - 2\hat{L}(\hat{f}) - 2 + 4(\hat{L}(\hat{f}) + \theta) - 2(\hat{L}(\hat{f}) + \theta)^2$$

$$= 2(\hat{L}(\hat{f}) + \theta - (\hat{L}(\hat{f}) + \theta)^2 + \theta)$$

$$= 2(\hat{L}(\hat{f}) + \theta)(1 - (\hat{L}(\hat{f}) + \theta)) + 2\theta.$$

When $\theta = 0$, then $div(\hat{R}_{set}(\mathcal{F}, 0)) \leq 2\hat{L}(\hat{f})(1 - \hat{L}(\hat{f}))$. $\qquad\square$

## J  Proof for Theorem 11

We state and prove Theorem 11 below.

**Theorem 11.** *Consider a hypothesis space $\mathcal{F}$, 0-1 loss, and a dataset $S$. Let $\rho \in (0, \frac{1}{2})$ be the probability with which each label $y_i$ is flipped independently, and let $S_\rho \sim \Omega(S_\rho)$ denote a noisy version of $S$. For the Rashomon parameter $\theta \geq 0$, if $\inf_{f \in \mathcal{F}} \mathbb{E}_{S_\rho} \hat{L}_{S_\rho}(f) < \frac{1}{2} - \theta$ and $\hat{L}_S(\hat{f}_S) < \frac{1}{2}$, then adding noise to the dataset increases the upper bound on pattern diversity of the expected Rashomon set:*

$$U_{div}(\hat{R}_{set_S}(\mathcal{F}, \theta)) < U_{div}(\hat{R}_{set_{\mathbb{E}_{S_\rho \sim \Omega(S_\rho)} S_\rho}}(\mathcal{F}, \theta)).$$

*Proof.* Given the noise model, hypothesis $f$ is in the expected Rashomon set if $\mathbb{E}_{S_\rho} \hat{L}_{S_\rho}(f) \leq \inf_{f \in \mathcal{F}} \mathbb{E}_{S_\rho} \hat{L}_{S_\rho}(f) + \theta$. Let $\bar{f} \in \mathcal{F}$ be such that $\bar{f} \in \arg \inf_{f \in F} \mathbb{E}_{S_\rho} \hat{L}_{S_\rho}(f)$. Since $\rho \in (0, \frac{1}{2})$, and $\hat{L}_S(\hat{f}_S) < \frac{1}{2}$ by assumption, from (3) and the definition of the empirical risk minimizer, we have that:

$$\inf_{f \in \mathcal{F}} \mathbb{E}_{S_\rho} \hat{L}_{S_\rho}(f) = \mathbb{E}_{S_\rho} \hat{L}_{S_\rho}(\bar{f})$$
$$= (1 - 2\rho)\hat{L}_S(\bar{f}) + \rho$$
$$\geq (1 - 2\rho)\hat{L}_S(\hat{f}_S) + \rho$$
$$> \hat{L}_S(\hat{f}_S).$$

Consider $g(x) = 2(x + \theta)(1 - x - \theta) + 2\theta$. For $x \in [0, \frac{1}{2} - \theta)$, $g(x)$ is monotonically increasing, as $g'(x) = 2(1 - x - \theta) - 2(x + \theta) = 4\left(\frac{1}{2} - x - \theta\right) > 0$. Given monotonicity, assumption of the theorem $\inf_{f \in \mathcal{F}} \mathbb{E}_{S_\rho} \hat{L}_{S_\rho}(f) < \frac{1}{2} - \theta$, and since $\hat{L}_S(\hat{f}_S) < \inf_{f \in \mathcal{F}} \mathbb{E}_{S_\rho} \hat{L}_{S_\rho}(f)$, we have that

$$U_{div}(\hat{R}_{set_S}(\mathcal{F}, \theta)) = 2\left(\hat{L}_S(\hat{f}_S) + \theta\right)\left(1 - \hat{L}_S(\hat{f}_S) - \theta\right) + 2\theta$$
$$< 2\left(\inf_{f \in \mathcal{F}} \mathbb{E}_{S_\rho} \hat{L}_{S_\rho}(f) + \theta\right)\left(1 - \inf_{f \in \mathcal{F}} \mathbb{E}_{S_\rho} \hat{L}_{S_\rho}(f) - \theta\right) + 2\theta$$
$$= U_{div}(\hat{R}_{set_{\mathbb{E}_{S_\rho} S_\rho}}(\mathcal{F}, \theta)).$$

$\square$

Interestingly, in the proof of Theorem 11, the empirical risk minimizer of dataset $S$, $\hat{f}_S$, also minimizes the expected risks over noisy datasets, meaning that $\hat{f}_S \in \arg \inf_{f \in F} \mathbb{E}_{S_\rho} \hat{L}_{S_\rho}(f)$. To see this, assume that $\hat{f}_S \notin \arg \inf_{f \in F} \mathbb{E}_{S_\rho} \hat{L}_{S_\rho}(f)$, then there is $\bar{f} \in \arg \inf_{f \in F} \mathbb{E}_{S_\rho} \hat{L}_{S_\rho}(f)$, such that:

$$\mathbb{E}_{S_\rho} \hat{L}_{S_\rho}(\bar{f}) < \mathbb{E}_{S_\rho} \hat{L}_{S_\rho}(\hat{f}_S).$$

Applying (3) to both sides of the inequality above, we get that:

$$(1 - 2\rho)\hat{L}_S(\bar{f}) + \rho < (1 - 2\rho)\hat{L}_S(\hat{f}_S) + \rho,$$

which after simplification becomes:

$$\hat{L}_S(\bar{f}) < \hat{L}_S(\hat{f}_S).$$

This is a clear contradiction, since $\hat{f}_S$ is the empirical risk minimizer on $S$, and thus $\hat{L}_S(\hat{f}_S) \leq \hat{L}_S(f)$ for any $f \in \mathcal{F}$, including $\bar{f}$. Therefore our assumption was incorrect, and $\hat{f}_S \in \arg \inf_{f \in F} \mathbb{E}_{S_\rho} \hat{L}_{S_\rho}(f)$.

## K  Setup for experiments

### K.1  Datasets Description

Please see Table K.1 for the description of datasets used in the paper and all the processing steps. We normalize all real-valued features.

Table 1: Preprocessed datasets

| Dataset | Number of Samples | Number of Features | Notes |
|---|---|---|---|
| Car Evaluation | 1728 | 16 | We use one-hot encoding for features |
| Breast Cancer Wisconsin | 699 | 11 | We use one-hot encoding for features |
| Monks 1 | 124 | 12 | We use one-hot encoding for features |
| Monks 2 | 169 | 12 | We use one-hot encoding for features |
| Monks 3 | 122 | 12 | We use one-hot encoding for features |
| SPECT | 267 | 23 | We use one-hot encoding for features |
| COMPAS | 6907 | 13 | Processed in [52] |
| FICO | 10459 | 18 | Processed in [52] |
| Bar 7 (Coupon) | 1913 | 15 | Processed in [52] |
| Expensive Restaurant | 1417 | 16 | Processed in [52] |
| Carryout Takeaway | 2280 | 16 | Processed in [52] |
| Cheap Restaurant | 2653 | 16 | Processed in [52] |
| Coffee House | 3816 | 16 | Processed in [52] |
| Bar | 1913 | 16 | Processed in [52] |
| Telco Bin | 7043 | 6 | We use only the binary features |
| Iris | 100 | 4 | We consider classes Versicolour and Setosa |
| Wine | 130 | 13 | |
| Wine 4 | 130 | 4 | We use PCA to create 4 features |
| Seeds 4 | 140 | 4 | We consider classes 1 and 2 and use PCA to create 4 features |
| Immunotherapy 4 | 90 | 4 | [21, 22]. We use one-hot encoding for feature "type". We use PCA to create 4 features |
| Penguin 4 | 265 | 4 | We use one-hot encoding for feature "island." We consider classes "adelie" and "gentoo" only and use PCA to create 4 features |
| Digits 0-4 4 | 359 | 4 | We consider digit 0 and digit 4. We use PCA to create 4 features |

## K.2 Illustration of Cross-Validation Process in Step 3

We considered uniform label noise where each label is flipped independently with probability $\rho$. For each dataset, we performed five random splits into a train set and a validation set, where the validation set size is 20% of the number of samples. For the tree depth of CART, we considered the values $d \in \{1, \ldots, m\}$, where $m$ is the number of features for a given dataset.

For Figure 1(a), we tuned the parameters and then added noise to see what happens, which is that performance degrades. For every train/validation split we performed 5-fold cross-validation

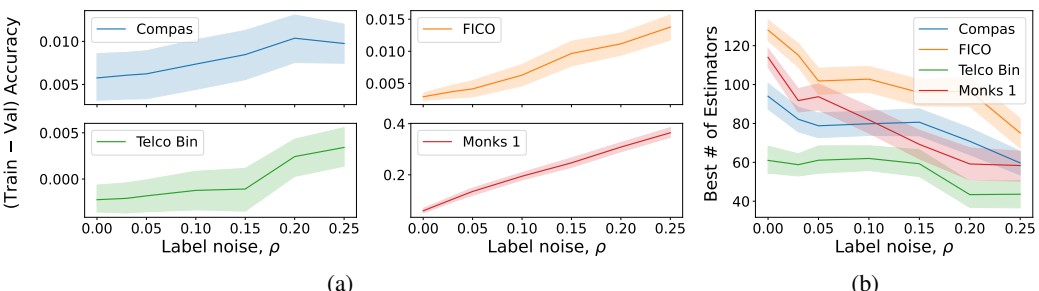

Figure 7: Practitioner's validation process in the presence of noise for gradient boosted trees. For a fixed number of estimators, as we add noise, the gap between training and validation accuracy increases (Subfigure a). As we use cross-validation to select the number of estimators, the best number of estimators decreases with noise (Subfigure b).

on the training set and computed the best depth. We fixed this depth (and thus hypothesis space). Then, we start adding noise to the dataset. We considered six different noise levels, $\rho \in \{0, 0.03, 0.05, 0.10, 0.15, 0.20.0.25\}$. For every level, we performed 25 draws of $S_\rho$. For every noise level, noise draw, and train/validation split, we evaluated train and validation performance and reported the average.

For Figure 1(b), we tuned the parameters for each noise level. We will see that noisier datasets lead us to use more regularization. We started adding noise to the dataset and then chose the best parameter based on cross-validation. More specifically, we considered six different noise levels, $\rho \in \{0, 0.03, 0.05, 0.10, 0.15, 0.20.0.25\}$. For every level, we performed 25 draws of $S_\rho$. Then we performed 5-fold cross-validation on the training data to choose the best depth for CART. For every noise level, noise draw, and train/validation split, we report mean depth based on cross-validation results.

We performed a similar experiment to Figure 1 for the gradient boosting algorithm, where we varied the number of tree estimators. We observe similar behaviors, where, with more noise, the best number of estimators (according to cross-validation) decreases. We used the same level of noise and cross-validation procedure as discussed above. For the number of estimators, we considered values $d \in \{5, 10, 20, \dots, 150\}$.

### K.3  Branch and Bound Method to Compute Patterns in the Rashomon set

Here we describe a two-step method that allows us to compute all patterns in the pattern Rashomon set. In the first step, we reduce the complexity of the problem, by discarding points that have low sample agreement. In the second step, we use a branch-and-bound approach in order to enumerate patterns and discard prefixes of those patterns that will not be in the Rashomon set based on the Rashomon parameter and the empirical risk of the empirical risk minimizer.

Consider a dataset $S = \{z_i\}_{i=1}^n$. For every point $z_i$ assume that we have sample agreement $a_i$. If $a_i = 0$, it means that all patterns in the pattern Rashomon set assign an incorrect label to sample $z_i$. On the other hand, if $a_i = 1$, all patterns assign the correct label. If we exclude all $z_i$ such that $a_i = 0$ or $a_i = 1$ from the dataset, then the number of patterns will not change in the Rashomon set. Therefore, for a given point $z_k$ ($k = 1..n$) we will try to answer a question: is there a model in the Rashomon set such that it classifies $\bar{z}_k = (x_k, -y_k)$ correctly and still stays in the Rashomon set. If there is no such model, then sample $z_k$ has no influence on the pattern Rashomon set. Since it is harder to optimize for 0-1 loss, we instead consider exponential loss. If the problem is separable by 0-1 loss, then exponential loss will converge to a separable solution exponentially fast (which is known from the convergence analysis of AdaBoost [4]). Then given hypothesis space of linear models $\mathcal{F} = \{w^T x\}$, for every $z_k$, we aim to solve following optimization problem:

$$\min \frac{1}{n} \sum_{i=0}^n e^{-y_i w^T x_i} \tag{8}$$

$$y_k w^T x_k \leq 0, \tag{9}$$

and then check if $w^T x$ is in the Rashomon set defined by 0-1 loss.

Since we optimize exponential loss, it is fast to solve the optimization problem with gradient descent. More importantly, we can run the optimization in parallel for samples $z_k$. After, we consider dataset $S_{\text{inside}}$ that contains only those samples for which models were in the Rashomon set that could accommodate misclassified $z_k$. We formally define the dicard point procedure in procedure DISCART POINTS below:

**procedure** DISCARD POINTS(dataset S, ERM $\hat{f}$, the Rashomon parameter $\theta$)
    Initialize $S_{dp}$.
    **for** every $z = (x, y) \in S$ **do**
        Solve optimization problem (8)-(9). Let $\bar{f}$ be a solution.
        **if** $\hat{L}(\bar{f}) > \hat{L}(\hat{f}) + \theta$ **then**
            add $z$ to $S_{dp}$    (this point has a single predicted label for the entire Rashomon set).
        **end if**
    **end for**
    **return** $S_{dp}$.
**end procedure**

In the second step, we build a search tree over the set of patterns that are formed by samples in $S_{\text{inside}}$. We use breadth-first search over subsets of data. We "bound" (i.e., exclude part of the search space) when the prefix of the pattern (which is the part of the dataset we are working with) misclassifies more samples than the threshold to stay in the Rashomon set, which is $\hat{L}(\hat{f}) + \theta$. Since not all patterns can be realized by the model class. We "bound" if the prefix or pattern can not be achieved (the pattern is achievable when all points with labels matching the pattern are classified correctly by some model from the hypothesis space). In order to perform branch and bound more effectively, given an empirical risk minimizer (ERM), we sort points in the dataset based on their distance to the decision boundary of the ERM. More specifically, we split the points into four categories depending on whether the point is a true positive, false positive, true negative, or false negative. Then for every category, we compute the distances from each point to the decision boundary of the ERM and then sort points from least distance to greatest distance. Finally, we cyclically choose one point from each category until all samples have been considered. Conceptually, true positive and true negative samples that are closest to the decision boundary determine most of the patterns in the pattern Rashomon set. We add false positives and false negatives early to the order of points as they are more likely to be misclassified, allowing us to bound the prefixes sooner. We describe the branch and bound procedure in Algorithm 1. We use bit vectors to represent prefixes and patterns to speed up computations. Since we apply this approach to linear models, we use logistic regression without regularization to check the achievability of the patterns and their prefixes. However, the algorithm in general can be applied to other hypothesis spaces and losses (for example, hinge loss).

---

**Algorithm 1** Branch and bound approach to find the pattern Rashomon set

---

    **Input:** The Rashomon parameter $\theta$, dataset $S = X \times Y$, ERM $\hat{f}$, algorithm $A$.
    **Output:** Pattern Rashomon set $\pi(\mathcal{F}, \theta)$.

1: Run DISCARD POINTS$(S, \hat{f}, \theta)$ to exclude points that have the same predicted label for all models in the Rashomon set. Let $S_{dp}$ be the set of discarded points, and $S_{\text{inside}}$ be the rest of the points.
2: Divide points in $S_{\text{inside}}$ into four categories: true positive, false positive, true negative, and false negative.
3: Compute the distance from the decision boundary of $\hat{f}$ to every point for every category.
4: Sort points in ascending order for every category.
5: Create a new order of the points in $S_{\text{inside}}$ by iteratively choosing points from each of the four categories until all points in $S_{\text{inside}}$ are re-ordered.
6: Concatenate $S_{dp}$ and $S_{\text{inside}}$ to form $S = X \times Y$ based on the new order, where discarded points are followed by the sorted points.
7: Initialize the prefix $p_{init}$ of length $|S_{dp}|$ based on the labels of the samples in $S_{dp}$.
8: Initialize $Q$ as the queue for the breadth-first search over the prefixes.
9: **while** $i \leq |S_{\text{inside}}|$ (loop over all points in $S_{\text{inside}}$) **do**
10:     **for** every $elem$ in $Q$ (loop over all prefixes in $Q$) **do**
11:         **for** $e \in [0, 1]$ (loop over possible labels; this is a "branch" step) **do**
12:             $Y_a = Y_{S_{dp}} \cup elem \cup e$ (consider potential prefix).
13:             Form the training data $(X_a, Y_a)$ to check if the prefix is achievable by algorithm $A$. $X_a$ consists of the first $|S_{dp}| + i$ samples of sorted $X$.
14:             Fit algorithm $A$ on $(X_a, Y_a)$ and compute $accuracy$ and $loss$.
15:             **if** $accuracy = 1$ and $loss \leq \hat{L}(\hat{f}) + \theta$ **then**
16:                 $Q.append(elem \cup e)$ (the prefix is achievable and the pattern has the potential to be achieved in the pattern Rashomon set, thus we add this element to the queue. This is a "bound" step).
17:             **end if**
18:         **end for**
19:     **end for**
20: **end while**
21: As we have now looped over all samples, $Q$ contains all the achievable patterns that are in the Rashomon set, set $\pi(\mathcal{F}, \theta) = Q$.

---

### K.4 Computation of Rashomon Ratio and Pattern Rashomon Ratio for Step 4

We considered the hypothesis space of sparse decision trees of various depths and the hypothesis space of linear models with a given number of non-zero coefficients.

For decision trees (Figure 2 (a)), we varied the depth of the maximum allowed decision tree from 1 to 7. To compute the numerator of the Rashomon ratio, we used TreeFARMS [52]. We set the Rashomon parameter to 5%. To compute the denominator of the Rashomon ratio, we considered all possible sparse decision trees up to a given depth $d$ and used the following recursive formula to compute the size of hypothesis space with $m$ features:

$$C(d, m) = 2 + mC(d - 1, m - 1)^2,$$

where $C(0, \cdot) = 2$. In the base case, the only possible trees classify every point as $0$, or every point as $1$. Then for decision trees up to depth $d$ with $m \geq d$ features, there are two cases. The first case is when the tree has depth $0$ which produces $2$ possible trees. The other case is when the tree has depth at least $1$. In this case, there are $m$ possible features to initially split on, and then the left and right subtrees are of depth at most $d - 1$ with $m - 1$ features to choose from. The left and right subtrees can be chosen independently of each other, so we have $mC(d - 1, m - 1)^2$ trees in this case, which proves the overall recursive formula.

Note that for decision trees of depth exactly equal to $d$ for every tree path, following recursive formula holds

$$C_{complete}(d, m) = mC_{complete}(d - 1, m - 1)^2,$$

which is equivalent to closed-form solution described in appendix E.

For the hierarchy of regularized linear models (Figure 2 (b)), we considered regularization for 1 non-zero coefficient, 2 non-zero coefficients, 3 non-zero coefficients, and 4 non-zero coefficients. To compute the numerator of the pattern Rashomon ratio, we used the approach described in Section K.3. We set the Rashomon parameter to 3%. To compute the denominator of the Rashomon ratio, we used the formula that gives the number of all possible patterns for the hypothesis space of linear models: if no $m - 1$ points are coplanar, $C(n, m) = 2 \sum_{i=0}^{m} \binom{n-1}{i}$ [10].

### K.5 Experiments for Pattern Diversity and Label Noise for Linear Models

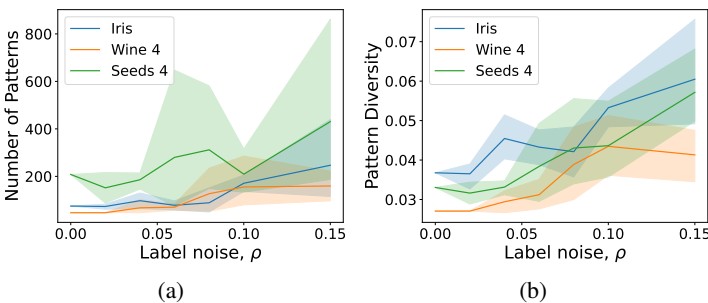

(a)                                    (b)

Figure 8: Rashomon set characteristics such as the number of patterns in the Rashomon set (Subfigure a) and pattern diversity (Subfigure b) tend to increase with uniform label noise for hypothesis spaces of linear models.

For the hypothesis space of linear classifiers, we show the pattern diversity and the number of patterns in the Rashomon set for different datasets in the presence of noise in Figure 8. We considered uniform label noise, where each label is flipped independently with probability $\rho$. We set noise level $\rho$ to values in $\{0, 0.02, 0.04, 0.06, 0.08, 0.10, 0.15\}$ and performed five draws of $S_\rho$ for every noise level. We then computed the pattern Rashomon set for each draw using the method described in Appendix K.3 and finally computed the pattern diversity. Both the number of patterns and pattern diversity tend to increase with label noise.

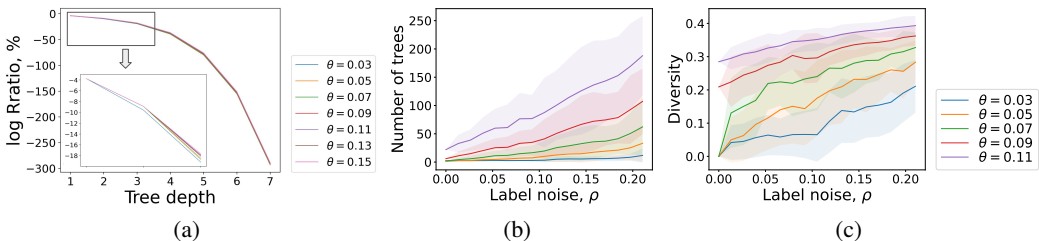

(a)                                        (b)                                        (c)

Figure 9: (a) Curve of the hypothesis space complexity vs. Rashomon ratio (as in Figure 2) stays the same shape for different Rashomon parameters for the Monks 3 dataset for the hypothesis space of sparse decision trees. (b, c) Rashomon characteristics tend to increase with uniform label noise for the hypothesis space of decision trees (as in Figure 3) for different Rashomon parameters for the Monks 3 dataset. For (b) and (c), we averaged over 25 iterations.

## K.6    The Choice of the Rashomon Parameter does not Influence Results

For Figures 2(a) and 3, we set the Rashomon parameter to be 5%. In Figure 9, on the example of Monks 3 dataset, we show that the results in Figures 2(a) and 3 hold for different values of the Rashomon parameter.

## K.7    Computation Resources

We performed experiments on Duke University's Computer Science Department cluster. We parallelized computations for the majority of the figures. It took up to 3 hours to compute the Rashomon sets for the hypothesis space of sparse decision trees for different noise levels and draws (Figures 3 and 9), and up to 48 hours for the hypothesis space of linear models (Figures 2 and 8).

