# OpenReview forum: "A Path to Simpler Models Starts With Noise"
_NeurIPS.cc/2023/Conference — NeurIPS 2023 poster_

### Official Review · Reviewer_26ns · 2023-06-26

**Soundness:** 3 good
**Presentation:** 3 good
**Contribution:** 2 fair
**Rating:** 6
**Confidence:** 3

**Summary:**

The authors propose a possible explanation for the often large Rashomon ratios in tabular datasets (criminal justice, healthcare, etc.). The explanation involves both the dataset generation process and choices made by the practitioners who train the model. The main thesis is that label noise leads to the adoption of simpler models. The authors provide empirical and theoretical justifications to support this hypothesis. In addition, the authors propose a new Rashomon metric.

**Strengths:**

1. The paper is well-written and well-organized.
1. The paper focus on an important topic with practical implications for critical domains like criminal justice, healthcare, etc.
1. The authors provide an in-depth theoretical and empirical analysis to support their claims.
1. The authors introduce a new Rashomon metric.


**Weaknesses:**

1. It would be beneficial to give some definition/example of a “simple model” in the introduction.
1. In the definition of the true Rashomon set, did you actually mean to take $L(f)<L(f^*)+\gamma$ in the definition of $R_{set}$? Otherwise, we should take $\gamma>=1$.
1. It would be interesting to see if the results generalize to more complex datasets and models.
1. Could you provide references to support the claim that the assumptions in Prop. 6 generally holds in practice?
1. How does the choice of the Rashomon coefficient affect the analysis? For example, could you provide Figures similar to Fig. 1, but for different choices of the Rashomon coeff?
1. Could you please elaborate on the motivation for defining the pattern diversity and on the connection to larger Rashomon ratios? This is not clear from Sec 5. If I understand correctly, the relation is only established through the analysis which shows that both Rashomon ratios and pattern diversity increase with noise.
1. While the arguments in Sec 4 regarding the path taken by the practitioner appear plausible, I’m not entirely convinced.

Minor:
1. Line 41: there’s a missing period before “Our”.
1. Line 88: “...that often there are often…”
1. Line 139: $p_f$ -> $p^f$.


**Questions:**

1. Is there a way to define (an equivalent to) the Rashomon ratio for a continuous hypothesis class?
1. Does the provided analysis hold for a continuous hypothesis class?


**Limitations:**

Addressed

---

> ### Author Rebuttal · Authors · 2023-08-10
>
> Thank you for the review, we really appreciate it. Please see below our response to your questions.
>
> **Weakness point 1 (example of a “simple model”)**: Yes, we will add an example to Introduction. Consider a hypothesis space of linear models with real-valued coefficients in m dimensions. Then a simpler model will be a linear model with at most m’ coefficients, where m’ is significantly less than m.
>
> **Weakness point 2 (definition of the true Rashomon set)**: Yes it is correct, thank you for identifying the typo on our side. We corrected it.
>
> **Weakness point 3 (generalization to more complex datasets and models)**: We used 20+ real-world datasets, the most complex of which involved 10K+ data points and 18 features. We expect that further advances in methods for efficiently computing the number of patterns (e.g., along the lines of Appendix K.3) will permit even larger models and datasets to be considered in the future.
>
> **Weakness point 4 (Proposition 6 assumptions)**: An example of when the leaf assumption of Proposition 6 is satisfied is the GOSDT algorithm (see  Lin et al, Generalized and Scalable Optimal Sparse Decision Trees, 2020). For GOSDT, each leaf is created only if adding this leaf will provide sufficient improvement to the loss compared to the regularization on the number of leaves. We believe that the assumption on features is not strong, as we expect that with high probability trees that have low accuracy (and thus are not in the Rashomon set) will not contain good features. (That is, the model should be at least as good as its best feature, so models that are poor can’t even have one good feature.)
>
> **Weakness point 5 (does Rashomon parameter effect analysis)**: Figure 1 actually does not depend on the Rashomon parameter. It illustrates the practitioners' validation process. Similar trends to Figure 1 hold for other hypothesis spaces as well. Regarding Figures 2 and 3 from the paper, in the global response pdf file we show that for a real-world dataset, the choice of the Rashomon parameter does not influence results (see Figure 3 in the global response file). We illustrate this point for one dataset, but it holds for the other datasets as well.
>
> **Weakness point 6 (Section 4 and 5 connection)**: That is correct. Sections 4 and 5 make two different points, where both of them show that Rashomon set metrics increase in the presence of noise.
>
> In Section 5, we wanted to show that when the practitioner does not follow the path, there are still large Rashomon sets in the presence of label noise. Therefore, we showed empirically that different Rashomon set metrics tend to increase with noise and this is a connection between diversity and Rashomon ratios. We do not have a bound that connects the two yet, and this is a part of future work. In the case of pattern diversity, we were able to bound it and show an increase in the upper bound with noise in a more general case than our empirical results.
>
> **Weakness point 7 (regarding practitioner following the path)**: With respect to Step 3, we assumed that the machine learning practitioner follows the common ML pipeline and we will add clarification to the paper by defining the practitioner as such. Thank you for pointing this out. Please note that we believe that this ML pipeline is extremely common and we haven't seen someone not following it. Reducing complexity (or regularization) is a very common tactic to prevent overfitting for various ML algorithms.
>
> **Question 1 (definition of the Rashomon ratio for a continuous hypothesis space)**: In the paper we use the pattern Rashomon ratio for continuous hypothesis space and classification, as the set of classification patterns is finite for a finite dataset (for example, for linear models see Figure 2b and Figure 4 in Appendix K.5). There is a general definition of the Rashomon ratio for continuous space (Semenova et al, On the Existence of Simple Machine Learning Models, 2022), however, it depends on the prior over the hypothesis space which might not be known.
>
> **Question 2 (does the analysis hold for continuous hypothesis space)**: Yes, let’s walk the path for the continuous hypothesis space. Step 1 (variance of the loss increases with noise) does not depend on the size of the hypothesis space. For Step 2 (higher variance leads to worse generalization), the generalization bound can be easily extended to the covering number bound when the hypothesis space is continuous (in this case, instead of the cardinality of the hypothesis space, the bound will rely on the size of the cover of the hypothesis space). Step 3 (practitioner chooses a simple space) is a regularization technique that is common for continuous spaces as well. Finally, we show empirically that Step 4 (Rashomon ratio goes up) holds for continuous hypothesis space of linear models in Figure 2b.
>
> **Minor**: Thank you for finding typos, we corrected them.
>
> Thank you again for the review. Hopefully provided answers, new analysis, and clarification addressed weaknesses and questions highlighted in the review.

---

> > ### Comment · Reviewer_26ns · 2023-08-18
> >
> > Thank you for your response. I have read the author's reply and the other reviews, and I will keep my score.

---

### Official Review · Reviewer_CJvU · 2023-06-27

**Soundness:** 3 good
**Presentation:** 3 good
**Contribution:** 3 good
**Rating:** 6
**Confidence:** 3

**Summary:**

How does noise influence the set of models that have similar performance? This paper presents a study that decomposes the problem in three stages, first showing how noise in labels harms generalisation, second, lower generalisation capability leading to more restrictive model choices, and lastly showing that the restrictive model space has a larger Rashomon ratio (i.e. larger proportion of models in this space with similar performance). The main claim is in presenting a technical argument for use of simpler (interpretable) models in critical domains, as these tend to be noisy.

**Strengths:**

- The central idea of promoting simpler models on account of noise in real-world datasets is noteworthy. The use of arguments, albeit limited, to demonstrate a plausible sequence of choices leading to simpler model specification is also interesting.

**Weaknesses:**

- The presented results use a patchwork of assumptions, model classes and conditions to hold for each result. I appreciate a comprehensive evaluation is out of scope and this is intended to be a plausible mechanism. This implies that the overall conclusions are only weakly supported.

- How does label noise influence the Rashomon ratio? Empirical evidence of this central point should be directly provided. Figure 2 breaks this down by tree depth, and Figure 3 offers characteristics of the set. The paper would benefit from a more direct noise-to-metric plot.

**Questions:**

None.

**Limitations:**

- While there are some useful analytical results, the paper largely rests on empirical work. The authors have alluded to this in their limitations.

---

> ### Author Rebuttal · Authors · 2023-08-09
>
> Thank you for appreciating the goals of our paper, which makes initial steps towards understanding broad trends that we believe have been under-studied.
>
> **Regarding weakness point 1**: Given that we can’t prove things in all generality (though of course, we would like to), we did our best to provide empirical evidence for those steps in the path we described. We hope that the combination of theory and experiments we provided in the paper are valuable and show that there are multiple angles from which this issue can be productively studied, as is often the case with interesting machine learning phenomena. If there are specific experiments you would like to see, we are happy to do them. We further hope that the gap between theory and experiments can be narrowed in future work, and welcome your suggestions for how to do so.
>
> **Regarding weakness point 2**: Figure 3 in the paper provides the noise-to-Rashomon ratio, the noise-to-pattern Rashomon ratio, and the noise-to-diversity plots for uniform random label noise. For Figures 3a and 3b, on the x-axis, we have the probability with which each label is flipped, and on the y-axis the numerator of the Rashomon set metric. Since the hypothesis space is chosen before seeing data, it does not change as we add more noise to the dataset, so the denominator of the Rashomon ratio and pattern Rashomon ratio stays the same. This means that only the numerator of the Rashomon ratio changes, which is the number of trees in the Rashomon set (Figure 3a) and the number of patterns in the Rashomon set (Figure 3b). In other words, the y-axis in Figure 3a and Figure 3b is a constant multiplied by the Rashomon ratio (3a)  and pattern Rashomon ratio (3b).
>
> Thank you again for the review.

---

### Official Review · Reviewer_y6w5 · 2023-07-01

**Soundness:** 3 good
**Presentation:** 3 good
**Contribution:** 2 fair
**Rating:** 5
**Confidence:** 4

**Summary:**

The Rashomon set is the collection of all models that perform almost equally well in a given dataset. The Rashomon ratio is the fraction of models that are members of a given hypothesis class and simultaneously are in the Rashomon set. The paper studies the relationship between noise in the data generation process and the Rashomon ratio. The authors also argue that noisier datasets lead to a larger Rashomon ratio. Finally, the authors also define a measure that captures the average difference in prediction across different classification patterns in the Rashomon set.

**Strengths:**

* The paper is well-written, and the problem of studying how data noise affects the Rashomon set is interesting.

* Proposition 6 is interesting and confirms (in a specific scenario) that more complex hypothesis classes would lead to smaller Rashomon ratios. The experiments in Figure 2 (a) and (b) help illustrate Proposition 6.

* The idea of defining a pattern Rashomon set is interesting, and using it as a proxy for exploring the Rashomon set can be helpful in other applications.

**Weaknesses:**

* Some of the provided bounds are non-computable. For example, the bounds in Theorem 4 (as the authors discussed in the paper) and Theorem 9 (the empirical risk minimizer is not always attainable) can not be computable.

* The pattern diversity metric (listed in the Abstract as a contribution) is a slight modification of the pairwise disagreement metric -- specifically, it assumes that all patterns are equiprobable. The result of Proposition 16 shows that these metrics may have different values in a particular (and maybe unrealistic) scenario. I encourage the authors to explore the differences further. Moreover, the authors could discuss when pattern diversity is useful but pairwise disagreement is not.

* The paper shows that when label noise is large enough, the loss variance is larger; hence, the Rashomon ratio may also be larger. With this chain of thought, one can conclude that simple models may exist. However, given a dataset, it needs to be clarified how to decide whether the label noise is high. Moreover, even assuming that there exists a black box that gives practitioners the level of label noise, it is necessary to answer the question: How large the noise needs to be for simpler models to exist?

 * Most of the results depend on the fact that the 0-1 loss is being used. However, other losses may be preferred in practice, and discussing how the results extend to other losses is necessary.

**Questions:**

* How would the results change when other losses (beyond the 0-1 loss) are used? Is it expected that the same/similar results will hold? It might be interesting to illustrate the results using other losses experimentally.

* How can we compute the pattern diversity metric when listing all models in the Rashomon set is impossible? For example, how to compute the pattern diversity metric for SVMs?

* How does Theorem 11 relate to Theorem 10? What should a practitioner expect when there is noise in both the label and the features?

* How tight is the upper bound in Theorem 9?

**Limitations:**

The authors discuss the limitations of the paper.

---

> ### Author Rebuttal · Authors · 2023-08-10
>
> Thank you for the review.
>
> **Weakness 1 (Theorem 4 and 9 bounds):** Theorem 4: Indeed the size of the true Rashomon set is not computable as we pointed out. We also pointed out that it doesn’t really matter - the bound still tells us what’s going on whenever the Rashomon set is large. The paper (Semenova et al, On the Existence of Simple Machine Learning Models, 2022) provides a way to assess whether the Rashomon set is large using several different ML algorithms. So, we often have indirect indicators that the Rashomon set is large, which are sufficient to give confidence that Theorem 4 is meaningful qualitatively. For Theorem 9, we actually might be able to compute the empirical risk minimizer for some hypothesis spaces (for example, it can be computed exactly for sparse decision trees, see the GOSDT algorithm of Lin et al). Even if the algorithm produces an approximation $L^u(f^u)$ of $\hat{L}(\hat{f})$, the right-hand side of equation (2) in Theorem 9 is monotonic with respect to $\hat{L}(\hat{f})$ when the loss is below 0.5. Therefore, the quantity $2L^u(f^u)(1-L^u(f^u))$ is a tight upper bound that we can compute.
>
> **Weakness 2 (pattern diversity vs pairwise disagreement):** The pairwise disagreement metric and pattern diversity work in different domains. While pairwise disagreement metric compares different functions, the pattern diversity looks into all possible unique classification patterns on the data. Pattern space allows us to avoid problems with reparameterization that can cause issues, even completely change the loss landscape [see Dihn et al, Sharp minima can generalize for deep nets, 2017]. In Figure 4 in the global response file, on a simple example, we show how different parameterizations change the pairwise disagreement metric. The advantage of pattern diversity is more evident when we think about the computation of the two metrics. We can compute the pattern diversity by enumerating all possible patterns on the given finite dataset as described in Appendix K.3. We cannot do the same for the pairwise disagreement metric without additional assumptions on patterns’ support (how many functions achieve this pattern).
>
> **Weakness 3 (the necessity of knowing when noise is high):** Interestingly, the user does not ever need to know the exact level of label noise! If they suspect label noise is present, our results say that when they try to apply an interpretable ML method, they are likely to maintain performance compared to a black box model. So, all the user needs to do is apply an interpretable ML method and see if they get a good performance. They do not need to assess label noise empirically at all.
> Our paper simply explains the reason why they would expect good performance from the simpler models. Since there is a long-held belief in an accuracy-interpretability tradeoff, our goal is to explain that such tradeoff is not always present, and why that is.
>
> **Weakness 4 (dependence on 0-1 loss):** This is true, but hopefully this is not a critical limitation of the paper. 0-1 loss is a useful starting point for understanding many ML concepts and has been actively used in theory. We acknowledge that going beyond 0-1 loss would be worthwhile future work in the Limitations section. There is an *enormous* literature just on 0-1 loss so we are not unique - this reflects that there exist many yes/no prediction problems.
>
> **Question 1 (other losses):** We tried to answer this question in the Limitations section of the paper. For least squares and ridge regression, the noise is equivalent to regularization and we show in Theorem 11 that the Rashomon volume decreases. However, the hypothesis space will change as well with the regularization, so our results might still hold in this case, we just have no way to assess it theoretically yet.
>
> **Question 2 (diversity for continuous space):** The pattern diversity relies on patterns, which are finite for finite datasets. Therefore, we can enumerate all patterns and compute pattern diversity, even if the hypothesis space is infinite. The algorithm described in Appendix K.3 will work for SVM as well as other hypothesis spaces. It will be computationally expensive for larger datasets, so additional optimizations (similar to the procedure to discard non-relevant points in Appendix K.3) might help to handle the complexity.
>
> **Question 3 (Theorem 10 and 11):** Those two theorems have different setups and are designed to make different points. Theorem 11 is for ridge regression and least squares loss, while Theorem 10 is designed for classification and 0-1 loss. For classification and 0-1 loss, if the dataset has attribute and label noise, we expect the results identified in Section 4 to hold. To support this point, we empirically show that the variance from Step 1 increases with attribute noise. We consider additive noise for data generated from Gaussians and uniform random noise for binary real-world datasets (Figure 1(b)-(c) in the global response file). Steps 2, 3, and 4 of our path in Section 4 are noise model independent. For Section 5, under attribute and label noise, we empirically observe that the characteristics tend to increase for the majority of the datasets (Figure 2 in the global response file). Because we use algorithms that regularize the number of leaves, with more attribute noise regularization can distort the trend as we get shallower trees (Figure 2d).
>
> **Question 4 (Bound in Theorem 9):** The bound holds with equality when the Rashomon parameter and the empirical risk are 0. The bound is tighter when the risk is lower and the Rashomon parameter is smaller. Since the bound is general, it becomes looser as the risk increases since we might encounter many different possible scenarios of point disagreement distributions.
>
> Thank you for the review and pointing out ways to improve our paper. Hopefully, the new analysis based on different noise models, our answers, and clarification can handle the weaknesses that you identified.

---

> > ### Comment · Reviewer_y6w5 · 2023-08-14
> >
> > I thank the authors for their answers.
> > I will increase the soundness score to 3.
> >
> > It is still unclear how a practitioner would use the author's result. In the rebuttal, the authors mention, "If they suspect label noise is present, our results say that when they try to apply an interpretable ML method, they are likely to maintain performance compared to a black box model. So, all the user needs to do is apply an interpretable ML method and see if they get a good performance". This reviewer suggests the authors add extensive experiments using various black-box and interpretable models, showing their claim is correct in practice.
> > Moreover, I suggest the authors include all discussions in the rebuttal to the main paper; this will significantly improve the clarity of the paper.

---

> > > ### Author Response · Authors · 2023-08-15
> > >
> > > Thank you, we will add the discussion points to the paper to improve its clarity.
> > >
> > > Mentioned experiments were performed in the previous literature, please see the works of Holte, Very simple classification rules perform well on most commonly used datasets, 1993; Rudin et.al., Interpretable Machine Learning: Fundamental Principles and 10 Grand Challenges, 2021; Semenova et.al., On the Existence of Simpler Machine Learning Models, 2022. Semenova et.al. compared two interpretable with three black box methods for 38 datasets (see Section 6, Appendix E) and concluded that similar performance of different complexity ML algorithms is an indirect indicator of larger Rashomon ratios. In our paper and reply, we relied on the results of mentioned previous works. Apologies for not citing this related body or works more explicitly in our rebuttal reply.
> > >
> > > In Section 5 of our paper, we show empirically that the Rashomon ratio tends to increase with noise. So putting together the two types of experiments, we get that (1) similar performance of different machine learning algorithms correlates with larger Rashomon ratios, and (2) noise can be the reason for larger Rashomon ratios. Therefore, if a practitioner suspects that noise is present in the dataset, they can use different machine learning algorithms to verify if the Rashomon ratio is likely large. However, please note that the main focus of our paper is not to measure the amount of noise in data, but rather to explain what the noisy data tell us about Rashomon ratios and simplicity.
> > >
> > > Our paper explains why machine learning practitioners are able to find simple models (including interpretable, fair, or obeying monotonicity constraints) as accurate as black boxes for a lot of real-world datasets. Because of the belief in the accuracy/interpretability trade-off, the black box models are often used, even for high-stakes decisions. Our paper is showing theoretically that the accuracy/interpretability trade-off assumption is wrong as it does not hold for a lot of datasets that are noisy. Therefore, our work can be used, for instance, to argue in court why a designer of a black box model who insists that this model is needed is most likely wrong. It’s a useful legal argument from the perspective of those struggling against black box models being used unnecessarily.
> > >
> > > Please let us know if you would like to see any other experiments. Thank you for your time, and we look forward to hearing from you.

---

> > > > ### Comment · Reviewer_y6w5 · 2023-08-18
> > > >
> > > > Thank you for the careful rebuttal.
> > > >
> > > > The authors answer my questions, and the paper's main applications are clearer to me; therefore, I will increase my score and suggest acceptance.

---

> > > > > ### Author Response · Authors · 2023-08-18
> > > > >
> > > > > Thank you for your time and for helping us improve our paper.

---

### Official Review · Reviewer_9wZ5 · 2023-07-01

**Soundness:** 3 good
**Presentation:** 3 good
**Contribution:** 3 good
**Rating:** 6
**Confidence:** 3

**Summary:**

The authors explore how the Rashomon Ratio (the fraction of all models that are in the Rashomon set) changes in the presence of label noise. They show that increased label noise causes the expected variance of the ERM’s performance to increase. Then, they hypothesize that this increased variance leads practitioners to use simpler model classes, which are known to have higher Rashomon Ratios. They also introduce pattern diversity as a metric for how diverse the prediction patterns are across members of the Rashomon set. The authors provide empirical support across several datasets that (i) simpler models perform better when a dataset is noisy, (ii) simpler model classes have higher Rashomon ratios for decision trees and linear models, and (iii) Rashomon set size, number of Rashomon patterns, and diversity of these patterns tends to increase with label noise. The authors are also forthright about the limitations of their work, and provide a nice overview of open questions related to Rashomon Ratios in the presence of noise.


**Strengths:**

* Provides one of the first theoretical studies on the Rashomon effect. As the users state, there are important policy implications of the Rashomon effect and understanding the phenomenon is important to being able to craft policy around it.
* The theoretical results are also shown to hold empirically on real datasets.
* The “limitations” section is very welcome, as it helps the reader to contextualize what this paper does and does not do. It also provides a good set of ideas for future research.
* The paper is generally well-written. The "path" framework is not perfect (see weaknesses), but I appreciate that the authors try to account for how both theoretical properties of the data and human choices lead to large Rashomon sets.

**Weaknesses:**

* Certain claims are not proven in a broad sense and this is not clearly delineated (or worse, the authors claim something more general than what they prove)
  * Uniform label noise vs real-world patterns of noise: Theorem 2 shows that variance of the loss increases with uniformly random label noise, however, in lines 167-168, the authors say that it “holds more generally,” which is not true
  * The paper is written in a way that assumes that ML practitioners are following the standard ML pipeline (e.g., switching to a simpler model class given poor test set performance) but these claims are not backed up by any data. I would encourage the authors to see if the HCI literature has studied ML practitioner behavior in this situation, and if not, to reframe the paper so that it doesn’t present the human behavior narrative as fact.
* Presentation
  * Generalization is a key part of the argument in section 4, but the paper never defines generalization or generalization bounds.
  * The discussion of learning with noise in the related works section also seemed very brief given the amount of work that this topic has received. For example, there is work that aims to learn more stable models in the presence of label noise.

**Questions:**

A couple of minor points:
* In section 5.1, wouldn’t it make more sense to call $a_i$ the sample agreement? (Or otherwise, to change its definition to sum over the indicator function with $p_k^i \neq y_i$? )
* In the equation under line 166, should $p$ (with no subscript) appear at all?

One bigger question:
* Do the results hold empirically when labels are not randomly corrupted? E.g., label errors are more likely in some subset of the population.

**Limitations:**

* Soundness limitations — see weaknesses above
* The paper only considers random, uniform label noise. This seems more acceptable as a first step for the theoretical contributions, but real-world label noise is often not random and can be correlated with protected demographic group attributes. Given that the Rashomon effect is of particular interest in social prediction tasks, I wish the authors had explored how their findings generalize to realistic noise settings.

---

> ### Author Rebuttal · Authors · 2023-08-10
>
> We thank the reviewer for the review. We address the questions point-by-point below.
>
> **Weakness point 1.1 (uniform label noise vs real-world patterns of noise in Theorem 2):** Apologies for the confusion. In the text, we were not trying to assert that Theorem 2 holds more generally, but that variance of losses generally increases with noise. To strengthen this point we added more experiments with other types of noise (please see Figure 1 in the global response pdf file). We considered two other noise models, including margin noise (when mistakes in labels are made near the decision boundary) and attribute noise (when mistakes are made in the feature matrix) noise for high dimensional Gaussians and binary real-world datasets, and show that variance of loss increases with noise.
>
> Margin noise is realistic, common, and occurs when data arises from Gaussians. Because of the central limit theorem, data often follow gaussian distributions. Also, a combination of gaussian and uniform random noise is quite realistic due to the central limit theorem and because of clerical errors causing label noise (extreme points on either side do not necessarily have perfect labels).
>
> Additionally, we generalized Theorem 2 to non-uniform noise models (please see author rebuttal for Theorem statement). Now, for a sample $(x, y)$, each label $y$ is flipped independently with probability $\rho_x$ (instead of label $y$ being flipped with uniform probability $\rho$). We show that the variance of losses increases with this non-uniform label noise. In the non-uniform noise model, noise can depend on $x$. This includes cases related to fairness where one subpopulation has much more noise than another.
>
> **Question 3: (the case when label errors are more likely in some subset of the population):** That’s a good point. Yes, our results will hold in this case. We mentioned above that we generalized Theorem 2 to the case when noise over labels is non-uniform, and each label is flipped independently with probability $\rho_x$ which depends on $x$. Such change allows one to design noise models differently for subpopulations. Note that Steps 2, 3, and 4 in the path we described in the paper are independent of the noise model.
>
> Even if we consider only uniform label noise, in practice, typically when a practitioner is facing a subgroup that is different from others, they would create a separate model for that subgroup to increase the chances of performing well on all groups. In that case, our results would also apply to each model separately.
>
> **Limitation point 2 (concern about fairness):** Besides the generalization of Theorem 2 and experiments described above, we would like to add that the Rashomon effect is actually very important for fairness [Coston et al, Characterizing Fairness Over the Set of Good Models Under Selective Labels, 2021], and knowing when the Rashomon effect is large is beneficial for the search of a fair model. Unintuitively, even if the noise generation is not fair (we do not support this happening, we are saying if data happen to be this way), large noise will still lead to larger Rashomon ratios as explained in our paper. If the practitioner knows that the Rashomon effect is large, they can explore the Rashomon set in order to find a more fair model [Coston et al, Characterizing Fairness Over the Set of Good Models Under Selective Labels, 2021].
>
> **Weakness point 1.2 (ML practitioners that follow the standard ML pipeline):** Thank you for pointing out the assumption on our part. We did assume that the machine learning practitioner follows the common ML pipeline and is knowledgeable about overfitting and common steps in preventing overfitting. We will add the assumption that we define a machine learning practitioner as someone who operates in the standard machine learning pipeline to the paper - we hope that this should solve the issue. We point out that the pipeline we assumed is extremely common and we haven’t seen someone not following this. Choosing a simpler hypothesis space (basically regularization) is a common step for essentially all ML algorithms. There also could be many ways to control the complexity of the model. For example, for tree boosting, parameters that can control complexity include a number of estimators, maximum tree depth, minimum leaf weight, and so on.
>
> **Weakness point 2.1 (generalization):** In the paper, we say that there is good generalization when the generalization error (as the difference between training and test performance, see Vladimir Vapnik, Statistical Learning Theory) is small. Correspondingly, we say that the generalization is bad when the generalization error is large. We will add the definition to the paper.
>
> **Weakness point 2.2 (learning with noise related works):** The learning with noise literature is only tangentially related, despite the similarity of the name. These papers focus on specialized techniques for cleaning up or compensating for noisy data, while our paper focuses on studying what noisy data tells us about simple models (models created with standard algorithms, not the ones specialized for specific types of noise). Thank you for pointing it out anyway, we will add more citations to the related work section in case the audience of our paper would like to learn more about algorithms with noise correction.
>
> **Question 1:** Yes, we can call a_i sample disagreement.
>
> **Question 2:** Thank you for finding a typo in line 166. We have corrected it.
>
> Thank you for the review and for identifying ways to improve our paper. We believe the definitions and clarifications handle both weaknesses you’ve pointed out. Hopefully the new analysis based on different noise models and generalized theorem additionally supports the path that noisy datasets essentially lead to larger Rashomon ratios, which enables the search for simpler (and fair) models.

---

> > ### Comment · Reviewer_9wZ5 · 2023-08-17
> >
> > Thank you for the rebuttal, I feel more confident about the paper's soundness now and will increase the scores accordingly.
> >
> > To clarify one point from my review, I agree with you that the proposed path outlines the standard ML pipeline that practitioners "should" follow. I guess my concern was more, as more and more organizations use ML, how common is it that the ML practitioners have the appropriate formal training to know that they should follow this pipeline? But I understand if this is out of scope for your paper.

---

> > > ### Author Response · Authors · 2023-08-18
> > >
> > > Thank you! Any trained ML scientist should follow this pipeline since how to control overfitting is an essential component of any introductory machine learning class. So our guess would be upwards of 99% but of course, we don’t have the data to verify that (and it might be challenging to conduct such a study and to gather a proper sample). So yes, we should declare this as well beyond the scope of our paper. Good question though.
> > >
> > > Thank you for your time and valuable feedback on our paper.

---

### Author Rebuttal · Authors · 2023-08-10

We thank all the reviewers for the reviews. Below we provide a generalization of Theorem 2 to non-uniform label noise. In the response file, we also provide additional figures and analysis.

**Theorem** (Generalized Theorem 2 to non-uniform label noise). Consider 0-1 loss $l$, infinite true data distribution $D$, and a hypothesis space $\mathcal{F}$. Assume that there exists at least one function $\bar{f}\in\mathcal{F}$ such that $L(\bar{f}) < \frac{1}{2} - \gamma$. For a fixed $f \in \mathcal{F}$, let $\sigma^2 (f, D)$ be the variance of the loss, $\sigma^2(f,D) = Var_{z\sim D} l(f,z)$ on data distribution $D$. Consider non-uniform label noise, where each label $y$ is flipped independently with probability $\rho_x$, $(x, y) \sim D$. Let $D_{\rho}$ denote the noisy version of $D$. For any $\delta >0$, let $D_{{\rho}^{\delta}}$ be a noisier data distribution than $D_{\rho}$, meaning that for every sample $(x, y)$ the probabilities of labels being flipped are higher by $\delta$: $\rho_x^{\delta}=\rho_x+\delta$. If for a fixed model $f \in R_{set}(\mathcal{F},\gamma)$, $L_{D_{{\rho}^{\delta}}}(f)<0.5$, then
$\sigma^2 (f, D_{\rho}) < \sigma^2 (f, D_{{\rho}^{\delta}}).$

---

### Decision · Program_Chairs · 2023-09-21

**Decision:**

Accept (poster)

**Comment:**

The reviewers accepted the paper unanimously and maintained their positive scores post-discussion. Exploring the Rashomon effect is an exciting research direction in machine learning. Rashomon ratios provide a concrete metric for quantifying the impact of the Rashomon effect on different aspects of ML. The paper's intriguing finding that lower complexity function classes lead to larger Rashomon ratios can impact when interpretable models *must* be used in practice, i.e., when there is "no excuse" not to use an interpretable model. The paper's findings and the discussion it promotes are a significant contribution to NeurIPS.

I encourage the authors to take the reviews seriously when reviewing their manuscript and incorporate the answers given in the rebuttal.